ARTICLES

# Type I IFNs promote cancer cell stemness by triggering the epigenetic regulator KDM1B

Martina Musella[1], Andrea Guarracino [2,3], Nicoletta Manduca[1], Claudia Galassi[1,4], Eliana Ruggiero[5], Alessia Potenza[5], Ester Maccafeo[1], Gwenola Manic [6,7], Luca Mattiello[6,7], Sara Soliman Abdel Rehim[2,6], Michele Signore[8], Marco Pietrosanto[2], Manuela Helmer-Citterich[2], Matteo Pallocca[9], Maurizio Fanciulli[10], Tiziana Bruno[10], Francesca De Nicola[10], Giacomo Corleone[10], Anna Di Benedetto[11], Cristiana Ercolani[11], Edoardo Pescarmona[11], Laura Pizzuti[12], Francesco Guidi[1,13], Francesca Sperati[14], Sara Vitale[1], Daniele Macchia[15], Massimo Spada[15], Giovanna Schiavoni [16], Fabrizio Mattei[16], Adele De Ninno[17], Luca Businaro[17], Valeria Lucarini[18], Laura Bracci[16], Eleonora Aricò[19], Giovanna Ziccheddu[20], Francesco Facchiano[16], Stefania Rossi[16], Massimo Sanchez [21], Alessandra Boe [21], Mauro Biffoni[16], Ruggero De Maria [1,13,23] ✉, Ilio Vitale [6,7,23] ✉ and Antonella Sistigu [1,13,22,23] ✉

Cancer stem cells (CSCs) are a subpopulation of cancer cells endowed with high tumorigenic, chemoresistant and metastatic potential. Nongenetic mechanisms of acquired resistance are increasingly being discovered, but molecular insights into the evolutionary process of CSCs are limited. Here, we show that type I interferons (IFNs-I) function as molecular hubs of resistance during immunogenic chemotherapy, triggering the epigenetic regulator demethylase 1B (KDM1B) to promote an adaptive, yet reversible, transcriptional rewiring of cancer cells towards stemness and immune escape. Accordingly, KDM1B inhibition prevents the appearance of IFN-I-induced CSCs, both in vitro and in vivo. Notably, IFN-I-induced CSCs are heterogeneous in terms of multidrug resistance, plasticity, invasiveness and immunogenicity. Moreover, in breast cancer (BC) patients receiving anthracycline-based chemotherapy, KDM1B positively correlated with CSC signatures. Our study identifies an IFN-I → KDM1B axis as a potent engine of cancer cell reprogramming, supporting KDM1B targeting as an attractive adjunctive to immunogenic drugs to prevent CSC expansion and increase the long-term benefit of therapy.

CSCs, also known as tumor-initiating or tumor-propagating cells, are a relatively rare stem-like cell subpopulation within the tumor capable of self-renewal and multilineage differentiation, and responsible for tumor initiation, progression, spreading and therapy resistance[1,2]. Mounting evidence indicates that CSCs can evolve over space and time leading to a high degree of genotypic, phenotypic and functional heterogeneity[2,3]. Along with this, it is emerging that non-CSC subsets can adapt to the changes in the tumor microenvironment (TME), undergoing cell reprogramming and (re)generating CSCs[2].

Epigenetic dysregulations critically affect cancer–immune cell interactions and coevolution during disease onset, progression and response to therapy by influencing cellular states and fates[4]. Not surprisingly given their role in normal stem cell maintenance, epigenetic mechanisms have also been involved in CSC preservation[5]. This feature, together with the inherent reversibility of epigenetic modifications, makes the use of epigenome-targeting drugs (epidrugs) a unique opportunity to rationally target CSCs in combination with conventional therapies[4,6–9].

One key concept in tumor immunology is that some chemotherapeutics, including (but not limited to) anthracyclines (for example, doxorubicin, DOX), oxaliplatin (OXP) and cyclophosphamide[10,11] induce cancer immunogenic cell death (ICD), a form of regulated cell death that initiates adaptive immune responses by the emission

[1]Dipartimento di Medicina e Chirurgia Traslazionale, Università Cattolica del Sacro Cuore, Rome, Italy. [2]Department of Biology, University of Rome 'Tor Vergata', Rome, Italy. [3]Genomics Research Centre, Human Technopole, Milan, Italy. [4]Department of Radiation Oncology, Weill Cornell Medical College, New York, NY, USA. [5]Experimental Hematology Unit, IRCCS San Raffaele Scientific Institute, Milan, Italy. [6]Italian Institute for Genomic Medicine (IIGM), Candiolo, Italy. [7]Candiolo Cancer Institute, FPO - IRCCS, Candiolo, Italy. [8]RPPA Unit, Proteomics Area, Core Facilities, Istituto Superiore di Sanità, Rome, Italy. [9]UOSD Clinical Trial Center, Biostatistics and Bioinformatics, IRCCS Regina Elena National Cancer Institute, Rome, Italy. [10]SAFU Unit, IRCCS Regina Elena National Cancer Institute, Rome, Italy. [11]Pathology Unit, IRCCS Regina Elena National Cancer Institute, Rome, Italy. [12]Division of Medical Oncology 2, IRCCS Regina Elena National Cancer Institute, Rome, Italy. [13]Fondazione Policlinico Universitario 'A. Gemelli' - IRCCS, Rome, Italy. [14]UOSD Clinical Trial Center, Biostatistics and Bioinformatics, IRCCS San Gallicano Dermatological Institute, Rome, Italy. [15]Center of Animal Research and Welfare, Istituto Superiore di Sanità, Rome, Italy. [16]Department of Oncology and Molecular Medicine, Istituto Superiore di Sanità, Rome, Italy. [17]Institute for Photonics and Nanotechnologies, Italian National Research Council, Rome, Italy. [18]Department of Paediatric Haematology/Oncology and of Cell and Gene Therapy, Ospedale Pediatrico Bambino Gesù, IRCCS, Rome, Italy. [19]FaBioCell, Core Facilities, Istituto Superiore di Sanità, Rome, Italy. [20]Oncogenomics and Epigenetics, IRCCS Regina Elena National Cancer Institute, Rome, Italy. [21]Cytometry Unit, Core Facilities, Istituto Superiore di Sanità, Rome, Italy. [22]Tumor Immunology and Immunotherapy Unit, IRCCS Regina Elena National Cancer Institute, Rome, Italy. [23]These authors jointly supervised this work: Ruggero De Maria, Ilio Vitale, Antonella Sistigu. ✉e-mail: ruggero.demaria@unicatt.it; ilio.vitale@gmail.com; antonella.sistigu@unicatt.it

of damage-associated molecular patterns (DAMPs)[12,13] and cytokines. In particular, the IFN-I family of proinflammatory cytokines, upon binding to the interferon α and β receptor (IFNAR), triggers the production of the IFN-stimulated gene (ISG) C–X–C motif chemokine ligand 10 (CXCL10), a chemoattractant for inflammatory monocytes and T cells[11]. Nonetheless, depending on the duration and intensity of the transduced signaling and/or the nature of the unleashed ISGs, IFN-I can also display protumorigenic effects[14], promoting the expression of the immune checkpoint (IC) ligand CD274 (best known as PD-L1)[11,15,16]. Moreover, innate immune signaling upstream of IFN-I has been associated with nuclear reprogramming and malignant transformation[17].

In this work, we elucidated the downside of IFN-I during ICD. We demonstrated that IFN-I reprograms cancer cells toward a more aggressive, stem-like phenotype by upregulating KDM1B, an epigenetic regulator also known as LSD2, which erases mono- and dimethyls on histone H3 at lysine 4 (H3K4me1 and H3K4me2)[18]. Such detrimental resetting represents a hitherto undescribed mechanism of tumor evolution, which drives acquired resistance and immune evasion.

## Results

### IFN-I administration drives enrichment and de novo induction of CSCs.
To investigate the impact of the IFN-I→IFNAR axis on the appearance of cancer cells with a stem-like phenotype (hereafter referred to as CSCs), we selected a panel of cancer cell lines of distinct origin (epithelial or mesenchymal) and species (human or mouse) and treated them for 72 h with $6 \times 10^3$ U ml$^{-1}$ IFN-I before assessing, by flow cytometry, the levels of prominin 1 (Prom1, best known as CD133), CD24 and CD44 surface markers, whose expression, alone and in combination, has been associated with putative CSCs. In this setting, we observed that IFN-I favors the enrichment of rare CD133$^+$CD24$^+$CD44$^+$ putative CSCs (IFN–CSCs) in all analyzed murine cancer cell lines. Specifically, we identified two main populations of IFN–CSCs in MCA205 sarcoma cells: the CD133$^+$CD24$^+$CD44$^{+low}$ (CD44L, ~7 times higher compared with the untreated condition, (CTR)) and the CD133$^+$CD24$^+$CD44$^{+high}$ (CD44H, ~9 times higher compared with the CTR) CSC subsets (Fig. 1a). Putative IFN–CSCs were also detected in AT3 breast carcinoma, namely the CD133$^+$CD44$^+$CD24$^{+low}$ (CD24L, ~3.5 times higher compared with the CTR) and CD133$^+$CD44$^+$CD24$^{+high}$ (CD24H, ~2.6 times higher compared with the CTR) CSC subsets, but we focused on the former, the widely recognized CSC subpopulation in breast carcinoma[19] (Fig. 1a). Similarly, we found (1) CD133$^+$CD44$^+$CD24$^+$ in CT26 colon carcinoma cell line and (2) CD133$^+$CD44$^+$CD24$^{+low}$ and CD133$^+$CD44$^+$CD24$^{+high}$ in B16. F10 melanoma cell line (Extended Data Fig. 1a). These results are in line with the intra- and intertumoral heterogeneity often

ascribed to CSCs[20]. To assess whether this phenomenon was exclusive of the murine cancer model, we treated human osteosarcoma (U2OS), breast carcinoma (MCF7, HMLER) and mammary epithelial (MCF10A) cell lines with recombinant human IFN-α2a and then analyzed the expression of standard human CSC markers. We detected IFN–CSC subpopulations in U2OS (CD133$^+$CD44$^+$ and CD44v6$^+$CD24$^+$) and MCF7 (CD44$^+$CD24$^{-low}$ and CD44v6$^+$CD24$^{-low}$) but not in the nontumorigenic MCF10A and in the highly CSC-enriched HMLER (CD44$^+$CD24$^{-low}$) (Extended Data Fig. 1b).

We then isolated MCA205 CD133$^+$ and CD133$^-$ (that is, non-CSC) cell fractions by fluorescence-activated cell sorting (FACS) and exposed them to IFN-I. By flow cytometry, we found that IFN-I treatment led to a significant increase in the CD44H and CD44L cell fraction and in the levels of the pluripotency transcription factor (TF) SRY (sex determining region Y)-box 2 (SOX2) in both the CD133$^+$ and CD133$^-$ subsets (Fig. 1b). In parallel, by quantitative PCR with reverse transcription (qRT–PCR) analyses of common stem-related TFs and CSC markers, we found that exogenous IFN-I significantly upregulates Kruppel-like factor 4 (*Klf4*), POU domain, class 5, transcription factor 1 (*Pou5f1*, best known as *Oct3/4*), *Sox2* and nestin (*Nes*) in FACS-isolated CD133$^-$ cells and Nanog homeobox (*Nanog*) in FACS-isolated CD133$^-$ and CD133$^+$ cells (Fig. 1b). These results suggest that IFN-I-mediated CSC enrichment depends on the co-occurrence of positive selection of rare, pre-existing CSCs and de novo generation of CSCs.

Phenotypic and transcriptional profiles of IFN–CSCs revealed that IFN-I-treated epithelial cancer cells (AT3 and B16.F10) acquired a typical stem-like elongated morphology (Extended Data Fig. 1c). Moreover, IFN-I promoted the emergence of the side population (SP, a bona fide CSC feature) accompanied by a significant increase in cell death (Fig. 1c and Extended Data Fig. 1d). As expected, SP was significantly reduced by cotreatment with verapamil (VRP), the blocker of ATP-binding cassette transporters. Accordingly, IFN-I exposure induced significant upregulation of *Klf4*, *Oct3/4*, *Sox2*, *Nanog*, hes family bHLH transcription factor 1 (*Hes1*) and *Nes* (Fig. 1d and Extended Data Fig. 1e), and endowed MCA205 and AT3 cancer cells with increased sphere-forming ability (Fig. 1e). Moreover, when serially replated in standard CSC culture conditions, only spheres pre-exposed to IFN-I retained a CSC-related phenotypical and transcriptional profile (Extended Data Fig. 1f).

Notably, the local treatment of MCA205-derived tumors in syngeneic immunocompetent C57Bl/6J mice with one single dose of $10^5$ U IFN-I promoted a significant accumulation of CD44H CSCs, while treatment with repeated doses of $2 \times 10^4$ U IFN-I did not enrich for CSCs (Fig. 1f). Moreover, at odds with one single $6 \times 10^3$ U ml$^{-1}$ IFN-I administration (Fig. 1a), repeated treatment with lower doses IFN-I ($3 \times 10^3$ U ml$^{-1}$ and $10^3$ U ml$^{-1}$) did not induce CSC accumulation in MCA205 and AT3 cell lines (Extended Data Fig. 1g).

**Fig. 1 | Emergence of CSCs following IFN-I treatment. a**, Multiparametric flow cytometry analysis of the illustrated CSC surface markers in MCA205 and AT3 cells treated with mock (CTR) or IFN-I ($6 \times 10^3$ U ml$^{-1}$, 72 h). Representative biparametric plots and histograms showing CD133$^+$CD24$^+$CD44$^+$ percentages (mean ± s.e.m. with individual data point, $n=3$ and $n=4$ independent experiments) are shown. For more details on gating strategies, see Supplementary Fig. 1. **b**, Flow cytometry analyses of CD44L and CD44H percentages (top) and qRT–PCR analyses of the reported TF (bottom) in FACS-isolated CD133$^-$ and CD133$^+$ MCA205 cells treated as in **a**. Mean ± s.e.m. with individual data point, $n=3$ independent experiments. qRT–PCR data are reported as mean fold change (FC) ± s.e.m. over CTR after *Ppia* intrasample normalization, $n=3$ and $n=2$ independent experiments. $^*P < 0.05$, $^{**}P < 0.01$, $^{***}P < 0.001$; for exact $P$ values, see Supplementary Table 1. **c**, SP (Hoechst 33342$^-$ within propidium iodide, PI$^-$) in MCA205 and AT3 cells left untreated (black), treated with VRP (100 μM, light green), IFN-I (blue) or VRP + IFN-I (dark green). Mean ± s.e.m. with individual data point, $n=9$ and $n=6$ independent experiments. **d**, TF expression levels in IFN-I-treated MCA205 cells. Data are reported as in **b**, $n=3$ and $n=4$ independent experiments. $^*P < 0.05$, $^{**}P < 0.01$, $^{***}P < 0.001$, see Supplementary Table 1 for exact $P$ values. **e**, Clonogenicity of MCA205 and AT3 cells plated in soft-agar upon treatment as in **a**. The number (mean ± s.e.m. and individual data point) of biologically independent samples collected over three independent experiments is shown. **f**, Ex vivo flow cytometry of CD44L and CD44H cells within the CD45 negative (CD45$^-$) fraction of MCA205 tumors from C57Bl/6J mice either treated with one single dose ($1 \times 10^5$ U) or repeated doses ($2 \times 10^4$ U) of IFN-I. Mean ± s.e.m. and individual data points for 10 mice per group from two experimental replicates. **a,b,d** Unpaired two-sided Student's *t*-test and unpaired two-sided Student's *t*-test with Welch's correction compared with CTR. **c,f**, Brown–Forsythe test with Dunnet's correction and ordinary one-way ANOVA test followed by Bonferroni's correction. **e**, Unpaired two-sided Student's *t*-test with Welch's correction and two-tailed Mann–Whitney test compared with CTR.

Collectively, these data demonstrate that depending on the dose and time of administration, IFN-I may favor the appearance of putative CSCs in multiple murine and human cancer cell lines.

**IFN-I during immunogenic chemotherapy triggers cancer stemness.** As IFN-I plays a role during ICD[11], we asked whether immunogenic chemotherapy could enrich for CSCs. We took advantage of a

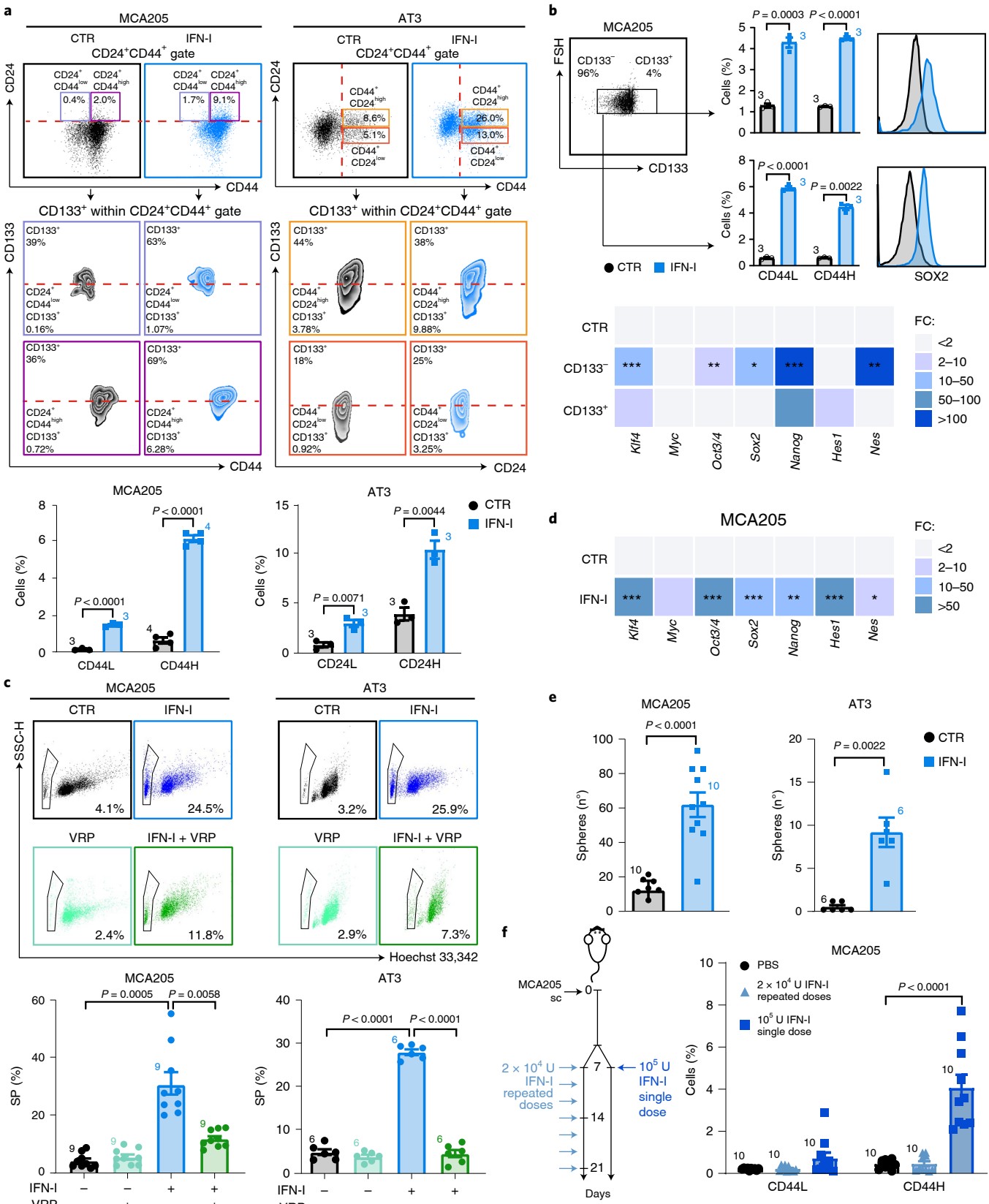

library of prevalidated MCA205-derived clones deficient for cardinal elements of the IFN-I pathway, including: (1) *Ifnar1*, (2) stimulator of interferon response cGAMP interactor 1 (*Sting1*, best known as *Sting*), (3) toll-like receptor 3 (*Tlr3*), (4) toll-like receptor adapter molecule 1 (*Ticam1*, best known as *Trif*), (5) interferon induced with helicase C domain 1 (*Ifih1*, best known as *Mda5*) and (6) mitochondrial antiviral-signaling protein (*Mavs*, also known as *Ips-1*) (Fig. 2a)[11]. We exposed these clones to the ICD inducer OXP ('donor' dying cells), then cocultured donor dying cells with untreated clones of the same genotype ('receiving' viable cells) for 24 h, and, finally, analyzed receiving cells at phenotypic and transcriptional levels (Extended Data Fig. 2a). Wild-type (*Wt*) clones responding to OXP displayed a significant increase in the two CD44H and CD44L CSC subpopulations (ICD–CSCs, Fig. 2b). On the contrary, the vast majority of clones deficient in the IFN-I pathway presented a certain degree of impairment of ICD–CSC enrichment (Fig. 2b), indicating dependence on IFN-I signaling. This effect was not paralleled by differential cell death induction, as all clones displayed similar sensitivity to OXP (Extended Data Fig. 2b). The comparison within each genotype revealed a significant ICD–CSC enrichment in OXP-treated versus untreated conditions in all but *Ifnar*[−/−] clones, suggesting a compensation between nucleic acid-sensing pathways (Fig. 2b). Accordingly, both IFN-I and OXP treatment induced the accumulation of CSC-related transcripts in *Wt* clones and, to a lesser and heterogeneous extent, in *Sting1*[−/−], *Tlr3*[−/−], *Ticam1*[−/−], *Ifih1*[−/−] and *Mavs*[−/−] clones, but failed to do so in *Ifnar*[−/−] clones (Fig. 2c). Moreover, the abrogation of the AIM2 and RIG-I signaling significantly reduced, but did not completely abrogate ICD–CSC enrichment in *Wt* and *Ifih1*[−/−] clones (Extended Data Fig. 2c and Supplementary Fig. 2). Finally, DOX-mediated ICD induction favored a complete transcriptional rewiring toward pluripotency, enhancing the expression of the entire panel of TFs analyzed, while the non-ICD drug cisplatin (CDDP), which induces very low levels of IFN-I (ref. [11]), affected the expression of only few TFs (Fig. 2d).

We then exploited DOX red fluorescence, observing two distinct cell subsets (DOX[+low] and DOX[+high]) in DOX-treated MCA205 cells differing for the capability to extrude DOX and Hoechst 33342 (Extended Data Fig. 2d). Notably, following drug withdrawal, only DOX[+low] cells survived and resisted rechallenge with distinct ICD inducers (Extended Data Fig. 2e), indicating multidrug tolerance/resistance[21]. To explore the in vivo appearance of ICD–CSCs, we evaluated the effect of DOX and CDDP on syngeneic immunocompetent mice bearing MCA205 tumor grafts, analyzing tumor growth control as well as CSC markers 15 days after (the first) treatment, that is, when starting to escape growth control[11]. We found a twofold increase of CD44H and NANOG[+] cells upon DOX, but not CDDP administration (Fig. 2e and Extended Data Fig. 2f). Also, when used as an adjunctive to DOX treatment, repeated doses of $2 \times 10^4$ U IFN-I prevented ICD–CSC accumulation, favoring tumor control and animal survival (Fig. 2f).

Altogether, these results demonstrate that IFN-I production upon ICD can promote CSC enrichment, both in vitro and in vivo, pointing to this effect as an adaptive response deployed by cancer cells to escape therapy control.

**Nucleic acid transfer transduces stem signaling between cancer cells.** To dissect the molecular mechanisms underlying ICD–CSC enrichment, we cocultured OXP-treated donor MCA205 cells with untreated receiving MCA205 cells alone or in combination with benzonase (BNZase), which degrades all nucleic acids, or RNase A, RNase H or DNase, which selectively degrade single-strand RNAs, double-strand RNAs or DNA. We observed differential effects in the two CD44H and CD44L ICD–CSC subsets, with BNZase preventing the enrichment of both CSC populations, while RNase A, RNase H and DNase significantly affecting only CD44L cells (Fig. 3a). Accordingly, BNZase halved the proportion of ICD–CSCs in receiving AT3 and CT26 cells (Extended Data Fig. 3a). The observation that only the depletion of all nucleic acids nullifies ICD–CSC enrichment, again suggests that this phenomenon depends on intact IFN-I signaling.

We next investigated the involvement of extracellular vesicles (EVs) in ICD–CSC enrichment. EVs isolated from donor MCA205 cells and stained with the nontoxic fluorescent membrane dye PKH26 were added to receiving MCA205 cells (Extended Data Fig. 3b). EV uptake in receiving cells, confirmed by fluorescence microscopy and flow cytometry (Fig. 3b), induced a considerable increase in CD44H and CD44L cells and in the expression of most TFs, which was impaired by cotreatment with the actin inhibitor cytochalasin D (cyto D) (Fig. 3c,d). Intriguingly, EVs from OXP-treated cancer cells carried messenger RNAs (mRNAs) for TFs (*Myc*, *Oct3/4*, *Sox2*, *Nanog*, *Hes1*, *Nes*), invasion molecules (Twist-related protein 1 (*Twist1*, also known as *bHLHa38*)), ICs (programmed cell death 1 ligand 2 (*Pdcd1lg2*, also known as *Pdl2*), lectin, galactose binding, soluble 9 (*Lgals9*, best known as *galectin-9*)) and *Ifnb1* (Fig. 3e), suggesting their contribution to cancer cell dedifferentiation and aggressiveness upon ICD.

Altogether, these data indicate that ICD–CSC enrichment occurs through paracrine processes involving free and EV-mediated transfer of nucleic acids and stem-related mRNAs.

**Behavioral and immunogenic features of IFN–CSCs and ICD–CSCs.** We then analyzed FACS-isolated CD44H and CD44L ICD–CSCs separately, and analyzed hallmark CSC features, including chemorefractoriness, tumorigenic/metastatic potential and capability to escape immune control. We observed that CD44H and CD44L MCA205 cells exhibit a distinct sensitivity to ICD inducers, with only CD44H cells showing higher therapeutic resistance than parental (PAR) cells, both in vitro (Extended Data Fig. 4a) and in vivo, in immunocompetent mice (Fig. 4a). In vivo studies also revealed higher tumorigenicity and less immunogenicity of CD44H

---

**Fig. 2 | CSC promotion during immunogenic chemotherapy. a**, Major intracellular pathways upstream of IFN-I and inflammation. **b**, Multiparametric flow cytometry analysis of CSC surface markers in MCA205 derived clones with the indicated genotypes left untreated (CTR) or treated with OXP (300 μM, 24 h). The histograms represent the percentage (mean ± s.e.m. and individual data points, *n* = 3 independent experiments) of CD44H and CD44L cells. **c,d**, Quantification by qRT–PCR of the expression levels of the illustrated reprogramming factors in MCA205 clones left untreated or exposed to OXP (3, 30, 300 μM, 24 h) or IFN-I ($6 \times 10^3$ U ml$^{-1}$) (**c**) and in MCA205 and AT3 cells left untreated or administered with DOX (0.25, 2.5, 25 μM), OXP (3, 30, 300 μM) or CDDP (1.5, 15, 150 μM) (**d**). Data are reported as mean FC over untreated condition after intrasample normalization to the expression levels of *Ppia*, *n* = 2, for **c**, and *n* = 3, for **d**. *$P < 0.05$, **$P < 0.01$, ***$P < 0.001$, see Supplementary Table 1 for exact *P* values. **e,f**, MCA205 tumors grown in C57Bl/6J mice treated intratumorally as illustrated. Ex vivo flow cytometric analysis of the percentage of CD44L and CD44H cells in the CD45 negative (CD45$^-$) fraction are reported in **e**, while tumor growth curves (mean tumor surface ± s.e.m.) and the percentage of tumor-free mice are shown in **f**. In **e**, data are presented as mean ± s.e.m. along with individual data points for 15 and 12 mice from two experimental replicates; the results for CSC enrichment upon one single dose of $1 \times 10^5$ U of IFN-I or repeated doses of $2 \times 10^4$ U of IFN-I of this experiment are reported in Fig. 1f. In **f**, data are presented as mean ± s.e.m. along with individual data points for 6 and 8 mice from two experimental replicates. **b**, Unpaired two-sided Student's *t*-test with Welch's correction compared with CTR cells with each clone. **d,e**, Ordinary one-way ANOVA test followed by Bonferroni's correction compared with CTR cells (**d**) and PBS-treated and DOX-treated mice (**e**). **f**, Ordinary two-way ANOVA test and log-rank (Mantel–Cox) test.

ICD–CSCs compared with CD44L ICD–CSCs. Although both sub-populations were able to generate tumors in immunocompromised NOD SCID γ (NSG) mice, only CD44H ICD–CSCs developed neoplasms at the lowest doses (Fig. 4b). Along with this, CD44H (but not CD44L) ICD–CSCs were able to overcome immunosurveillance, developing tumors at high incidence in immunocompetent

hosts when injected at the highest number (Fig. 4b). Several findings confirmed the unique low immunogenicity of CD44H cells. First, DOX-treated PAR cells were able to vaccinate 85% of mice against PAR and CD44L ICD–CSCs, but only 30% of mice challenged with CD44H ICD–CSCs (Fig. 4c). Second, while only 15% of immunocompetent mice rejecting CD44H ICD–CSCs were

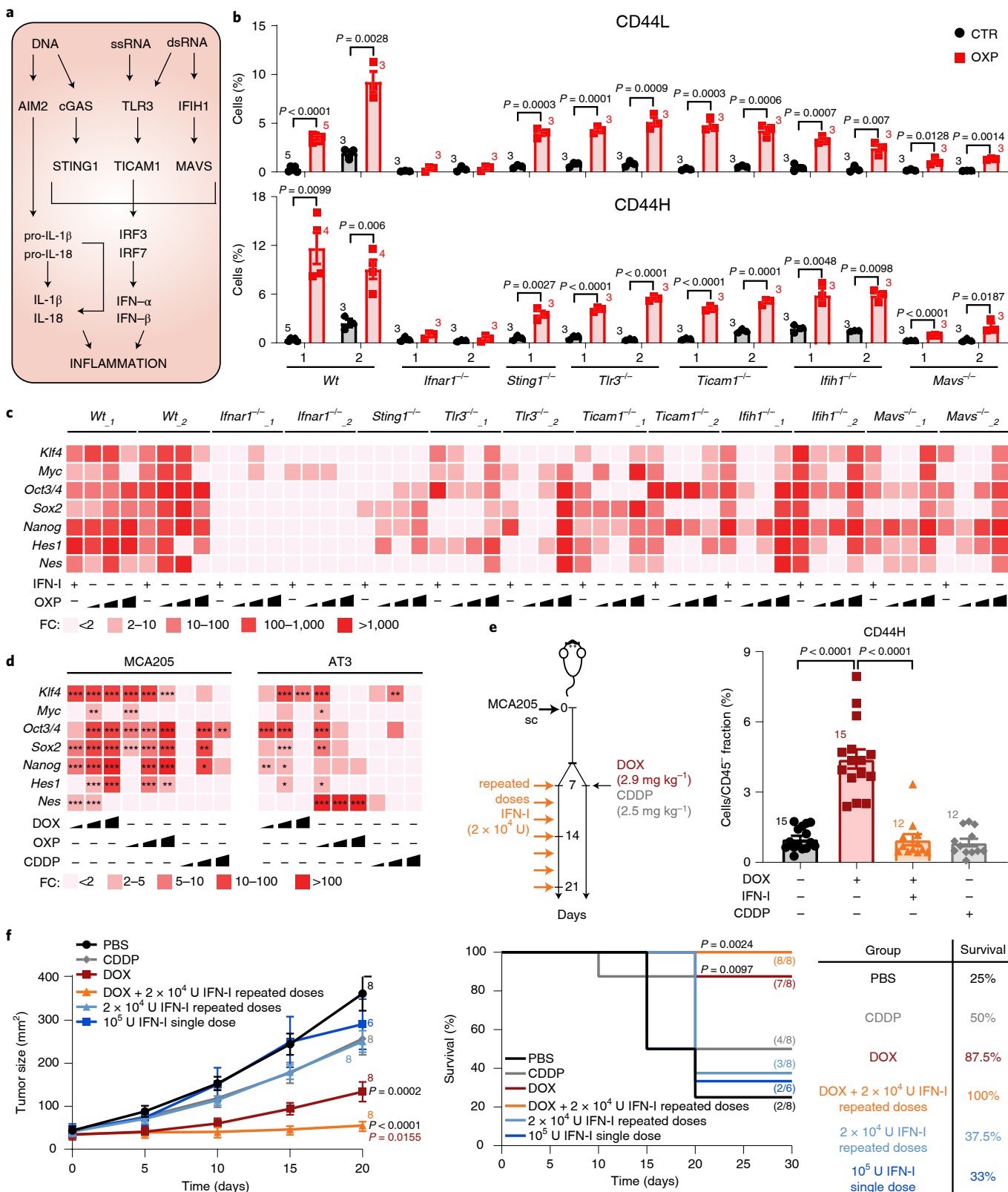

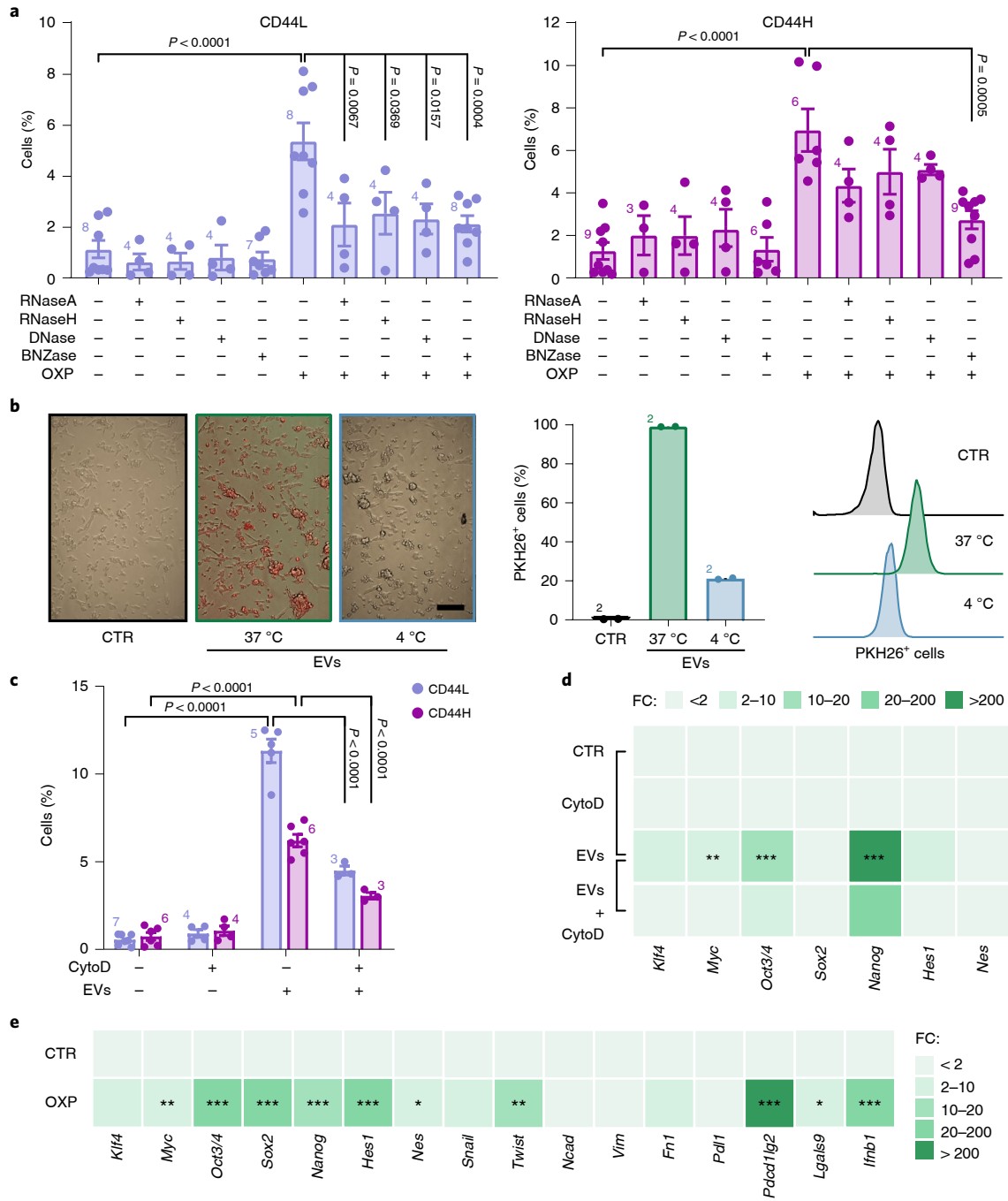

**Fig. 3 | Cell-to-cell horizontal transfer of nucleic acids and dedifferentiating factors during immunogenic chemotherapy. a**, Multiparametric flow cytometry analysis of CSC surface markers in receiving viable MCA205 cells upon coculturing with donor MCA205 cells left untreated or previously treated with OXP (300 μM, 24 h) alone or in combination with the indicated nucleases. Columns represent the percentage of CD44H and CD44L cells, expressed as mean ± s.e.m. and individual data points. Number of biologically independent experiments are reported. **b**, Fluorescence microscopy (left) or flow cytometry (right) analysis of the internalization (at 37 °C and 4 °C) of donor cell-derived, PKH26-stained EVs by receiving MCA205 cells. Scale bar, 100 μm. One representative experiment out of two is shown. **c**, Multiparametric flow cytometry analysis of CSC surface markers in receiving MCA205 cells cocultured with donor MCA205 cell-derived EVs in the presence of cyto D (0.5 μM). Data are expressed as mean ± s.e.m. and individual data points; number of biologically independent experiments is reported. **d,e**, Assessment of the expression levels of the indicated reprogramming factors by qRT–PCR in receiving MCA205 cells stimulated with donor MCA205 cell-derived EVs alone or in the presence of cyto D, as before (**d**) and inside EVs (**e**). Data are reported as mean FC ± s.e.m. over control conditions, $n = 2$ and $n = 3$, for **d**, $n = 2$, $n = 3$, $n = 4$, $n = 6$, $n = 7$, $n = 9$ and $n = 10$ for **e**, independent experiments, after intrasample normalization to *Ppia* expression levels. *$P < 0.05$, **$P < 0.01$, ***$P < 0.001$, see Supplementary Table 1 for exact $P$ values. See also Extended Data Fig. 3. **a,c,d**, Ordinary one-way ANOVA test followed by Bonferroni's correction. **e**, Unpaired two-sided Student's *t*-test.

vaccinated against viable PAR cells, CD44L ICD–CSCs and PAR cells conferred a higher long-term protection against this rechallenge (Extended Data Fig. 4b). Finally, when injected intravenously into immunocompetent mice, CD44H (but not CD44L) ICD–CSCs developed lung metastases (Fig. 4d). In this experiment, CD44L ICD–CSCs reacquired metastatic potential in immunocompetent

mice depleted of CD4 and CD8 T cells and, even more, in immunodeficient NSG mice (Fig. 4d and Extended Data Fig. 4c), thus confirming their immune control. Of note, while a considerable fraction of CD44H ICD–CSCs divided asymmetrically (a common CSC feature), the vast majority of CD44L ICD–CSCs underwent symmetric division (Fig. 4e,f). Altogether, these results indicated that CD44H but not CD44L can be considered bona fide CSCs.

We thus focused on the CD44H ICD–CSCs subset. To gain insights into their immunogenicity, we analyzed the proliferation rate of isolated CD8[+] H-2Kb/ovalbumin (OVA)-specific OT-1 T cells previously primed with dendritic cells (DCs) that had taken up apoptotic OVA-expressing CD44H (CD44H-OVA) ICD–CSCs or PAR cells, and then boosted with viable cells of the same type. In line with the immune privileged nature observed in vivo (Fig. 4a–d), CD44H-OVA ICD–CSCs induced a significantly lower expansion of OT-1 CD8 T cells than PAR counterparts (Fig. 5a) and resisted CD8-mediated killing (Fig. 5b). These data prompted us to hypothesize that CD44H ICD–CSCs could escape immune control by inducing CD8 T cell exhaustion. To pursue this hypothesis, we analyzed common IC ligands, finding an increase in the percentage of cells positive to PDL1, PDCD1LG2, CEA1 and LGALS9 in CD44H cells (Fig. 5c). Consistently, CD8[+] T tumor-infiltrating lymphocytes isolated from MCA205-bearing mice 15 days after intratumoral injection of DOX (when CSC enrichment occurs), but not of CDDP, displayed a significant increase in the fraction of cells expressing the LGALS9 receptor IC Hepatitis A virus cellular receptor 2 (HAVCR2, best known as TIM3) (Fig. 5d). We extended the characterization of ICD–CSCs to AT3 cells (that is, the CD24L cell subset), confirming the increase in the percentage of cells displaying PDL1, PDCD1LG2 and LGALS9 (Fig. 5c).

To further characterize ICD–CSC immunogenicity, we measured cytokine production through Luminex Multiplex Assay, observing a unique chemokine secretion pattern in CD44H MCA205 and CD24L AT3 ICD–CSCs compared with their respective PAR cells. This encompasses reduced levels of proinflammatory chemokines CCL2 and CCL5, which mediate inflammatory monocyte trafficking and DC-T cell interactions[22], and enhanced capability to secrete CXCL1 and CXCL2 (the latter in CD24L AT3 cells), which promote chemoresistance and metastasis[23] (Fig. 5e). Notably, CD24L AT3 cells also showed higher levels of the regulatory T cell chemoattractant CCL22 (ref. [24]) than PAR AT3 cells. Accordingly, when CD24L ICD–CSCs or PAR AT3 cells were confronted with histocompatible splenocytes in ad hoc microfluidic devices[25] and then analyzed by videomicroscopy for their in vitro capability to recruit immune cells, only PAR cells were able to attract and stably interact with splenocytes at as early as 24 h (Fig. 5f,g and Supplementary Videos 1–4). At odds, CD24L ICD–CSCs failed to do so and, instead, migrated towards splenocytes starting a transient and unproductive

interaction only upon 48 h. Finally, when we confronted PAR and CD24L AT3 cells in a microfluidic 'competition' device[26] (Extended Data Fig. 4d), immune cells selectively migrated towards PAR cells, moving away from CSCs (Fig. 5h,i).

Altogether, these results indicate the existence of a mechanism of adaptation of cancer cells to immunogenic chemotherapy that actively contributes to intratumor heterogeneity, as the collection of induced CSC subpopulations has differential therapeutic response, aggressiveness and immunogenicity.

**Global chromatin remodeling downstream of IFN-I.** To dissect the mechanisms underlying cancer cell reprogramming downstream of IFN-I, we mapped the chromatin landscape of PAR (P) and CD44H (H) MCA205 cells by the assay for transposase-accessible chromatin with high-throughput sequencing (ATAC–seq) (Fig. 6a–c). By analyzing ATAC–seq peaks, we conceived a closed-to-open (C → O) and an open-to-closed (O → C) logic, and stratified genes in four groups. The C[P]C[H] and O[P]O[H] groups comprise genes with peaks permanently closed (that is, putatively repressed) or open (that is, putatively expressed) in both samples, while the C[P]O[H] and O[P]C[H] groups comprise genes whose peaks are closed in PAR cells and open in CD44H IFN–CSCs and vice versa. In particular, we focused on the C[P]O[H] group containing genes putatively more expressed in CSCs. As expected, we found genes dictating the CSC phenotype and behavior, including, but not limited to, cancer stemness (*Myc* and *Sox*) and epithelial-to-mesenchymal transition (EMT) (*Gata6* and *Tfcp2*). We also found genes involved in immune evasion, including the negative regulator of the antigen presentation machinery *Gpr17* and the inhibitor of granzyme activity *Serpin* (Fig. 6a). Consistently, the O[P]C[H] group contains tumor suppressor genes (*Cdh*, *Cdk2ap1*, *Dlg2*, *Ripk3* and *Fbxw2*) and genes involved in antigen presentation machinery (*Tap1*, *Tap2* and *Ctsl*) and inflammation (*Il24*, *Il27*, *Gsdmd* and *Uba7*) (Fig. 6a). Integration with RNA-sequencing (RNA-seq) analyses confirmed an increased expression of genes involved in tumorigenesis, tumor progression, invasiveness (*Csf1r*, *Trpm4*, *Itga5*, *Wee1*, *Baiap2*, *Ttll7* and *Spire1*) and immune escape (*Gpr17*), coupled with repression of genes involved in tumor suppression and immune recognition (*Cdh1*, *Il12b*, *Tlr5*, *Cdk2ap1*, *Il34*, *Il16* and *Ctsl*) in CD44H IFN–CSCs (Extended Data Fig. 5a).

Next, we performed TF-binding motif enrichment with the HOMER motif software, revealing considerable differences between CSCs and PAR cells for accessible motifs, indicating extensive global chromatin remodeling in CSCs (Fig. 6c and Supplementary Fig. 3a). In particular we found enrichment of motifs for various TFs of the helix–turn–helix superfamily (that is, RFX, Rfx1, Rfx2, Rfx5 and X-box), the Homeobox basic helix–loop–helix (bHLH) member Pitx1:Ebox, the Rel homology domain family member NFkB-p65 and the zinc-finger family member ZBTB

**Fig. 4 | Functional characterization of CSCs induced during immunogenic chemotherapy. a,** Tumor growth of PAR and CD44H MCA205 cells in C57Bl/6J mice either PBS- or DOX (2.9 mg kg[−1])-treated. Growth curves show the mean tumor surface ± s.e.m. in one representative experiment out of two. Number of biologically independent mice and *P* values for DOX-treated CD44H versus DOX-treated PAR cells (purple) and DOX versus PBS treatments in PAR cells (black) are shown. See Supplementary Table 1 for exact *P* values, and Extended Data Fig. 4a. **b,** In vivo evaluation of the tumorigenicity of PAR, CD44H and CD44L MCA205 cells in C57Bl/6J (*Wt*) or NSG mice at the indicated dose. The percentage of tumor-free mice out of 12 and 15 mice per group from two experimental replicates is shown. Tumor-free mice from this experiment were rechallenged as reported in Extended Data Fig. 4b. **c,** In vivo evaluation of the vaccination potential of MCA205 cells. CTR or PAR MCA205 cells treated with 25 μM DOX (vaccination/VAX condition) were inoculated in the flank of C57Bl/6J mice. Seven days later animals were challenged with 1×10[5] PAR, CD44H or CD44L MCA205 in the other flank. The percentage of tumor-free mice out of six biologically independent mice per group in CTR and VAX conditions is shown. **d,** In vivo evaluation of the metastatic potential of parental or ICD–CSC MCA205 injected in the tail vein of C57Bl/6J mice, NSG mice or C57Bl/6J depleted of CD4 and CD8 cells. Representative macroscopic observation and quantification (mean ± s.e.m. and individual data points, *n* = 6 biologically independent mice per group) of the number of lung metastases 15 days post injection are reported. See also Extended Data Fig. 4c. **e,f,** Immunofluorescence analysis of cell divisions in FACS-isolated CD44H and CD44L MCA205 cells upon NUMB staining (**e**) and videomicroscopy analysis of cell divisions in FACS-isolated CD44H upon PKH26 staining (**f**, scale bar, 20 μm). In **e**, the percentage of asymmetric divisions upon image analysis quantification of the fluorescent signal in the two daughter cells is reported (*n* = 100, pool of three independent experiments, scale bar, 5 μm). **a,** Ordinary two-way repeated measures (RM) ANOVA test followed by Bonferroni's correction. **b,c,** log-rank (Mantel–Cox) test. **d,** Ordinary one-way ANOVA test followed by Bonferroni's correction.

in CD44H cells. Conversely, the zinc-finger motifs CTCF, BORIS and NRSF, the transcriptional enhanced associate domain (TEA, TEAD) motifs (that is, TEAD and TEAD1-4), the Rel homology domain-basic leucine-zipper superfamily member NFAT-AP1, the ETS, RUNT, the interferon-sensitive response element and the CCAAT box-binding transcription factor motifs were more accessible in PAR cells. We finally reconstructed protein–protein interaction subnetworks and biological processes specifically modulated in CD44H IFN–CSCs using the clusterProfiler and enrichPlot R packages (Fig. 6d and Supplementary Fig. 3b). Gene ontology (GO)

analysis showed that most of the upregulated genes in CD44H cells (red module) have significant functional connections with stemness maintenance, tissue remodeling, immune suppression, response to stress and enhanced chromatin accessibility.

Altogether, these results provide clues about a global chromatin remodeling and a modular reorganization of specific pathways downstream of IFN-I.

**Epigenetic regulation of cancer stemness by KDM1B.** Among the genes specific for the CSC fraction (CD44H cells), we identified

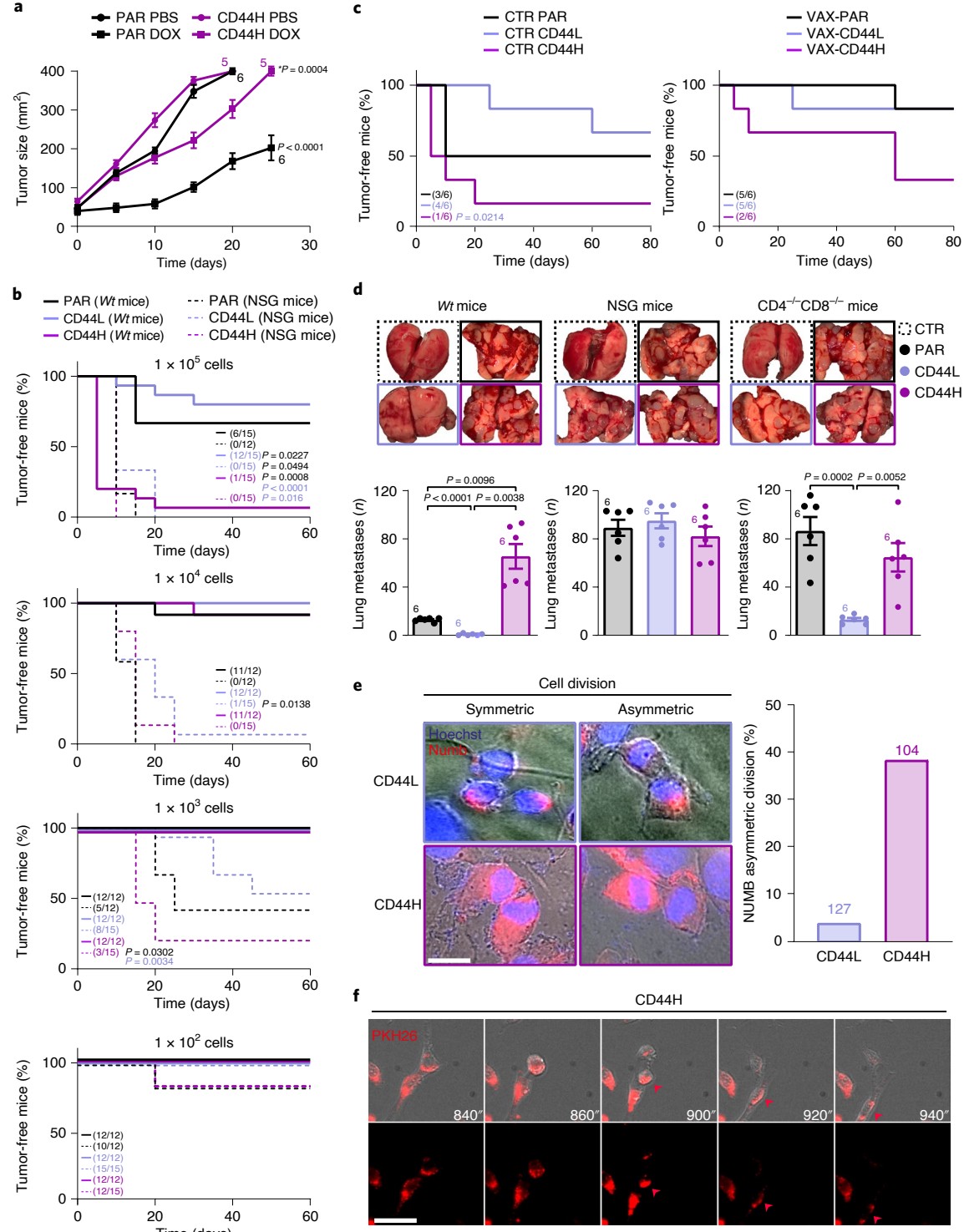

multiple ISGs, including (but not limited to) *Ifi27l2a*, *Ifi27l2b* and the epigenetic regulator *Kdm1b* (Fig. 6a,b). We were particularly intrigued by *Kdm1b* given the crucial role of chromatin remodeling in cancer evolution, cellular plasticity and immune escape[5,27–29].

We first performed ATAC–seq studies on MCA205 cells engineered to either silence (KD) or overexpress (OVER) *Kdm1b* (Extended Data Fig. 5b). We revealed the presence, in *Kdm1b*[OVER] cells, of open peaks for genes involved in cancer stemness (*Klf4*, *Myc*, *Pou5f1*, *Sox2*, *Nanog* and *Hes1*), embryonic development (*Tbx4*), EMT (*Gata6* and *Tfcp2*), cancer cell invasiveness and metastatization (*Spire1* and *Trpm4*), tumorigenesis, tumor progression and therapy resistance (*Csf1r*, *Itga*, *Baiap* and *Slc6a6*) and immune escape (*Gpr17*) (Extended Data Fig. 5c). Of great interest, all these peaks were closed or significantly less open in *Kdm1b*[KD] cells, thus supporting their epigenetic regulation by KDM1B. To further address the epigenetic role of KDM1B, we performed chromatin immunoprecipitation sequencing (ChIP–seq) analysis on CD44H ICD–CSCs. We found that KDM1B interacts with genes involved in stemness maintenance, embryonic development, EMT, invasiveness, wound healing and sprouting angiogenesis, cell-to-cell and cell-to-extracellular matrix adhesion, response to stress (DNA damage, hypoxia, starvation), epigenetics, regulation of gene expression (at transcriptional, translational and post-translational levels), senescence and apoptosis, metabolism, cell cycle and viral signature (Extended Data Fig. 5d and Supplementary Fig. 4).

To explore in-depth the role of the ISG KDM1B in the induction of ICD–CSCs, we added the KDM1B inhibitor tranylcypromine (TCP) to the donor–receiving in vitro coculture, finding a significant reduction of CD44H percentages in receiving cells (Fig. 6e). Accordingly, when coadministered with DOX, TCP prevented the enrichment of ICD–CSCs and the increase in the percentage of TIM3+ CD8+ tumor-infiltrating lymphocytes in MCA205 tumor-bearing mice (Fig. 6f), resulting in improved tumor growth control and mice survival compared with DOX alone (Fig. 6g). Moreover, through ex vivo (MCA205) and in vitro (MCA205, CT26 and B16.F10) analyses, *Kdm1b*[OVER] cells displayed higher basal levels of CD44H and higher expression of most reprogramming factors than *Kdm1b*[KD] cells (Fig. 6h and Extended Data Fig. 6a,b). *Kdm1b* overexpression also boosted the in vitro migration in CT26 and B16.F10 cells (Extended Data Fig. 6c), and the in vivo metastatic potential in MCA205 cells, which, of note, was annulled in condition of *Kdm1b* depletion (Fig. 6i). Furthermore, *Kdm1b* overexpression decreased in vitro and in vivo sensitivity to DOX (Fig. 6j and Extended Data Fig. 6d). Finally, *Kdm1b*[OVER] MCA205 cells displayed exquisite tumorigenicity, developing tumors much more frequently than *Kdm1b*[KD] MCA205 cells when transplanted in immunocompetent mice (Fig. 6k). Accordingly, in vitro extreme limiting dilution analysis (ELDA) revealed that *Kdm1b* overexpression confers high sphere-forming potential in the different cancer cell lines analyzed (Extended Data Fig. 6e). Corroborating the impact of the immune system, no differences in the growth of *Kdm1b*[OVER] and *Kdm1b*[KD] MCA205 cells were observed in NSG mice (Fig. 6k).

Overall, these data demonstrate that KDM1B operates downstream of IFN-I, editing the epigenome of cancer cells toward stemness, immune escape and therapy resistance.

**KDM1B correlates with stemness in BC patients.** To investigate the clinical relevance of the IFN-I → KDM1B axis, we first calculated the correlation between *KDM1B*, IFN-I-related metagenes, stem-related reprogramming factors, IFN-I signatures and stemness signatures using publicly available transcriptomic data on BC patients responsive to anthracyclines[11,30]. We observed that the expression levels of *KDM1B* positively correlated with a signature composed of Yamanaka factors and two previously described stemness signatures[31,32] in at least two analyzed datasets (Fig. 7a). Moreover, we observed a positive correlation, in most analyzed databases, between stemness signatures (and in particular that reported in ref. [32]) and IFN-I signatures[11,33,34], including a signature characterized in our previous work that we dubbed 'viral mimicry'[11] (Fig. 7a and Extended Data Fig. 7a). Next, we used the BC cohort METABRIC (which includes 1,903 patients) and performed a multivariate survival analysis by stratifying patients into two groups, according to risk behavior. Of note, high-risk group patients exhibiting a significantly reduced disease-specific survival presented high expression of *KDM1B* and IFN-I or stemness signatures (Fig. 7b). Similar results were obtained for distant recurrence-free incidence (Extended Data Fig. 7b), indicating that KDM1B combined with IFN-I signature or with stemness signature positively associated with dismal prognosis.

To further correlate IFN-I and CSC signatures, we performed longitudinal immunohistochemistry (IHC) analyses on consecutive formalin-fixed paraffin-embedded BC biopsies, assessing the levels of KDM1B, IFN-I-related factors (MX1 and CXCL10) and CSC markers (CD44–CD24 and CD133) on CD45[neg] cancer cells at pre- (T0; at diagnosis) and post- (T1; at surgery) neoadjuvant anthracycline-based chemotherapy (Fig. 7c and Supplementary Table 4). We found increased CSC Allred scores (either CD44[pos]CD24[low/neg] or CD133[pos]) in 15% of cases, which positively correlated with an increased KDM1B Allred score (Fig. 7c). Confirming the mutual correlation, KDM1B levels decreased in four out of six cases in which CSC marker levels were reduced at T1. When checking for other clinically relevant parameters, we observed that three patients with increased CSC and KDM1B levels at T1 were negative for the Erb-B2 Receptor Tyrosine Kinase 2 (ERBB2, best known as HER2), of which two were triple-negative and one luminal A (Fig. 7c). Intriguingly, although no differential impact was observed in classical BC subtypes, *KDM1B* combined

**Fig. 5 | Phenotypic and functional profiling of IFN–CSC immunogenicity. a**, Flow cytometry analysis of proliferation rate of CFSE-stained CD8+ OT-1 T cells stimulated with PAR or CD44H OVA-expressing cells. The histograms represent the FC (mean ± s.e.m. and individual data points, *n* = 3 independent experiments) of nonproliferating CFSE[+high]CD8+ cells. **b**, Flow cytometry analysis of CD45− OVA-expressing PAR and CD44H cell resistance to CD8+ OT-1-mediated killing. The histograms represent the FC (mean ± s.e.m. and individual data points, *n* = 3 independent experiments) of dying PI+CD45− cells. **c**, Multiparametric flow cytometry analysis of the indicated IC molecules in MCA205 or AT3 cells. Data are presented as mean ± s.e.m. and individual data points, with number of biologically independent samples collected over three independent experiments reported. **d**, Flow cytometry analysis of TIM3 in CD8+ tumor-infiltrating lymphocytes from MCA205-derived tumor grafts 15 days post in vivo treatment with PBS, DOX (2.9 mg kg−1), or CDDP (2.5 mg kg−1). Data are presented as mean ± s.e.m. and independent data points for 15 mice per group from three experimental replicates. **e**, Quantification of released chemokines in supernatants from MCA205 and AT3 cells by Luminex Multiplex Assay. One representative experiment out of two is shown. **f–i**, Time-lapse analysis of H-2Kb splenocyte migration towards PAR and CD24L AT3 cells in microfluidic devices. Plots in (**f**) represent individual splenocyte trajectories towards target cancer cells (black spots) upon time-lapse recording. Quantification of interaction times between individual splenocytes and PAR or CD24L ICD–CSCs are shown in **g**, see also Supplementary Videos 1–4. Pictures of splenocytes in competition microfluidic devices (scale bar, 100 μm) and quantification of splenocytes migrated towards PAR or CD24L ICD–CSCs are shown in **h** and **i**. Data are expressed as mean ± s.e.m. and individual data points; number of biologically independent samples collected over three (**f,g**) and two (**h,i**) independent experiments is reported. See also Extended Data Fig. 4. **a–c**, Unpaired two-sided Student's *t*-test and unpaired two-sided Student's *t*-test followed by Welch's correction. **d,f,g**, Two-tailed Mann–Whitney test compared with PBS (**d**) and CTR (**f,g**). **i**, Ordinary two-way RM ANOVA test followed by Bonferroni's correction.

with IFN-I signature or with stemness signature positively associated with dismal prognosis in HER2 negative (HER2neg) but not in HER2 positive (HER2pos) tumors (Extended Data Fig. 7c,d).

Altogether, these results suggest a clinically relevant correlation between KDM1B levels and CSC markers during anthracycline-based immunogenic chemotherapy.

## Discussion

IFN-I may either restrain or promote tumor growth depending on the duration and intensity of the transduced signaling, two features that jointly delineate the patterns of ISG expression, so-called

IFN signature[14], and shape the accessibility to chromatin, so-called IFN-mediated epigenomic signature[35,36]. The leverage of transcriptional and epigenetic changes defines cell responses to environmental hints, dictating the efficacy of natural and therapy-induced immunosurveillance[4,10,37,38]. Here, we provide preclinical and clinical evidence that, depending on the dosage and timing of administration, IFN-I can favor the appearance of CSCs. This occurs via positive selection of pre-existing CSCs and KDM1B-dependent de novo reprogramming of cancer cells toward stemness. Therefore, beyond stimulating antitumor immunity, IFN-I can foster malignant progression leaving a detrimental 'imprint' on cancer cells.

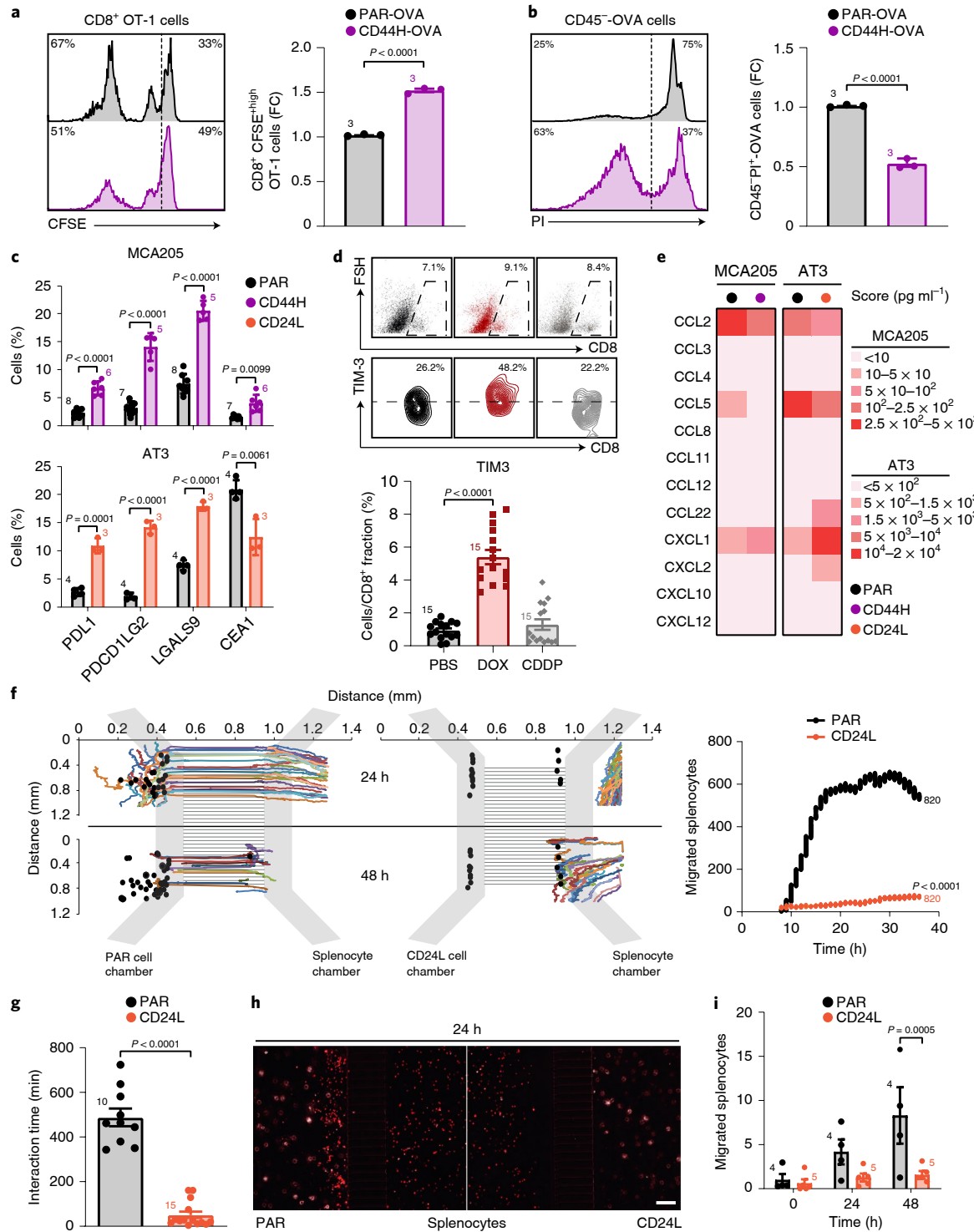

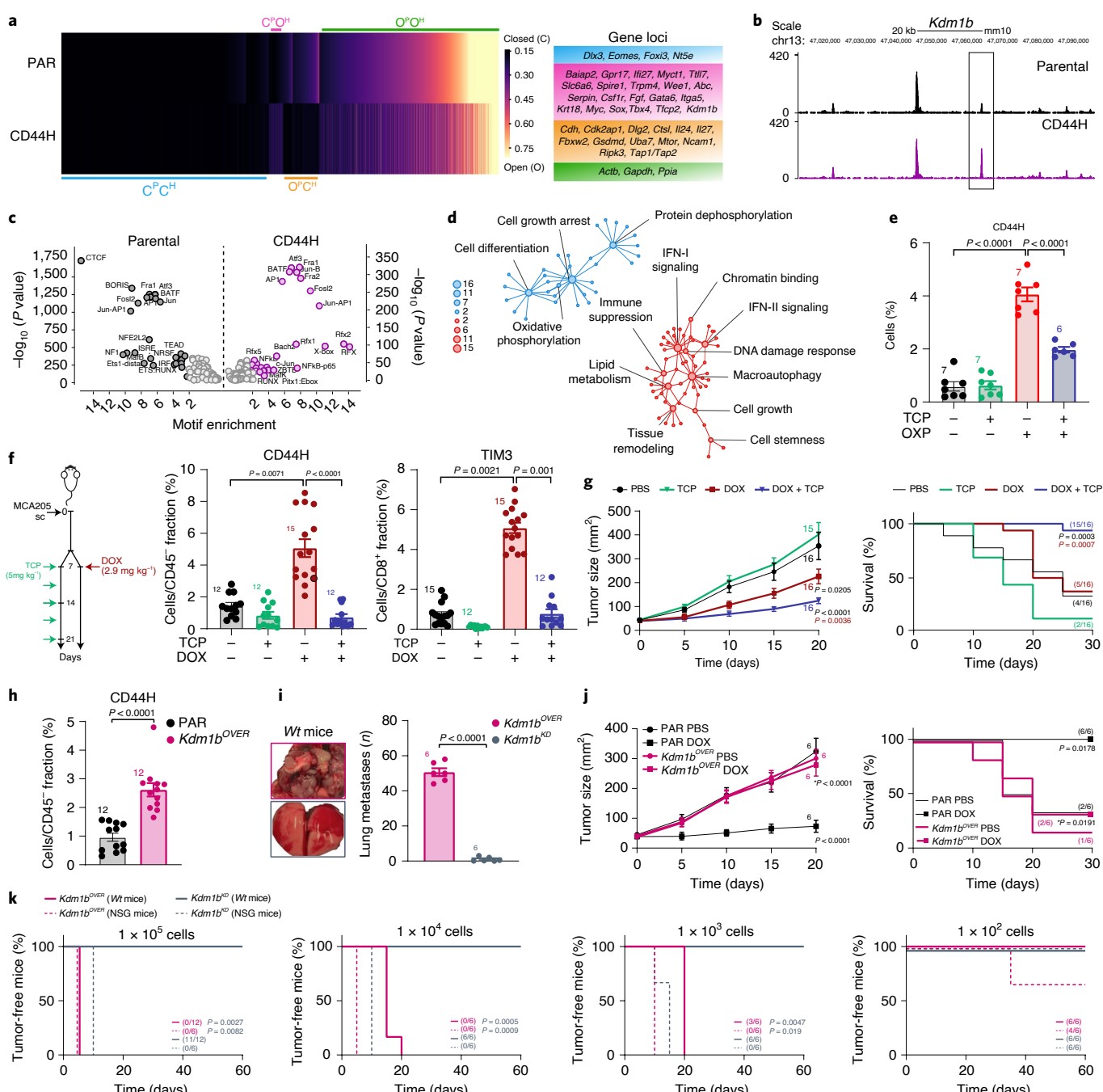

**Fig. 6 | IFN-I-driven chromatin remodeling. a–d**, ATAC–seq (**a**–**c**) and RNA-seq (**d**) analysis in PAR or CD44H MCA205 cells. Heatmap illustrating global open (O) or closed (C) genes and representative gene subgroups in PAR/P and CD44H/H are shown in **a**, representative *Kdm1b loci* within C$^P$O$^H$ group in **b**, TF binding motifs enriched more than twofold in PAR (black) or CD44H (purple) cells (x-axis, TF motif enrichment log FC in target/nontarget cells; y-axis, significance enrichment level) in **c**, and GO network analysis of upregulated (red) and downregulated (blue) genes in CD44H cells (nodes, enriched GO terms, node size, false discovery rate-adjusted enrichment P value (q value)) in **d**. **e**, Multiparametric flow cytometry analysis showing CD44H cell percentages upon OXP or OXP + TCP. Mean ± s.e.m. and individual data points. Number of biologically independent samples collected over two independent experiments is reported. **f**, Schematic experimental protocol of in vivo KDM1B inhibition and multiparametric flow cytometry analysis of CD44H and CD8+TIM3+ percentages in tumors from mice upon DOX or DOX + TCP treatment. Mean ± s.e.m. and individual data points for 12 and 15 mice per group from three experimental replicates. **g**, In vivo MCA205 tumor growth control in mice treated as illustrated. Tumor growth curves (mean tumor surface ± s.e.m. for 15 and 16 mice per group from three experimental replicates) and tumor-free mice percentages are reported. **h**, Ex vivo multiparametric flow cytometry analysis of CD44H percentages in PAR and *Kdm1b*-overexpressing (*Kdm1b*$^{OVER}$) MCA205-derived tumors. Mean ± s.e.m. and individual data points for 12 mice per group from two experimental replicates. **i–k**, In vivo evaluation of *Kdm1b*$^{OVER}$ and *Kdm1b*-depleted (*Kdm1b*$^{KD}$) MCA205 metastatic potential (**i**), DOX-based therapeutic response (**j**) and tumorigenicity (**k**) in C57Bl/6J (**i–k**) and NSG (**k**) mice. Mean ± s.e.m. and individual data points for 6 mice per group from two experimental replicates (**i**, **j**), and for 12 and 6 mice per group from two experimental replicates (**k**). See also Extended Data Figs. 5 and 6. **c**, One-sided binomial test. **e**, Ordinary one-way ANOVA test with Bonferroni's correction. **f**, Kruskal–Wallis test with Dunn's multiple comparisons. **g,k,j**, Ordinary two-way RM ANOVA test with Bonferroni's correction (**g**) and log-rank (Mantel–Cox) test (**g,k**). **h**, Two-tailed Mann–Whitney test compared with PAR. **i**, Unpaired two-sided Student's t-test with Welch's correction.

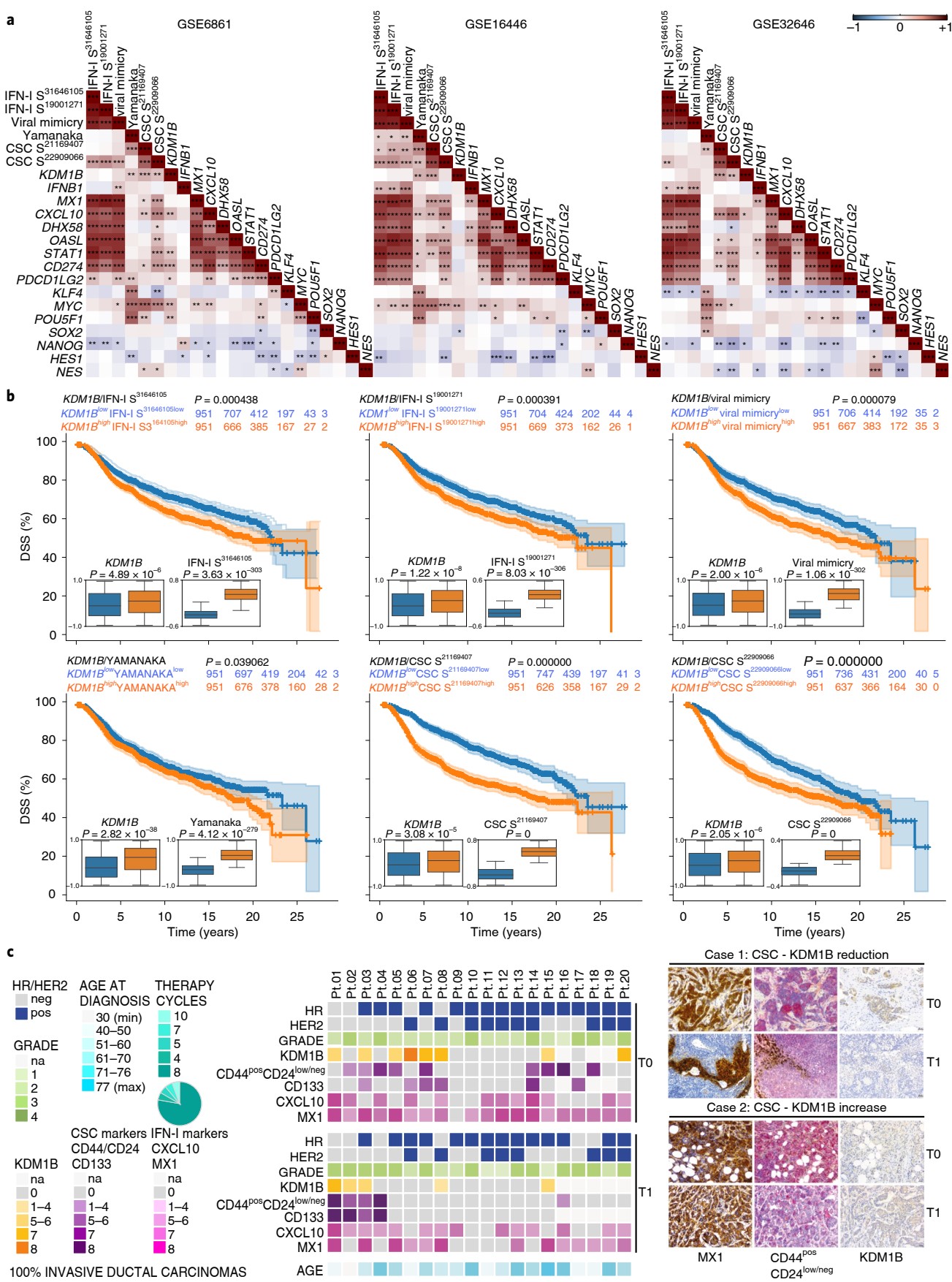

**Fig. 7 | Correlation between *KDM1B*, stemness signature and IFN-I signatures in BC patients. a**, Spearman correlations between expression score of *KDM1B* and the reported IFN-I related metagenes, stem-related reprogramming factors, IFN-I signatures and stemness signatures from microarray data of three publicly available cohorts of BC patients treated with neoadjuvant anthracycline-based chemotherapy. *$P < 0.05$, **$P < 0.01$, ***$P < 0.001$; see Supplementary Table 2 for exact correlation and $P$ values and Extended Data Fig. 7a for other datasets. **b**, Kaplan–Meier plots depicting the disease-specific survival (DSS) in BC patients from the METABRIC cohort stratified according to risk behavior and boxplots reporting the expression levels of *KDM1B* and the illustrated stemness or IFN-I signatures across the two groups. $P$ value was calculated using $P$ Cox, log-rank (Mantel–Cox). $P$ values <0.05 were considered statistically significant. The relative expression of the indicated genes and signatures is reported as mean ± s.e.m. from 1,903 patients. For statistics of boxplots see Supplementary Table 3. The correspondent distant relapse-free incidence is reported in Extended Data Fig. 7b. **c**, IHC analysis of 20 paraffin-embedded paired BC biopsies at T0 (diagnosis) and T1 (surgery) using antibodies to KDM1B, MX1, CXCL10, CD44 + CD24 and CD133. Representative IHC images from sections of two representative patients with reduced (one patient out of four) and increased (one patient out of three) KDM1B/CSC marker score are reported on the right (scale bar, 30 μm). A heatmap reporting relevant information regarding tumor grade, the mutational status of the illustrated genes and the Allred score for all analyzed markers is reported on the left. See also Extended Data Fig. 7 and Supplementary Table 4. na, not available. **a**, Two-sided Spearman's rho.

Our study sheds light on the debated and poorly investigated contribution of IFN-I signaling on tumor heterogeneity and CSC induction. On the one hand, we and others previously reported a host-protecting role of IFN-I in HER2/neu transgenic mice and triple-negative BC, because the abrogation of steady-state endogenous IFN-I signaling leads to the emergence of breast CSCs[39,40]. On the other hand, exogenous administration of IFN-I favored cancer stemness in mouse models of pancreatic cancer[41] and human BC and squamous carcinoma cell lines[42]. Nonetheless, in these studies the molecular mechanisms underlying IFN-I-CSC expansion have not been analyzed, and this phenomenon has been neither investigated in the context of ICD, nor associated with potential cancer cell reprogramming. In this respect, it appears of interest that the induction of the ISG *IFI27* in ovarian carcinoma biopsies and cell lines drives EMT, cancer stemness, invasiveness and therapeutic resistance[43]. Whether IFI27 is involved in ICD–CSC expansion requires further investigations. Irrespective of this unknown, on the basis of our results, we surmise that, depending on its duration and intensity, IFN-I signaling can either limit CSC proliferation and survival, restraining tumor growth, or favor the survival of pre-existing CSCs and cancer cell dedifferentiation, potentially leading to therapy resistance/failure. The use of a reporting system measuring IFNAR signaling in the TME upon immunogenic therapies will provide formal confirmation of this hypothesis.

Here, we also found a certain degree of phenotypic and functional heterogeneity within IFN–CSCs, consistently with the current view of an adaptable, evolutive and dynamic nature of CSCs[44,45]. In particular, we observed specific IFN-I–CSC subsets characterized by resistance to (immuno)chemotherapy, elevated tumorigenic and metastatic potential and low immunogenicity, in line with previous observations[46,47]. In our setting, CSC immune privilege encompasses a reduced capability to attract and stably interact with effector immune cells, in part due to decreased secretion of proinflammatory chemokines and enhanced capability to suppress T cell activation, and in part due to upregulated expression of IC ligands and cognate receptors. Of note, IFN-I-related immune escape has been previously associated with the upregulation in cancer (stem) cells of (1) PD-L1 and LGALS9 (ref. [16]), (2) nitric oxide synthase 2, which favors the recruitment of regulatory cells[48] and (3) SERPINB9, which inhibits granzyme B activity and thus CD8+ T cell cytotoxicity[49]. Intriguingly, through ATAC–seq and RNA-seq analyses, we found, in CD44H IFN–CSCs, upregulation of *Serpins* and downregulation of *Uba7*, a tumor suppressor ISG which codes for a protein able to attract effector T cells[50]. Whether these factors play a major role in protecting CSCs from immune attack remains to be established.

The ability of IFN-I to induce cancer stemness relies on an autocrine/paracrine cancer cell circuitry centered on the IFN-I → IFNAR → KDM1B signaling pathway. We propose a model whereby CSC induction lies on the horizontal transfer of nucleic acids and possibly stem-related encoding mRNAs from cancer cells undergoing ICD to viable cancer cells. In this regard, the cytotoxic effect of IFN-I on cancer cells can also have a contributive role by fueling this circuitry. Notably, such intercellular communication can also occur via EVs, according to the role recently ascribed to EVs in conferring resistance and metastatic recurrence to anthracyclines[51]. Intriguingly, we showed that DNA from dying/dead cells triggers the STING pathway once internalized by bystanding cells. We surmise that such exogenous, yet self, DNA is internalized though EVs and then released in the cytosol of acceptors cells where it activates the cyclic GMP-AMP synthase (cGAS). Although the precise mechanisms underlying this transfer remain to be determined, we speculate that once transferred from dying to viable cells, nucleic acids act as DAMPs leading to IFN-I production, which ultimately drives KDM1B-mediated cancer cell reprogramming, and, thus, therapy failure and tumor regrowth.

Although we are aware of the limitations of our study, and in particular the need for further confirmation in human models, we hypothesize that the activation of the IFN-I signaling directly stimulates CSCs in tumors undergoing ICD. We thus surmise the existence of a mechanism similar to that underlying virus-induced cell transdifferentiation that leads to the upregulation of core pluripotency genes[17]. Supporting our hypothesis, IFN-I was recently ascribed to have a role in chromatin remodeling and gene expression reprogramming[35,36,52]. Moreover, the expression of diverse KDMs has been correlated with 'cold' TMEs in different tumor models, as also the use of epidrugs with the reinstatement of inflammation[53–55]. Recently, epidrug-related immune modulation was shown to co-occur with MYC suppression[56]. Of relevance, here, by combining the analysis on publicly BC databases and our retrospective studies on BC patients that had received anthracycline-based therapy, we found a mutual correlation between KDM1B and stemness. In particular, in our cohort, we reported clinical evidence of combined enrichment of CSCs and KDM1B upregulation upon immunogenic treatments, especially in a HER2-negative context. Further validation on a larger cohort of patients with patient follow-up will be launched.

In conclusion, we demonstrated that IFN-I can elicit a protective but ephemeral anticancer response. By triggering KDM1B, IFN-I promotes the appearance of CSCs with traits of immune privilege and therapy resistance. This evidence provides the basis for the use of epidrugs as adjunctives to anticancer immunogenic therapies, including conventional chemotherapies and current and upcoming immunotherapies, as therapeutic means to prevent CSC expansion and patrol tumor recurrence.

## Online content

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

## Methods

**Reagents.** Further information and requests for resources and reagents are provided in Supplementary Table 5 and should be directed to and will be fulfilled by the Lead Contact, Antonella Sistigu.

**Cell lines and culture conditions.** Unless otherwise indicated, plasticware was from Falcon, Corning B.V. Life Sciences. Cells were cultured in the reported growth media under standard culture conditions (37 °C, 5% $CO_2$). Murine MCA205 *Wt*, *Kdm1b*[KD], *Kdm1b*[OVER], MCA205-OVA fibrosarcoma cells, MCA205-derived clones (*Wt*, *Ifnar*[−/−], *Tlr3*[−/−], *Ticam*[−/−], *Ifih*[−/−], *Mavs*[−/−], *Sting1*[−/−], produced as in ref. [11]), AT3 mammary carcinoma, CT26 *Wt*, *Kdm1b*[KD], *Kdm1b*[OVER] colon carcinoma cells: Roswell Park Memorial Institute (RPMI) 1640 plus fetal bovine serum (FBS, 10% v/v), 2 mM L-glutamine (L-glut), 100 IU ml⁻¹ penicillin G sodium salt (pen), 100 µg ml⁻¹ streptomycin sulfate (strept) (R10). Murine B16.F10 *Wt*, *Kdm1b*[KD], *Kdm1b*[OVER] melanoma cells, human MCF7 breast carcinoma, U2OS osteosarcoma cells: Dulbecco's Modified Eagle Medium (DMEM) supplemented as above (D10). Puromycin (1:500) was added to *Kdm1b*[KD] cell medium. Human MCF10A normal breast cells: DMEM/F12 plus 5% horse serum, L-glut, pen, strept (as above), 10 µg ml⁻¹ insulin, 5 µg ml⁻¹ hydrocortisone, 100 ng ml⁻¹ cholera-toxin, 10 ng ml⁻¹ human epithelial growth factor. HMLER cells: 1:1 DMEM/F12 plus pen, strept, insulin (as above), 10 µg ml⁻¹ hydrocortisone and HuMEC Basal Serum-Free Medium plus HuMEC Supplement.

CSC growth potential was tested in culture conditions as described elsewhere[57].

**Cytofluorometric analysis, cell sorting.** To assess CSC surface markers, $1 \times 10^5$ tumor cells were cultured in 2 ml growth medium (6-well plates) and treated with purified mouse IFN-I (1–3×10³ U ml⁻¹, 10 days or 6×10³ U ml⁻¹, 3 days), recombinant human Roferon-A (6×10³ U ml⁻¹, 3 days), DOX (25 µM), OXP (300 µM) ± TCP (10 µM) thalidomide (10 µg ml⁻¹), amlexanox (5 µM), BX795 (100 nM) and MRT67307 (500 nM), 48 h. Cells were then washed in Dulbecco's phosphate buffered saline (D-PBS) and stained with fluorescently labeled monoclonal antibodies (mAbs) anti-CD44, anti-CD133, anti-CD24, anti-CD44v6 (1:20) in cold D-PBS-1% FBS solution, 30 min on ice. Appropriate Alexa Fluor (AF)−488 secondary Ab (1:500) was added to CD44v6 stained cells. In all experiments 4′,6′-diamidino-2-phenylindole (DAPI)/Sytox blue/ Viability 405-452 were used to distinguish live from dead cells; only live cells were analyzed. All acquisitions were performed with FACSCanto-II (BD Biosciences), MACSQuant-VYB Analyser-10 (Miltenyi Biotec), CytoFLEX (Beckman Coulter) cytofluorometers and data analyzed with the FlowJo software v10.0.7. Using the same costaining protocol, specific CSC-like subsets were isolated by fluorescence-activated cell sorting (FACS, FACSAria, BD Biosciences) and further characterized by in vitro/in vivo assays. For gating strategies, see Supplementary Fig. 1.

DOX[+low] and DOX[+high] MCA205 cells were sorted post DOX treatment (2.5 µM, 48 h). For assessment of IC expression, FACS-isolated AT3 and MCA205 ICD–CSCs were stained (4 °C, 30 min) with mAbs anti-PD-L1 (1:100); anti-PD-L2CD1LG (1:100), anti-LGALS9 (1:20) and anti-CEACAM1 (1:100).

To evaluate free nucleic acid-mediated CSC induction, $3 \times 10^5$ tumor cells were cultured in 6-well plates (2 ml medium per well) and OXP-treated (300 µM, 24 h, donor cells). Donor cells were washed and incubated (37 °C, 4 h) in 1.5 ml-Eppendorf microtubes containing growth medium ± 200 IU ml⁻¹ BNZase, 10 IU ml⁻¹ RNase A, 10 IU ml⁻¹ RNase H or 100 IU ml⁻¹ DNase. Donor cells were then cocultured with untreated live (receiving) cells, 24 h ± the indicated nucleases before assessment of CSC surface markers on receiving cells.

**Side population assay.** Hoechst 33342 dye efflux was detected as in ref. [58]. Tumor cells ($1 \times 10^5$) were cultured in 6-well plates (2 ml medium per well) and treated with IFN-I (6×10³ U ml⁻¹, 72 h) or DOX (2.5 µM, 48 h). Cells were washed and incubated in growth medium ± 100 µM VRP (30 min, 37 °C). Hoechst 33342 (5 µg ml⁻¹, 90 min, 37 °C) was added to cell suspension. SP were identified as distinct Hoechst 33342⁻ within PI⁻ cells.

**Quantitative RT–PCR.** Total RNA extraction and genomic DNA removal were performed with the RNeasy Plus Mini Kit. Total RNA (30 ng per sample) was reverse transcribed and amplified using GoTaq Probe 1-Step RT–qPCR System in the presence of the specific primers as listed in Supplementary Table 5. qRT– PCR was analyzed on a StepOnePlus Real-Time PCR System and data invariably normalized to the housekeeping gene *Ppia* expression levels.

**Asymmetric, symmetric division.** NUMB staining: FACS-sorted MCA205 ICD–CSCs, seeded on round coverglass (15 h), were fixed in paraformaldehyde (PFA) 4% D-PBS solution (10 min), washed and blocked (30 min, in 5% BSA, 0.05% TWEEN in D-PBS). Anti-NUMB mAb (1:400, overnight) was added. Slides were stained with secondary (AF) conjugates (1:500) and 10 µM Hoechst 33342. Fluorescence images were visualized, captured and analyzed with Leica-DMI3000 B microscope (100× objective, HCX PL Fluotar, AN1.3), Leica-DFC 310FX camera, and LAS-X acquisition software (Leica Microsystems). At least 100 late anaphases-telophases were analyzed. PKH26 staining: FACS-sorted cells (as above) were seeded and PKH26-stained. Cell divisions were tracked by live

videomicroscopy using Nikon-LIPSI system (Nikon) equipped with IRIS 15 photometrics camera and NIS Element acquisition software. Images were taken every 20 min for 48 h, with a 20× long-range objective (S-PLAN AN 0.4).

**Clonogenic assay.** IFN-I pretreated cancer cells ($1 \times 10^3$ U) were seeded between two layers of 0.4% agarose in CSC medium as in ref. [57] (500 µl, 24-well plates) and incubated under standard culture conditions for up to 15 days. Colonies were fixed/stained with crystal violet (0.02% in 20% methanol) and counted under an inverted microscope. Some spheres, prior fixation, were recovered, cultured in ultralow attachment flasks in CSC medium[57] and analyzed for morphology and transcriptional profiles.

**Multidrug resistance assay.** PAR, IFN–CSC *Kdm1b*[KD], *Kdm1b*[OVER] MCA205, CT26 *Kdm1b*[KD], *Kdm1b*[OVER], B16.F10 *Kdm1b*[KD], *Kdm1b*[OVER] cells ($5 \times 10^3$) were seeded in 96-well plates (90 µl medium per well) and either left untreated or treated with OXP (3–30–300 µM), DOX (0.25–2.5–25 µM), mitoxantrone (0.04–0.4–4 µM) or CDDP (1.5–15–150 µM) for 24–72 h. Cell viability/proliferation was determined by CellTiter-Glo Luminescent Cell Viability Assay via multimode reader (DTX-880; Beckman Coulter).

**T cell proliferation and cancer cell killing assays.** UV-irradiated MCA205-OVA as in ref. [59] were cocultured with BM-derived DCs (2:1 ratio, 24 h). DCs were cultured (5:1 ratio, 72 h) with splenic purified CD8⁺OT-1 cells. Cross-primed CD8⁺OT-1 cells were labeled with carboxyfluorescein succinimidyl ester (CFSE) dye (1 µM, 10 min, 37 °C) and restimulated with live PAR or CD44L MCA205-OVA cells (1:5 ratio, 3 days) before cytofluorometric CFSE level analysis on live-gated CD8⁺ cells and PI level analysis on CD45⁻ cells.

**Microfluidic devices.** H-2Kb splenocytes ($2 \times 10^6$) from C57Bl/6 J mice and $5 \times 10^4$ PAR or ICD–CSC AT3 cells were loaded into the device reservoirs in 200 µl R10. Time-lapse recordings were collected in the incubator for 72 h with a Juli Smart microscope (Bulldog Bio Inc.) that generated one microphotograph every 2 min. ImageJ v1.5 software (Manual Tracking and Trackmate plug-ins) was used for data analysis. For devices based on competition, $2 \times 10^4$ PAR and ICD–CSC AT3 cells were resuspended in Matrigel (3 µl, 2 mg ml⁻¹, on ice) and loaded in two opposite chambers. Splenocytes ($1 \times 10^6$), PKH26 labeled, were loaded in the central chamber in 10 µl R10 (ref. [26]). Phase-contrast, visible and fluorescence microphotographs were generated with the EVOS-FL fluorescence microscope (Life Technologies–Thermo Scientific) and analyzed with ImageJ v1.5 software.

**Extracellular vesicle isolation and uptake.** MCA205 cells ($3 \times 10^5$) were seeded in 6-well plates (2 ml R10 per well), treated with OXP (300 µM OXP, 4 h (donor cells)) and washed. EV from supernatants were purified using exoEasy Maxi Kit and added to receiving cells, 24 h, ± cyto D (0.5 µM). Receiving cells were analyzed by cytofluorometry and qRT–PCR. For uptake analysis, isolated EVs were PKH26 labeled, washed with Exosome Spin Columns and cocultured, 4 h, with receiving cells either at 37 °C or 4 °C. Cells were washed, fixed in 4% PFA in D-PBS and analyzed by cytofluorometry and EVOS-FL fluorescence microscopy.

**Luminex assay.** PAR and ICD–CSCs from MCA205 and AT3 cells were seeded in 24-well plates (1 ml R10 per well, 48 h). Supernatants were collected on ice, centrifuged and immediately frozen (−80 °C). Chemokines were measured by xMAP multiplex technology with Mouse Magnetic Luminex assay multiplex panel as specified in Supplementary Table 5. Analysis was performed with 50 µl of twofold diluted samples. Quantification was performed on a Bio-Plex 200 System (Bio-Rad) equipped with a magnetic workstation and a Bio-Plex Manager Software version 6.1. Chemokine levels were normalized to total cell number.

**ATAC–seq, ChIP-seq and RNA-seq.** PAR, CD44H IFN–CSC, *Kdm1b*[KD] and *Kdm1b*[OVER] MCA205 cells ($1 \times 10^5$) were treated with DNase I (37 °C, 30 min), washed and cryopreserved in R10 plus 5% dimethyl sulfoxide (DMSO) in 1.5-ml vials. Cryopreserved cells were either sent to Epigenetics Services Active Motif, Inc. for ATAC–seq or analyzed at the Regina Elena National Cancer Institute. Cells were thawed and tagmented as in ref. [60]. Tagmented DNA was purified (MinElute PCR purification kit), amplified, repurified (Agencourt AMPure XP beads), quantified (KAPA Library Quantification Kit for Illumina platforms) and sequenced 2 x 100 bp on a Novaseq 6000 instrument (Illumina). For data analysis, reads were aligned to the mouse genome (mm10, BWA algorithm). Duplicate reads were removed and only reads mapping as matched pairs and only uniquely mapped reads (mapping quality ≥1) were considered. Alignments were extended in silico at their 3′-ends to a 200 bp length and assigned to 32-nt bins along the genome. The resulting histograms (genomic 'signal maps') were stored in bigWig files. Peaks were identified using the MACS 2.1.0 algorithm at a cutoff of $P = 1 \times 10^{-7}$, without control file, and with the nomodel option. Peaks on the ENCODE blacklist of known false ChIP–seq peaks were removed. Signal maps and peak locations were used as input data to Active Motifs proprietary analysis program. A peak recalling strategy was used to reduce false positives. ChIP–seq assay in MCA205 CD44H IFN–CSCs were performed as previously described in ref. [61] using anti-LSD2 (1:80). Immunoprecipitations with no specific immunoglobulins

were performed as negative controls. Data analysis was performed as described in ref. [62]. To determine the overall transcriptional profile, $2.5 \times 10^5$ PAR MCA205 cells and their IFN–CSC counterparts were harvested, washed and cryopreserved in RNA-seq analysis performed by Epigenetics Services Active Motif, Inc. Total RNA was isolated from cells (RNeasy Mini Kit), 2 μg of total RNA/sample was used in Illumina's TruSeq Stranded mRNA Library kit. Libraries were sequenced on Illumina NextSeq 500 as paired-end 42-nt reads. Sequence reads were analyzed with the STAR alignment – DESeq2, edgeR, limma-voom software pipelines.

**Transcription factor motif discovery and network analysis.** Motif enrichment analysis was performed with HOMER software comparing TF motifs enriched in target set (from ATAC–seq) versus reference motifs (randomly selected background sequences). Only motif ratios ≥2 with $P \le 0.05$ (Benjamini–Hochberg correction) were considered biologically/statistically significant. The functional enrichment analysis was performed with the clusterProfiler package. Network visualizations were made with the enrichPlot package.

**Generation of $Kdm1b^{KD}$ and $Kdm1b^{OVER}$ cells.** MCA205, AT3, CT26 and B16.F10 cells were seeded at a $7.5 \times 10^3$ in 100 μl growth medium (96-well plates). For KD cells, lentiviral particle (LP) transduction was performed using polybrene (8 μg ml⁻¹) and $4 \times 10^2$ multiplicity of infection (MOI) of shRNA LP targeting Kdm1b or scrambled control. For OVER cells, cDNA encoding Kdm1b gene was cloned into a LP with a bidirectional promoter. Kdm1b (sense orientation) and ΔLNGFR (low affinity nerve growth factor receptor) reporter (antisense orientation) gene expression were driven by hPGK and mhCMV promoter, respectively. LPs were packaged by an integrase-competent third-generation construct and pseudotyped by the VSV envelope. LPs were added to target cells at $1 \times 10^2$ MOI. Cells were centrifuged (30 °C, 1,800g, 90 min) and let in culture, 24–48 h. KD cells were FACS-sorted for green fluorescence protein (GFP) expression and selected with puromycin (1:500). OVER cells were FACS-sorted for ΔLNGFR expression. Transduction efficiency was assessed by qRT–PCR and immunoblot.

**Extreme limiting dilution analysis.** Clonogenic ELDA assays were performed as in ref. [63]. $Kdm1b^{KD}$ or $Kdm1b^{OVER}$ cells were seeded in 96-well plates at doses from 1 to 50 cells per well with 60 replicate wells per cell dose and analyzed by http://bioinf.wehi.edu.au/software/elda/. Wells containing viable adherent cells 2 weeks after plating were scored as positive.

**Cell invasion, migration transwell assay.** Migration ability of $Kdm1b^{KD}$ and $Kdm1b^{OVER}$ cells were measured using Transwell cell culture chambers (8 μM pore size). Cells, $1 \times 10^4$ well, were seeded in 200 μl matrigel diluted 1:4 in RPMI 0.5% FBS in the upper chamber of the Transwell insert. R10 was placed in the lower chamber and incubated, 72 h, in standard culture conditions. Migrated cells were fixed with 4% PFA and stained with 0.2% crystal violet. Nonmigrated cells were removed by wiping the membrane upper side with a cotton swab. Photomicrographs of migrated cells were obtained using an inverted microscope and the percentage of scratch area in five random fields measured using ImageJ v1.5 software.

**Animals.** Mice were maintained in specific pathogen–free standard housing conditions (20 ± 2 °C, 50 ± 5% humidity, 12 h–12 h light–dark cycle, with food and water ad libitum). All in vivo experimentations were in compliance with the EU Directive 63/2010 and included in an experimental protocol approved by the Institutional Animal Experimentation Committee at the Istituto Superiore di Sanità (Rome) and the Italian Ministry of Health (858/2015-PR). Six to seven week-old female C57Bl/6J, NSG, C57Bl/6-Tg(TcraTcrb)1100Mjb/J OT-1 mice were from Charles River, housed in the animal facility at the Istituto Superiore di Sanità and employed after a 7-day acclimatization period. All experiments followed the Guidelines for the Care and Use of Laboratory Animals. A maximal tumor size of 15 mm for the longest axis of the tumor was accepted and was always observed during this study, with only the exception of later time points of therapy experiments (that is, Figs. 2f, 4e and 6g) as differences of tumor size 20–30 days post-treatment were crucial to evaluate therapy response/escape.

**Tumor models, vaccination and chemotherapy.** Tumorigenicity assessment: $1 \times 10^2 – 10^3 – 10^4 – 10^5$ PAR, IFN-CSC, $Kdm1b^{KD}$, $Kdm1b^{OVER}$ MCA205 cells, were subcutaneously inoculated into the flank of C57Bl/6J and NSG mice and tumor surface (longest × perpendicular dimension) routinely monitored using a common caliper. Vaccination experiments: $1 \times 10^5$ PAR MCA205 cells were subcutaneously inoculated in mice which rejected the first injection, and tumor growth monitored weekly. The absence of tumors was considered an indication of efficient vaccination.

Long-term protection of ICD-driven PAR cells on CD44H and CD44L ICD–CSCs: $1 \times 10^6$ PAR MCA205 DOX (25 μM) pretreated were subcutaneously inoculated into the flank of C57Bl/6J. Two weeks later, mice were challenged on the opposite flank with either $1 \times 10^5$ PAR, or CD44H or CD44L cells and tumor growth and mice survival monitored over time.

In vivo CSC induction and IFN–CSC, MCA205 $Kdm1b^{KD}$, $Kdm1b^{OVER}$ therapy response: $1 \times 10^6$ PAR, IFN–CSC, $Kdm1b^{KD}$ or $Kdm1b^{OVER}$ MCA205 cells were

subcutaneously inoculated into the flank of C57Bl/6J mice and tumor growth was monitored weekly. When the tumor surface reached 35–45 mm², mice were randomized to control and treatment groups and injected with D-PBS, CDDP (2.5 mg kg⁻¹), DOX (2.9 mg kg⁻¹), IFN-I ($2 \times 10^4$ U per mouse every other day or $1 \times 10^5$ U per mouse once) all intratumorally in 50 μl D-PBS, TCP (5 mg kg⁻¹) intraperitoneally every 3 days. All experiments contained 5–10 mice per group and were run at least two times, yielding similar results. GraphPad Prism was used for data analysis.

**Tumor dissection, flow cytometry and sorting.** Tumors from mice treated with CDDP, DOX, D-PBS, TCP, DOX + TCP, IFN-I or IFN-I + DOX were carefully removed 15 days post-treatment. Tumor burdens were digested with scissors in RPMI 1640 plus 400 U ml⁻¹ Collagenase-A, 200 U ml⁻¹ DNase I and incubated (30 min, 37 °C). Single cell suspensions obtained by grinding the digested tissue and filtering through a 70-μm cell strainer were purified using mouse CD45 MicroBeads, MACS columns and separators. CD45⁺ cells, including tumor-infiltrating lymphocytes, were resuspended at $1 \times 10^7$ cells ml⁻¹ and stained (4 °C, 30 min) with mAbs anti-CD45 (1:25); anti-CD8a (1:150); anti-TIM3 (1:100). CD45⁻ cells were stained with mAbs anti-CD45, anti-CD133, anti-CD44, anti-CD24 and anti-Nanog (1:5). For gating strategies, see Supplementary Fig. 1.

**In vivo invasiveness assay.** PAR, ICD-CSCs, $Kdm1b^{KD}$ and $Kdm1b^{OVER}$ MCA205 cells ($2 \times 10^5$) were injected into the tail vein of C57Bl/6J mice. In some experiments, mice were treated with 200 μg per mouse anti-CD4 and anti-CD8 Abs in D-PBS, at day-1 and then every 4 days for 15 days. Then, lungs were explanted and macrometastases counted. For CD4–CD8 in vivo depletion, at the end of the experiments, spleens were recovered and analyzed by cytofluorometry. Images of lung metastases were captured with a ZEISS STEMI 305 Stereo microscope (Carl Zeiss). GraphPad Prism was used for data analysis.

**Immunohistochemistry.** Sections (3 μm) of formalin-fixed paraffin-embedded BC biopsies and autologous surgery tissues were cut on SuperFrost Plus slides (Menzel-Gläser). Immunoreactions were revealed by Bond Polymer Refine Detection and ChromoPlex TM1 Dual Detection in an automated autostainer (Bond III, Leica Biosystems) using the following mAbs: mouse anti-CD45 (1:500), rabbit anti-CD133 (1:1000), rabbit anti-CD44 1:100), mouse anti-CD24 (1:100), rabbit anti-IP10 (1:50), the polyclonal rabbit anti-MX1 (1:100) and the recombinant rabbit anti-LSD2/AOF1 (1:500). Chromogenic substrates were diaminobenzidine and Fast Red.

**Patients included in neoadjuvant chemotherapy studies.** Twenty patients (female, 30–77 years old, see Supplementary Table 4), with histologically confirmed BC by the Pathology Unit at the Regina Elena National Cancer Institute, were included. All patients underwent biopsies and received neoadjuvant anthracyclines. This retrospective study was conducted according to the Declaration of Helsinki and, being a part of standard-of-care patient management, did not require a dedicated protocol. All patients signed a written informed consent to treatment and data collection. For metagene correlation analyses, publicly available patient cohorts (accession codes GSE6861, GSE20271, GSE25065, GSE16446, GSE41998, GSE32646, METABRIC) reported in refs. [11,30] were selected. Gene expression analyses were performed on tumor biopsies obtained at diagnosis. Survival analyses were performed by implementing Python (v.3.7.0) scripts. Kaplan–Meier curves for disease-specific survival and distant relapse-free incidence events were computed and drawn using the following Python libraries: lifelines (v.0.26.0, Davidson-Pilon, 2021), matplotlib (v.3.2.2, Hunter, 2007), seaborn (v.0.11.1, Waskom, 2021), numpy (v.1.17.4, Harris et al., 2020), pandas (v.1.0.4, Reback et al., 2021). Differences between Kaplan–Meier curves were evaluated by log-rank test (Bland & Altman, 1998) implemented in the logrank_test function of the lifelines library, and applying a P value threshold = 0.05. Patient stratifications were based on a prognostic index estimation on the SurvExpress online resource (Aguirre-Gamboa, 2013). Patients were stratified by splitting the ordered prognostic index by the median, obtaining two groups with (nearly) equal patient numbers. Gene signatures in correlation and survival analyses were included upon performing the gene set variation analysis as in ref. [64].

**Statistical analysis.** In vitro experiments: no statistical methods were used to determine sample size (n). Experiments were independently repeated at least three times with similar results, with few exceptions in which experiments were repeated twice or one replicate was excluded from the analysis due to technical problems (always specified in figures and/or figure legends). When data were not clear/inconclusive in terms of statistical trends, n was increased (>3) to improve statistical power. For each experiment every sample was processed identically and internal controls and normalization methods were included to avoid technical bias. In vivo experiments: n were defined based on our experience with the experimental models used to detect differences of ≥20% in continuous endpoints between groups (0.05 significance level, 80% statistical power). Exact n for each experimental group/condition, whether n represents technical or biological replicates, are reported in figures and/or figure legends. Data were analyzed with Microsoft Excel (Microsoft) and Prism (v.8.4.0, GraphPad Software), while

statistical analyses were performed using Prism and SPSS software (SPSS v.21, SPSS Inc-IBM). For each dataset of each in vitro experiment conducted at least three independent times, normal distribution was controlled with the Shapiro–Wilk test (SPSS and/or Prism). In case of normal distribution, statistical analysis was performed as follows. Comparisons of two sample groups: unpaired $t$-test, unpaired $t$-test with Welch's correction, depending on the group variance equality (compared using the $F$-test). Comparisons involving more than two sample groups: ordinary one-way analysis of variance (ANOVA) followed by Bonferroni post-hoc test, Brown–Forsythe and Welch one-way ANOVA followed by Dunnett T3 post-hoc test depending on variance equality (assessed with Brown–Forsythe test). Alternatively, in case of data not normally distributed or of two independent experiments, Mann–Whitney and Kruskall–Wallis tests were applied. In vivo growth curves and in vitro splenocyte migration: ordinary two-way RM ANOVA followed by Bonferroni's correction. IHC: Allred scores were calculated to assess the correlation between MX1, CXCL10, KDM1B, CD133 and CD44–CD24 markers. $P$ values <0.05 were considered to be statistically significant. All significant $P$ values are reported in Figs. $P$ values of qRT–PCR studies are reported in Supplementary Table 1, $P$ values of Spearman correlation studies are reported in Supplementary Table 2. Statistics of ELDA assay are reported in Supplementary Table 6. In in vitro experiments involving normalization of treated on untreated conditions, controls are expressed as percentages or FC ± s.e.m. calculated upon normalization on the average of raw control data of all experiments included in each analysis. Data collection and analysis were not performed blind to the conditions of the experiments.

**Reporting summary.** Further information on research design is available in the Nature Research Reporting Summary linked to this article.

## Data availability

All bulk ATAC–seq, ChIP–seq and RNA-seq datasets have been deposited in the Gene Expression Omnibus (GEO) under accession code GSE173851). The following published GEO datasets were also accessed: GSE6861, GSE20271, GSE25065, GSE16446, GSE41998 and GSE32646. Source data are provided with this paper.

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

## Acknowledgements

We thank R. Dattilo, P. Di Matteo, R. Ricci, A. Pacca and M. T. D'Urso (Istituto Superiore di Sanità, Rome, Italy) for technical assistance, E. Proietti and P. Sestili (Istituto Superiore di Sanità, Rome, Italy) for providing IFN-I, L. Zitvogel (Gustave Roussy Cancer Campus, Villejuif, France) for providing MCA205-derived clones, M. Oliviero, R. Albano and the Cell Culture Center (CCC) facility (Candiolo Cancer Institute, FPO - IRCCS, Candiolo, Italy) for providing CT26, B16.F10, U2OS, MCF7 and MCF10 cells, O. Kepp and S. Zhang (Gustave Roussy Cancer Campus, Villejuif, France) for providing MCA205-OVA cells, R. Weinberg (Whitehead Institute for Biomedical Research, Cambridge, MA, USA) for providing HMLER cells and I. Tattoli (Università Cattolica del Sacro Cuore, Rome, Italy) for language and grammar editing. In vivo experiments were performed at Istituto Superiore di Sanità (Rome, Italy). This work was supported by the Associazione Italiana per la Ricerca sul Cancro (AIRC, Start-Up 2016 No. 18418 to A.S. and IG 2017 No. 20417 to I.V.) and the Ministero Italiano della Salute (grant No. RF_GR-2013-02357273 to A.S.). M.M. is supported by the AIRC-FIRC Fellowship No. 25558. L.M. is supported by the AIRC Fellowship No. 26604. The other authors are supported by the AIRC (IG 2018 No. 21366 to G.S.; IG 2019 No. 16895 to M.H.C.; 5×1000 No. 9979 to R.D.M.), the Ministero Italiano della Salute (grant Nos. RF_GR-2016-02364847 to E.R.; RF_RF-2018-12367044 to R.D.M.), the Italian Institute for Genomic Medicine (start-up grant to I.V.) and the Compagnia di San Paolo (grant to I.V.).

## Author contributions

M.M. designed and performed the majority of in vitro and ex vivo experiments with the help of N.M., C.G., E.M. and G.M., and in vivo experiments with the help of A.S., F.G., S.V., D.M. and M.S., analyzed and interpreted data, prepared figures and wrote the manuscript. A.G. analyzed data and performed bioinformatic studies with the help of M.P., M.P., G.C., M.F. and M.H.C. E.R. and A.P. produced lentiviral particles for gene overexpression. L.M. and S.S.A.R. performed immunofluorescence analysis. M.S. performed WB and stereomicroscopic analysis. F.S. performed statistical analysis. A.D.B., C.E. and E.P. performed IHC experiments and analysis. L.P. provided clinical data. T.B. and F.D.N. performed ATAC–seq and ChIP–seq studies. A.D.N. and L.B. designed and realized microfluidic systems. G.S., F.M. and V.L. performed and analyzed experiments on microfluidic devices. F.F., S.R. and G.Z. performed and analyzed Luminex assay. M.S. and A.B. analyzed flow cytometry data. L.B. and E.A. produced IFN-I. M.B. provided infrastructure and preclinical input on the project. I.V. obtained funding, supervised the project, designed experiments and wrote the manuscript. A.S. obtained funding, supervised the project, designed and performed experiments, analyzed data and wrote the manuscript. R.D.M. provided infrastructure, obtained funding, supervised the project, designed experiments and wrote the manuscript.

## Competing interests

The authors declare no competing interests.

## Additional information

**Extended data** is available for this paper at https://doi.org/10.1038/s41590-022-01290-3.

**Correspondence and requests for materials** should be addressed to Ruggero De Maria, Ilio Vitale or Antonella Sistigu.

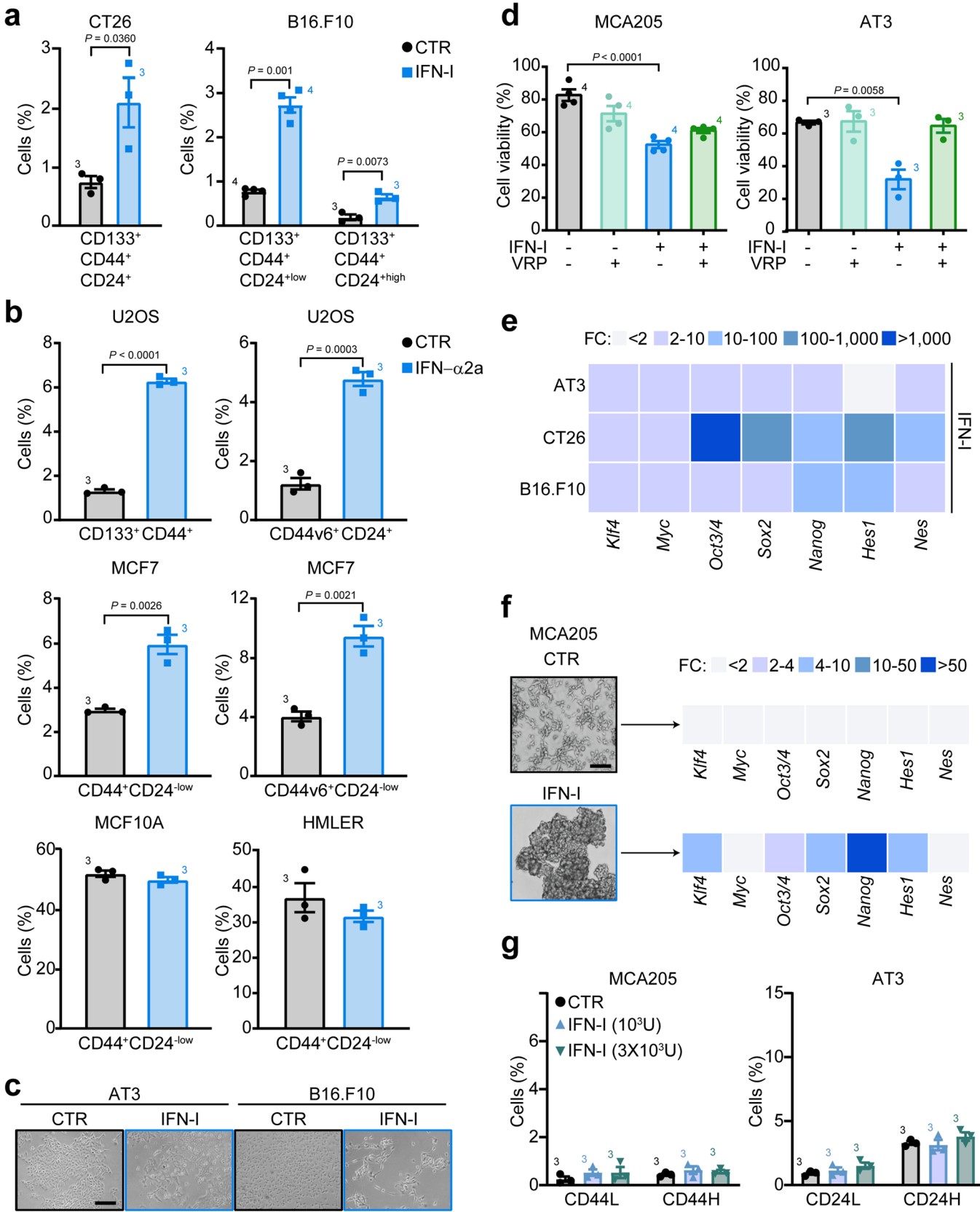

**Extended Data Fig. 1 | See next page for caption.**

**Extended Data Fig. 1 | Type I interferon (IFN-I)-mediated enrichment of putative cancer stem cells (CSCs).** (**a,b**) Multiparametric flow cytometry analysis of the indicated CSC surface markers in CT26 colon carcinoma and B16.F10 melanoma murine cell lines (**a**), and in U2OS osteosarcoma, MCF7 and HMLER breast carcinoma human cell lines and MCF10A epithelial breast cell line (**b**). Cells were treated with mock (control, CTR) or purified IFN-I (murine cells) or recombinant IFN-α2a (human cells) ($6 \times 10^3$ U ml$^{-1}$, 72 h). The percentage (mean ± s.e.m. and individual data points, $n = 3$ and $n = 4$ independent experiments) of CD133$^+$CD44$^+$CD24$^+$ CT26 cells, CD133$^+$CD44$^+$CD24$^{+low}$/CD133$^+$CD44$^+$CD24$^{+high}$ B16.F10 cells, CD133$^+$CD44$^+$/CD44v6$^+$CD24$^+$ U2OS cells, CD44$^+$CD24$^{-low}$/CD44v6$^+$CD24$^{-low}$ MCF7, MCF10A and HMLER cells is shown. (**c**) Representative pictures of AT3 and B16.F10 epithelial cell morphology under mock or purified IFN-I treatment ($n = 3$ independent experiments). Scale bar, 100 μm. (**d**) Flow cytometry analysis showing the proportion of viable (propidium iodide/PI$^-$) MCA205 and AT3 cells left untreated (black) or treated with verapamil (VRP, 100 μM, light green), or purified IFN-I (blue) or VRP + IFN-I (dark green). Data are presented as mean ± s.e.m. and individual data points, $n = 3$ and $n = 4$ independent experiments. (**e**) Expression levels of reprogramming factors in AT3, CT26 and B16.F10 cells treated with purified IFN-I. Data are reported as mean fold change (FC) ± s.e.m. ($n = 2$ biologically independent samples) over untreated cells after intrasample normalization to the levels of *Ppia*. (**f**) Representative images showing the capability of soft-agar-recovered IFN-I-treated MCA205 cells to grow as 3D spheres in standard CSC culture conditions and to maintain a CSC-like transcriptomic profile ($n = 2$ biologically independent samples). Scale bar, 100 μm. (**g**) Multiparametric flow cytometry analysis of CD133$^+$CD24$^+$CD44$^{+low}$ (CD44L) and CD133$^+$CD24$^+$CD44$^{+high}$ (CD44H) in MCA205 cells and of CD133$^+$CD44$^+$CD24$^{+low}$ (CD24L) and CD133$^+$CD44$^+$CD24$^{+high}$ (CD24H) in AT3 cells treated for 10 consecutive days with mock or IFN-I ($1 \times 10^3$ and $3 \times 10^3$ U ml$^{-1}$). Representative biparametric plots and a histogram showing the percentage (mean ± s.e.m. with individual data point, $n = 3$ independent experiments) of CSCs are reported. (**a,b**) Unpaired two-sided Student's *t*-test and unpaired two-sided Student's *t*-test with Welch's correction as compared to CTR cells. (**d**) Brown–Forsythe and Welch one-way ANOVA followed by Dunnett T3 post-hoc tests. (**g**) Ordinary one-way ANOVA test followed by Bonferroni's correction.

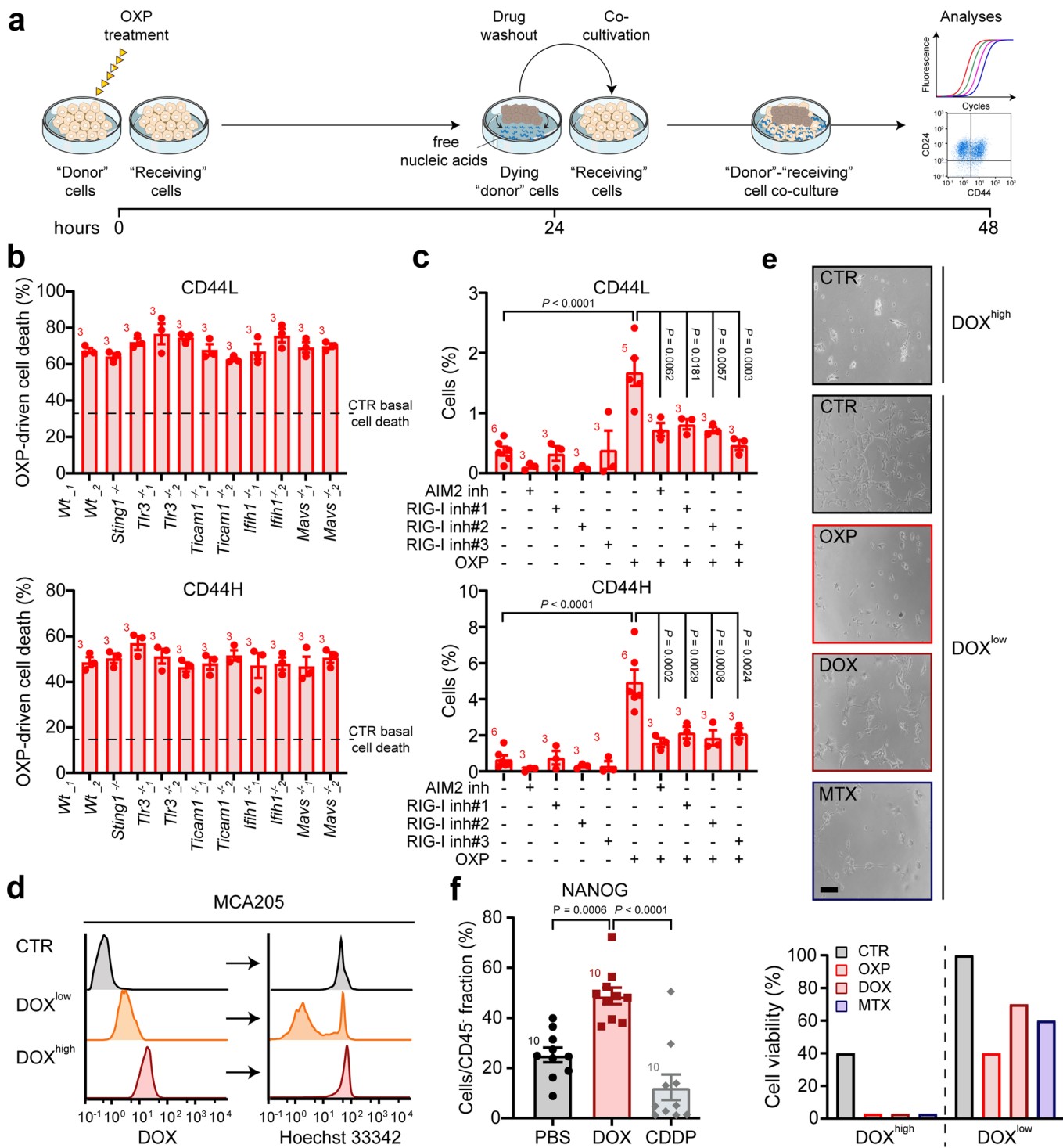

**Extended Data Fig. 2 | See next page for caption.**

**Extended Data Fig. 2 | Immunogenic chemotherapy triggers putative cancer stem cell (CSC) appearance.** (**a**) Schematic representation of the 'donor'-'receiving' cell coculture experimental protocol. (**b**) Flow cytometry analysis showing the induction of cell death upon oxaliplatin treatment (OXP, 300 μM, 24 h) in MCA205 cells with the illustrated genetic background. Data are presented as mean ± s.e.m. and individual data points, $n = 3$ independent experiments. (**c**) Multiparametric flow cytometry analysis of CSC surface markers in MCA205 cells treated with OXP alone or combined with the AIM2 inhibitor thalidomide (AIM2 inh, 10 μg ml$^{-1}$) or inhibitors of the RIG-I pathway amlexanox (RIG-I inh#1, 5 μM), BX795 (RIG-I inh#2, 100 nM) and MRT67307 (RIG-I inh#3, 500 nM). The histograms represent the percentage (mean ± s.e.m. and individual data points; the number of independent experiments) of CD133$^+$CD24$^+$CD44$^{+high}$ (CD44H) and CD133$^+$CD24$^+$CD44$^{+low}$ (CD44L) cells. (**d**) Flow cytometry analysis of doxorubicin (DOX) efflux ability in MCA205 cells left untreated (gray) or exposed to DOX (2.5 μM, 48 h). The two DOX$^{low}$ (orange) and DOX$^{high}$ (red) cell subsets display high and low capability to efflux DOX and Hoechst 33342 (one representative experiment out of three independent experiments). (**e**) Representative pictures of FACS-isolated DOX$^{+low}$ and DOX$^{+high}$ cells in standard culture conditions and under treatment with different chemotherapeutics (DOX$^{+low}$ cells). MCA205 cells were firstly treated with 2.5 μM DOX for 48 h, and then FACS-isolated based on their low or high positivity for red fluorescence. DOX$^{+low}$ and DOX$^{+high}$ sorted cells were then left untreated (control, CTR) or treated with OXP (30 μM), DOX (2.5 μM) or mitoxantrone (MTX, 0.04 μM) for 48 h. Representative pictures from one representative experiment out of two yielding similar results of CTR, DOX$^{+high}$ and treated DOX$^{+low}$ cells are shown. The percentage of counted cells is indicated for each condition, as determined by cell counts on pictures using ImageJ software. Scale bar, 100 μm. (**f**) *Ex vivo* flow cytometric analysis of the percentage of NANOG$^+$ MCA205 cells grown in C57Bl/6 J mice treated intratumorally with vehicle (PBS) or 2.9 mg/kg DOX or 2.5 mg/kg cisplatin (CDDP). Data are presented as mean FC ± s.e.m. and individual data points over PBS treatment for 10 mice/group from 2 experimental replicates. (**b,c,f**) Ordinary one-way ANOVA test followed by Bonferroni's correction.

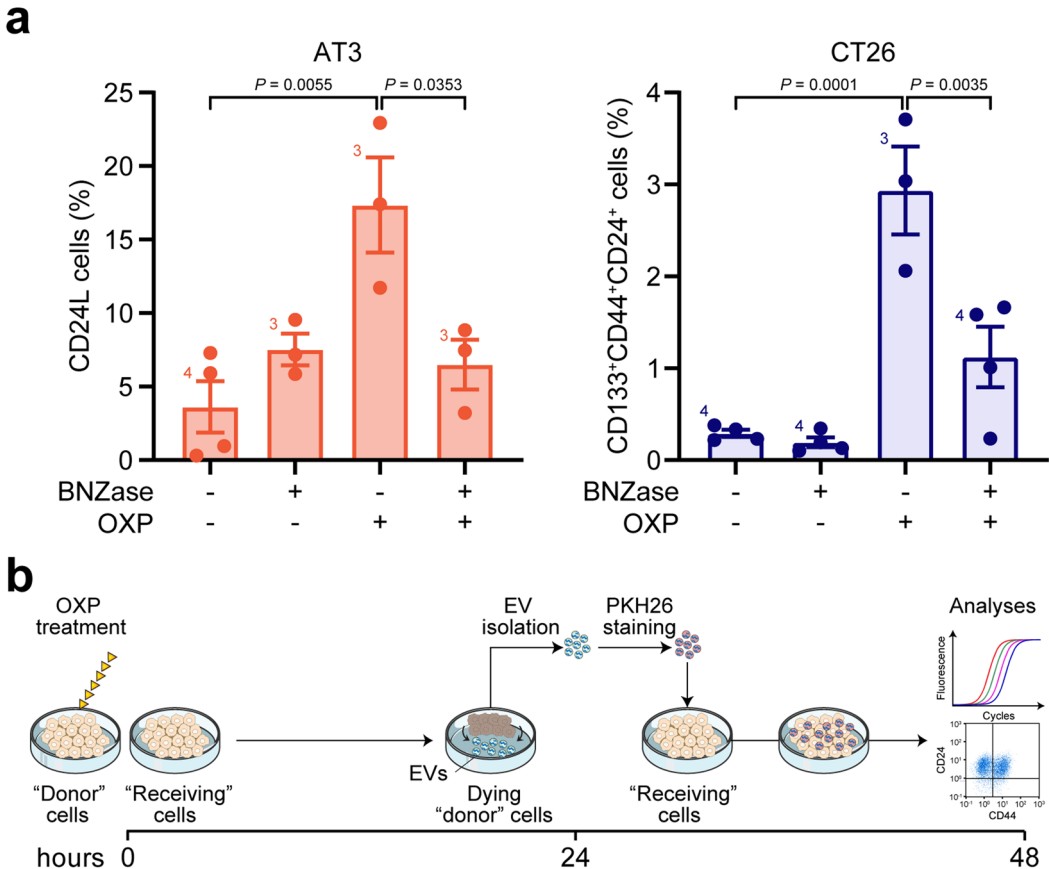

**Extended Data Fig. 3 | Cancer stem cell (CSC) enrichment through nucleic acid transfer.** (**a**) Flow cytometry analysis of CSC surface markers in 'receiving' viable AT3 breast carcinoma and CT26 colon murine carcinoma cells upon coculturing with 'donor' cells of the same type previously treated with oxaliplatin (OXP; 300 μM, 48 h) alone or in combination with benzonase (BNZase; 200 IU ml⁻¹, 48 h). Data are presented as mean ± s.e.m. and individual data points. Number of biologically independent experiments are reported. (**b**) Schematic representation of the extracellular vesicle (EV)-'receiving' cell coculture experimental protocol. (**a**) Ordinary one-way ANOVA test followed by Bonferroni's correction.

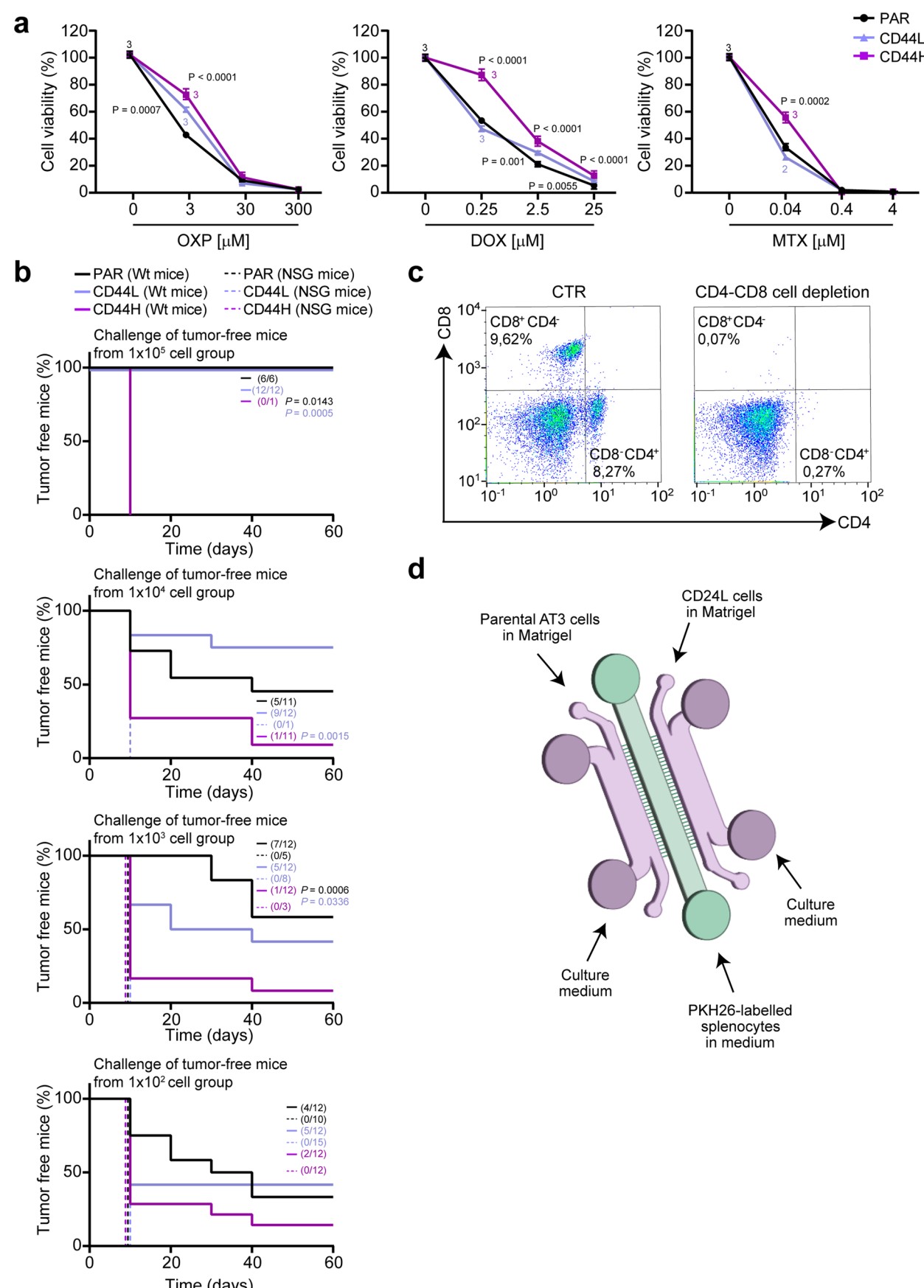

**Extended Data Fig. 4 | See next page for caption.**

**Extended Data Fig. 4 | Characterization of cancer stem cells (CSCs) enriched by type I interferons (IFN-I).** (**a**) Evaluation of cell proliferation/viability by CellTiter-Glo® assay in parental (PAR) and FACS-isolated CD133+CD24+CD44+low (CD44L) and CD133+CD24+CD44+high (CD44H) MCA205 cells (upon enrichment via IFN-I administration) treated for 72 h with oxaliplatin (OXP), doxorubicin (DOX) and mitoxantrone (MTX) as indicated. Results are reported as mean ± s.e.m., $n = 3$ biologically independent experiments. (**b**) In vivo evaluation of the prophylactic potential of PAR MCA205 and immunogenic cell death (ICD)-induced CSCs by using immunocompetent C57Bl/6J (Wild-type/Wt) mice or immunodeficient NSG mice that rejected the injections with PAR, CD44H and CD44L cells at the indicated dose in the experiment reported in Fig. 4b and rechallenging the animals with $1 \times 10^5$ PAR MCA205 in the other flank. The percentage of tumor-free mice is shown. (**c**) Ex vivo flow cytometric analysis of CD4 and CD8 expression in splenocytes from C57Bl/6J mice treated intraperitoneally with vehicle (CTR) or 200 µg/mouse of anti-CD4 and anti-CD8 (200 µg/mouse at day -1 and then every 4 days for 2 weeks). One representative experiment out of two is shown. (**d**) Schematic representation of 'competition' microfluidic devices. CD24L, CD133+CD44+CD24+low. (**a**) Ordinary one-way ANOVA test followed by Bonferroni's correction. (**b**) Log-rank (Mantel-Cox) test.

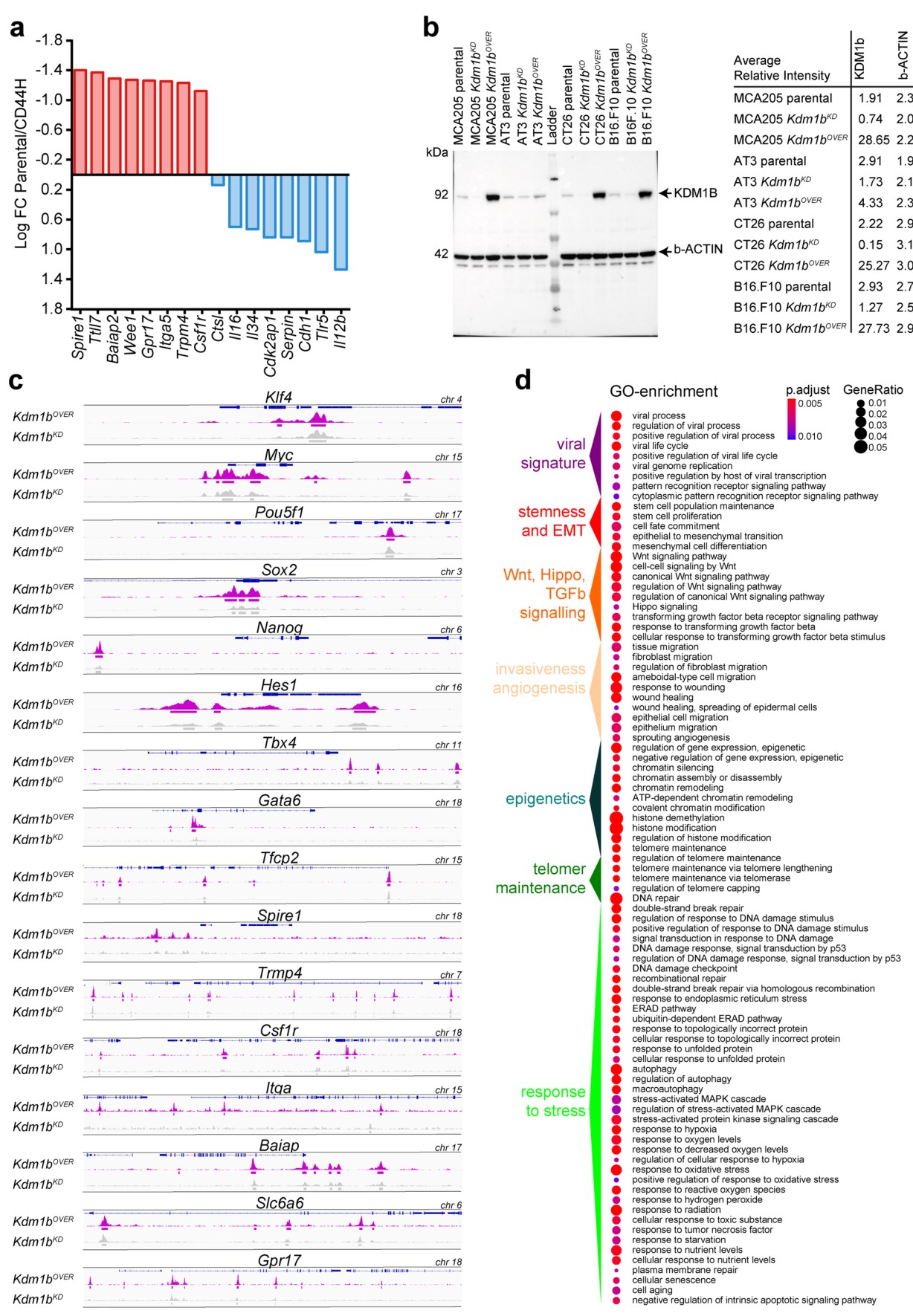

**Extended Data Fig. 5 | See next page for caption.**

**Extended Data Fig. 5 | Chromatin remodeling following type I interferon (IFN-I) exposure.** (**a**) Patterns of gene expression as determined by RNA-seq for representative ATAC–seq-identified genes. Genes upregulated and downregulated in CD133+CD24+CD44+high (CD44H) cells induced by IFN-I are in red and blue, respectively. (**b**) Western-blot (WB) analysis of the levels of KDM1B in the indicated parental (PAR) cell lines and the same cell lines engineered to overexpress or down-express KDM1B (*Kdm1b*OVER and *Kdm1b*KD). Actin beta (b-ACTIN) is used as loading control. The table reports data quantification from one experiment. (**c**) Evaluation of the impact on KDM1B on chromatin remodeling by ATAC–seq. Representative *loci* for the illustrated genes in *Kdm1b*OVER and *Kdm1b*KD MCA205 cells are reported. (**d**) Evaluation of gene regulatory mechanisms downstream of KDM1B by ChIP–seq on immunogenic cell death (ICD)-induced CD44H cells isolated from MCA205 cells and Gene Ontology (GO) terms enrichment analysis. Genes are categorized as illustrated. (**d**) One-sided hypergeometric test followed by Benjamini–Hochberg correction for multiple comparisons.

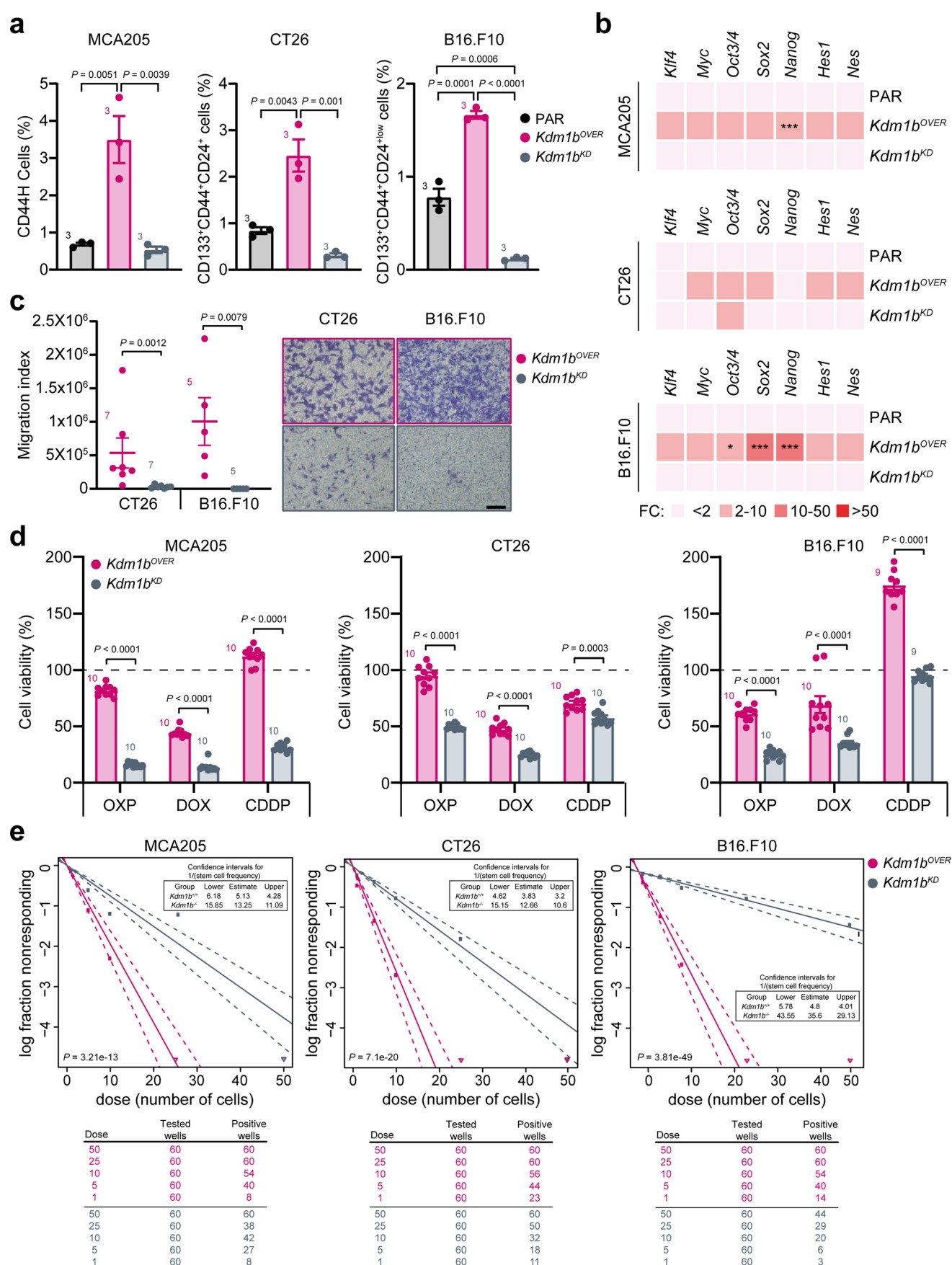

**Extended Data Fig. 6 | See next page for caption.**

**Extended Data Fig. 6 | Impact of KDM1B on cancer stemness, tumorigenicity, and invasiveness.** (**a,b**) Multiparametric flow cytometry analysis of cancer stem cell (CSC) surface markers (**a**) and qRT–PCR analyses of the reported reprogramming factors (**b**) in the indicated parental (PAR) cells and the same cell lines engineered to overexpress or down-express KDM1B (*Kdm1b*$^{OVER}$ and *Kdm1b*$^{KD}$). The histograms in (**a**) represent the percentage (mean ± s.e.m. and individual data points, *n* = 3 biologically independent experiments) of the indicated CSC subpopulation including CD133$^+$CD24$^+$CD44$^{+high}$ (CD44H) MCA205 cells. qRT–PCR data are reported as mean fold change (FC) over untreated condition after intrasample normalization to *Ppia* expression levels. *$P$ < 0.05, **$P$ < 0.01, ***$P$ < 0.001; the exact $P$ values are in Supplementary Table 2. (**c-e**) Evaluation of the assessment of migration ability by transwell assay (**c**), therapeutic response to the reported immunogenic cell death (ICD) inducers and non inducers (**d**) and in vitro tumorigenicity and self-renewal potential by ELDA assay (**e**) in the indicated *Kdm1b*$^{OVER}$ and *Kdm1b*$^{KD}$ cells. Number of biologically independent samples (mean ± s.e.m. and individual data points for **c** and **d**) collected over three independent experiments is reported. (**a,b**) Ordinary one-way ANOVA test followed by Bonferroni's correction as compared to control condition. (**c-e**) Unpaired two-sided Student's *t*-test followed by Welch's correction and two-tailed Mann–Whitney test. Exact calculations for ELDA assay are in Supplementary Table 2.

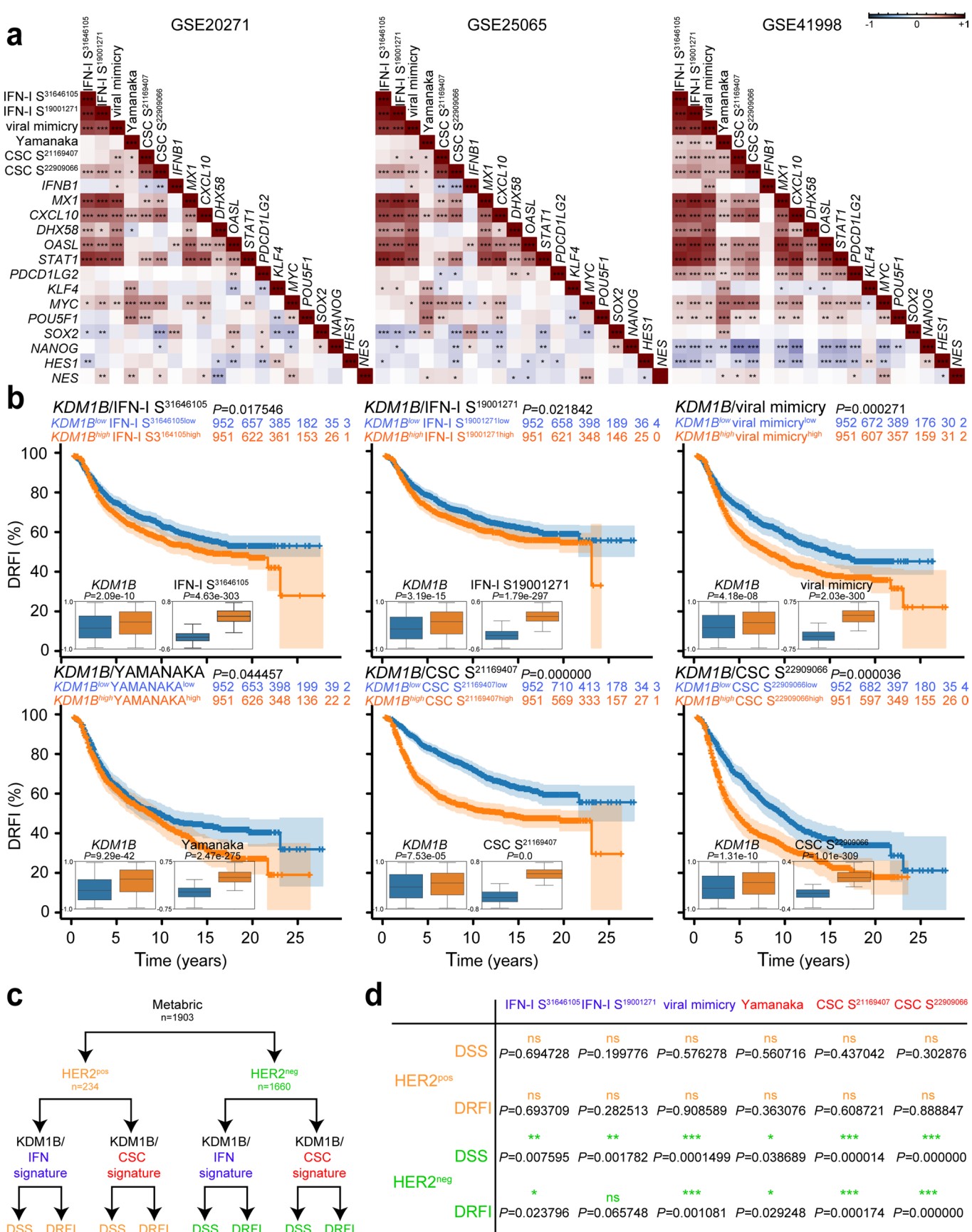

Extended Data Fig. 7 | See next page for caption.

**Extended Data Fig. 7 | Clinical correlation between KDM1B, type I interferon (IFN-I) signature, and stemness signature in breast cancer (BC) patients.**
(**a**) Spearman correlations between expression scores of *KDM1B* and the reported IFN-I-related metagenes, stem-related reprogramming factors, IFN-I signatures and stemness signatures from microarray data of three publicly available cohorts of BC patients treated with neoadjuvant anthracycline-based chemotherapy. *$P < 0.05$, **$P < 0.01$, ***$P < 0.001$. (**b**) Kaplan–Meier plots depicting the distant relapse-free incidence (DRFI) in BC patients from the METABRIC cohort stratified according to risk behavior and boxplots reporting the expression levels of *KDM1B* and the illustrated stemness or IFN-I signatures. *P* value was calculated using the *P* Cox, Log-Rank (Mantel-Cox) test. *P* values <0.05 were considered statistically significant. The relative expression of the indicated genes and signatures is reported as mean ± s.e.m. from 1,903 patients. For statistics of boxplots *see* Supplementary Table 3. The correspondent disease-specific survival (DSS) is reported in Fig. 7b. (**c,d**) Analysis of the combined impact of *KDM1B* and the illustrated stemness and IFN-I signatures on DRFI and DSS on BC patients form the METABRIC database upon their stratification according positivity or negativity to the Erb-B2 Receptor Tyrosine Kinase 2 (ERBB2, best known as HER2). *P* values are calculated as in (**b**). Ns, not-significant. (**a**) Two-sided Spearman's rho.

# Reporting Summary

Nature Research wishes to improve the reproducibility of the work that we publish. This form provides structure for consistency and transparency in reporting. For further information on Nature Research policies, see our Editorial Policies and the Editorial Policy Checklist.

## Statistics

For all statistical analyses, confirm that the following items are present in the figure legend, table legend, main text, or Methods section.

| n/a | Confirmed | |
|---|---|---|
| ☐ | ☒ | The exact sample size ($n$) for each experimental group/condition, given as a discrete number and unit of measurement |
| ☐ | ☒ | A statement on whether measurements were taken from distinct samples or whether the same sample was measured repeatedly |
| ☐ | ☒ | The statistical test(s) used AND whether they are one- or two-sided<br>*Only common tests should be described solely by name; describe more complex techniques in the Methods section.* |
| ☐ | ☒ | A description of all covariates tested |
| ☐ | ☒ | A description of any assumptions or corrections, such as tests of normality and adjustment for multiple comparisons |
| ☐ | ☒ | A full description of the statistical parameters including central tendency (e.g. means) or other basic estimates (e.g. regression coefficient) AND variation (e.g. standard deviation) or associated estimates of uncertainty (e.g. confidence intervals) |
| ☐ | ☒ | For null hypothesis testing, the test statistic (e.g. $F$, $t$, $r$) with confidence intervals, effect sizes, degrees of freedom and $P$ value noted<br>*Give P values as exact values whenever suitable.* |
| ☒ | ☐ | For Bayesian analysis, information on the choice of priors and Markov chain Monte Carlo settings |
| ☒ | ☐ | For hierarchical and complex designs, identification of the appropriate level for tests and full reporting of outcomes |
| ☒ | ☐ | Estimates of effect sizes (e.g. Cohen's $d$, Pearson's $r$), indicating how they were calculated |

*Our web collection on statistics for biologists contains articles on many of the points above.*

## Software and code

Policy information about availability of computer code

| Data collection | Flow Cytometry: BD FACSDiva™(BD Biosciences), MACSQuant® VYB Analyzer 10 (Miltenyi Biotech), CytExpert (Beckman Counter)<br>Cell Sorting: BD FACSDiva™(BD Biosciences)<br>Luminescence Detection: Multimode Detection Software (Beckman Coulter)<br>Conventional immunofluorescence microscopy: EVOS® FL Imaging System operated by embedded Software v.1.4 (Life Technologies-Thermo Scientific), Leica DMI3000 B microscope (HCX PL Fluotar, AN 1.3), Leica DFC 310FX camera, LAS X acquisition software (all from Leica Microsystems, Wetzlar, Germany)<br>Stereo microscopy: ZEN lite imaging (Zeiss)<br>Immunohistochemistry microscopy: LAS V4.8 (Leica)<br>qRT-PCR: StepOnePlus™ Real-Time PCR System operated by embedded StepOne™ Software (ThermoFisher Scientific)<br>RNA quantification: Nanodrop 2000C Spectrophotometer operated by embedded software (ThermoFisher Scientific), Qubit 4 Fluorometer operated by embedded software (ThermoFisher Scientific)<br>Luminex: Bio-Plex Manager v.6.1 (Bio-Rad) |
|---|---|
| Data analysis | Wet lab:<br>Flow Cytometry: FlowJo v.10.0.7 (FlowJo LLC, TreeStar, Inc.), Prism v.8.4 (GraphPad)<br>Analysis of images: Photoshop CC2015, ImageJ v.1.5, Excel 2013 (Microsoft), Prism v.8.4 (GraphPad), LAS X software<br>Tumor growth: Prism v.8.4 (GraphPad)<br>Luminex assay: Bio-Plex Manager Software v.6.1<br>All other experiments: Excel 2013 (Microsoft), Prism v.8.4 (GraphPad)<br><br>In silico:<br>ATAC-seq and Chip-seq: BWA MEM v.0.7.17, MACS v.2.1.0, HOMER v.4.10, Python v.3.7, matplotlib v.3.2.2, seaborn v.0.11.1<br>RNA-seq: GFF (General Feature Format) GENCODE Release M24 GRCm38 biomaRt v.2.42.1, org.Mm.eg.db v.3.10.0, STAR v.2.7.3.a, GATK v.4.1.2.0, featureCounts v.2.0.0, DESeq2 v.1.26.0, edgeR v.3.28.1, limma-voom v.3.42.2, R v.3.6.3 |

Network analysis: clusterProfiler v.3.14.3, org.Mm.eg.db v.3.10.0, enrichplot v.1.6.1, R v.3.6.3
Correlation analysis: GEOquery v.2.54.1, affy v.1.64.0, U133x3p.db v.3.2.3, hgu133a.db v.3.2.3, hgu133plus2.db v.3.2.3, hgu133a2.db v.3.2.3, corrplot v.0.84, RColorBrewer v.1.1.2, ggplot2 v.3.1.1, R v.3.6.3
Patient survival analysis: Python v.3.7, pandas v.1.0.0, numpy v.1.17.4, lifelines v.0.26.0,  Cox-Regression scipy v.1.5.1, (Wilcoxon rank-sum statistic for two samples) matplotlib v.3.3.1, seaborn v.0.9.0

Figure preparation: Illustrator 2020 (Adobe) CC 2015 and CC 2020
Movie preparation: ImageJ v.1.5 and https://veed.io/ for compression

For manuscripts utilizing custom algorithms or software that are central to the research but not yet described in published literature, software must be made available to editors and reviewers. We strongly encourage code deposition in a community repository (e.g. GitHub). See the Nature Research guidelines for submitting code & software for further information.

## Data

Policy information about availability of data

All manuscripts must include a data availability statement. This statement should provide the following information, where applicable:
- Accession codes, unique identifiers, or web links for publicly available datasets
- A list of figures that have associated raw data
- A description of any restrictions on data availability

All data supporting the findings of this study will be available in a publicly accessible repository. The METABRIC patient dataset can be publicly accessed via https://www.cbioportal.org/study/clinicalData?id=brca_metabric. The molecular signature datasets can be publicly accessed at https://www.ncbi.nlm.nih.gov/geo/ with accession codes GSE6861, GSE20271, GSE25065, GSE16446, GSE41998 and GSE32646. All bulk ATAC-seq, Chip-seq, RNA-seq datasets have been uploaded to the Gene Expression Omnibus repository (accession no. HYPERLINK "https://www.ncbi.nlm.nih.gov/geo/query/acc.cgi?acc=GSE173851"GSE173851, https://www.ncbi.nlm.nih.gov/geo/query/acc.cgi?acc=GSE173851).

# Field-specific reporting

Please select the one below that is the best fit for your research. If you are not sure, read the appropriate sections before making your selection.

☒ Life sciences　　☐ Behavioural & social sciences　　☐ Ecological, evolutionary & environmental sciences

For a reference copy of the document with all sections, see nature.com/documents/nr-reporting-summary-flat.pdf

# Life sciences study design

All studies must disclose on these points even when the disclosure is negative.

| | |
|---|---|
| Sample size | In vivo experiments: sample sizes were defined on the basis of our experience with the experimental models used in this study in order to detect differences of 20% or more in continuous endpoints between groups (0.05 significance level and 80% statistical power). In vitro experiments: no statistical methods were used to determine sample size. A minimum of three biologically independent samples were tested, and experiments were performed in at least 2 independent instances (mostly 3) with similar results. When this turned out to be insufficient to clarify statistically sub-significant trends between groups, sample number was increased to improve statistical power. |
| Data exclusions | Outliers or mice with symptoms not linked to cancer were excluded. |
| Replication | All experiments were performed in at least 2 independent instances (mostly 3) with similar results. In each individual experiment, each technical replicate was measured once. |
| Randomization | Mice were randomly allocated to treatment group at tumor detection. For IHC analysis on breast biopsies, intensity score was obtained by calculating 8-10 different fields which were selected randomly. For IF experiments of symmetric vs. asymmetric division, >100 anaphases/telophases were randomly selected. For tracking experiments on microfluidic devices, trajectories of >800 splenocytes were randomly selected. |
| Blinding | In vivo and in vitro analyses were not blinded but kept as unbiased as possible. Data were analysed by software with objective outcomes, and hence blinding was not relevant for the study. For in vitro studies every sample was processed identically to avoid technical bias. Tumor injections and measurements were performed by the same researcher to ensure reproducibility. Proper internal controls and normalization methods were included in each study for internal bias. |

# Reporting for specific materials, systems and methods

We require information from authors about some types of materials, experimental systems and methods used in many studies. Here, indicate whether each material, system or method listed is relevant to your study. If you are not sure if a list item applies to your research, read the appropriate section before selecting a response.

## Materials & experimental systems

| n/a | Involved in the study |
|---|---|
| ☐ | ☒ Antibodies |
| ☐ | ☒ Eukaryotic cell lines |
| ☒ | ☐ Palaeontology and archaeology |
| ☐ | ☒ Animals and other organisms |
| ☐ | ☒ Human research participants |
| ☒ | ☐ Clinical data |
| ☒ | ☐ Dual use research of concern |

## Methods

| n/a | Involved in the study |
|---|---|
| ☐ | ☒ ChIP-seq |
| ☐ | ☒ Flow cytometry |
| ☒ | ☐ MRI-based neuroimaging |

# Antibodies

| | |
|---|---|
| Antibodies used | Rat monoclonal anti-CD133 (13A4) eBioscience™ Cat# 17-1331-81, RRID:AB_823120 https://www.thermofisher.com/antibody/product/CD133-Prominin-1-Antibody-clone-13A4-Monoclonal/17-1331-81 |
| | Rat monoclonal anti-CD24 (M1/69) eBioscience™ Cat# 12-0242-82, RRID:AB_465602 https://www.thermofisher.com/antibody/product/CD24-Antibody-clone-M1-69-Monoclonal/12-0242-82 |
| | Rat monoclonal anti-CD44 (IM7) eBioscience™ Cat# 11-0441-82, RRID:AB_465045 https://www.thermofisher.com/antibody/product/CD44-Antibody-clone-IM7-Monoclonal/11-0441-82 |
| | Rat monoclonal CD44 (IM7) BioLegend® Cat#103020, RRID:AB_493683 https://www.biolegend.com/en-us/products/pacific-blue-anti-mouse-human-cd44-antibody-3099 |
| | Rat monoclonal anti-CD8a (53-6.7) eBioscience™ Cat# 17-0081-82, RRID:AB_469335 https://www.thermofisher.com/antibody/product/CD8a-Antibody-clone-53-6-7-Monoclonal/17-0081-82 |
| | Rat monoclonal anti-CD273 (122) eBioscience™ Cat# 11-9972-81, RRID:AB_465461 https://www.thermofisher.com/antibody/product/CD273-B7-DC-Antibody-clone-122-Monoclonal/11-9972-81 |
| | Mouse monoclonal anti-CD66a (CC1) eBioscience™ Cat# 12-0661-80, RRID:AB_1311201 https://www.thermofisher.com/antibody/product/CD66a-CEACAM1-Antibody-clone-CC1-Monoclonal/12-0661-80 |
| | Mouse monoclonal anti-H2-K1 (AF6-88.5.5.3) eBioscience™ Cat# 11-5958-80, RRID:AB_11151335 https://www.thermofisher.com/antibody/product/MHC-Class-I-H-2Kb-Antibody-clone-AF6-88-5-5-3-Monoclonal/11-5958-80 |
| | Rat monoclonal anti-CD274 (10F.9G2) BioLegend® Cat# 124312, RRID:AB_10612741 https://www.biolegend.com/en-us/products/apc-anti-mouse-cd274-b7-h1-pd-l1-antibody-6655 |
| | Rat monoclonal anti-Galectin-9 (108A2) BioLegend® Cat# 137903, RRID:AB_10568785 https://www.biolegend.com/en-us/products/pe-anti-mouse-galectin-9-antibody-6563 |
| | Rat monoclonal anti-CD366 (RMT3-23) eBioscience™ Cat# 11-5870-82, RRID:AB_2688129 https://www.thermofisher.com/antibody/product/CD366-TIM3-Antibody-clone-RMT3-23-Monoclonal/11-5870-82 |
| | Rat monoclonal anti-CD45 (30-F11) eBioscience™ Cat# MCD4528, RRID:AB_10373710 https://www.thermofisher.com/antibody/product/CD45-Antibody-clone-30-F11-Monoclonal/MCD4528 |
| | Mouse monoclonal anti-CD271 (ME20.4) BioLegend® Cat#53-9400-42, RRID:AB_2802341 https://www.thermofisher.com/antibody/product/CD271-NGF-Receptor-Antibody-clone-ME20-4-Monoclonal/53-9400-42 |
| | Mouse monoclonal anti-CD133/1 (AC133) Miltenyi Biotec Cat# 130-113-106 https://www.miltenyibiotec.com/IT-en/products/cd133-1-antibody-anti-human-ac133.html#apc:100-tests-in-200-ul |
| | Recombinant monoclonal anti-CD44 (REA690) Miltenyi Biotec Cat# 130-113-342 https://www.miltenyibiotec.com/IT-en/products/cd44-antibody-anti-human-reafinity-rea690.html#pe:100-tests-in-200-ul |
| | Human recombinant monoclonal anti-CD133/1 (REA753) Miltenyi Biotec Cat# 130-111-080 https://www.miltenyibiotec.com/IT-en/products/cd133-1-antibody-anti-human-reafinity-rea753.html#gref |
| | Human recombinant monoclonal anti-CD24 (REA832) Miltenyi Biotec Cat# 130-112-845 https://www.miltenyibiotec.com/IT-en/products/cd24-antibody-anti-human-reafinity-rea832.html#pe:100-tests-in-200-ul |
| | Human recombinant monoclonal anti-CD44 (REA690) Miltenyi Biotec Cat# 130-113-903 https://www.miltenyibiotec.com/IT-en/products/cd44-antibody-anti-human-reafinity-rea690.html#fitc:30-tests-in-60-ul |
| | Human recombinant monoclonal anti-CD44 (REAL259) Miltenyi Biotec Cat# 130-120-881 https://www.miltenyibiotec.com/IT-en/products/cd44-antibody-anti-human-realease-real259.html#fitc:100-tests-in-200-ul |
| | Rat monoclonal anti-CD4 (GK1.5) Miltenyi Biotec Cat# 130-120-750 https://www.miltenyibiotec.com/IT-en/products/cd4-antibody- |

anti-mouse-gk1-5.html#biotin:30-ug-in-1-ml

Mouse monoclonal anti-CD24 (ML5) BD Biosciences Cat# BBA13, RRID:AB_356935 https://www.rndsystems.com/products/human-cd24-alexa-fluor-700-conjugated-antibody-ml5_fab5247n

Mouse monoclonal anti-CD44v6 (2F10) R&D Systems Cat# BBA13, RRID:AB_356935 https://www.rndsystems.com/products/human-cd44v6-antibody-2f10_bba13

Rabbit polyclonal anti-MX1 Sigma-Aldrich Cat# HPA030917, RRID:AB_2680862 Lot. B115464 https://www.sigmaaldrich.com/catalog/product/sigma/hpa030917?lang=it®ion=IT

Rabbit monoclonal anti-CD44 (SP37) Sigma-Aldrich Cat# SAB5500068 Lot. 161214C https://www.sigmaaldrich.com/catalog/product/sigma/sab5500068?lang=it®ion=IT

Mouse monoclonal anti-CD24 (SN3) Millipore Cat# CBL561, RRID:AB_11212454 Lot. 2983172 https://www.merckmillipore.com/IT/it/product/Anti-CD24-Antibody-clone-SN3,MM_NF-CBL561

Mouse monoclonal anti-CD45 (2B11+PD7/26) Agilent Technologies Cat# M0701, RRID:AB_2661839 Lot. 20049267 https://www.agilent.com/store/productDetail.jsp?catalogId=M070101-2

Rabbit monoclonal anti-CD133 (EPR16508)  Abcam Cat# AB 222782 https://www.abcam.com/cd133-antibody-epr16508-ab222782.html

Rabbit monoclonal-IP10 (EPR24674-12)  Abcam Cat# AB 283681 https://www.abcam.com/ip10-antibody-epr24674-12-ab283681.html

Goat anti-mouse Alexa Fluor® Plus 488 Thermo Scientific Cat# A32723 https://www.thermofisher.com/antibody/product/Goat-anti-Mouse-IgG-H-L-Highly-Cross-Adsorbed-Secondary-Antibody-Polyclonal/A32723

Goat anti-mouse Alexa FluorTM 488 Thermo Scientific Cat# A21121 https://www.thermofisher.com/antibody/product/Goat-anti-Mouse-IgG1-Cross-Adsorbed-Secondary-Antibody-Polyclonal/A-21121

Goat anti-Rabbit IgG (H+L) Highly Cross-Adsorbed Secondary Antibody, Alexa Fluor™ 555 Invitrogen Cat# A-21429 https://www.thermofisher.com/antibody/product/Goat-anti-Rabbit-IgG-H-L-Highly-Cross-Adsorbed-Secondary-Antibody-Polyclonal/A-21429

Rabbit recombinant anti-LSD2/AOF1 (EPR18508) Abcam Cat# AB193080 https://www.abcam.com/lsd2--aof1-antibody-epr18508-ab193080.html

Mouse monoclonal anti-β-Actin  Sigma-Aldrich Cat# A5441 https://www.sigmaaldrich.com/IT/en/product/sigma/a5441

Rabbit IgG HRP linked whole antibody GE Healthcare Cat# GEHNA9341ML https://www.euroclonegroup.it/search_result

Mouse IgG HRP linked whole antibody GE Healthcare Cat# GEHNA9311ML https://www.euroclonegroup.it/search_result

Rabbit anti-Numb (C29G11)  Cell Signaling Technology Cat# 2756 https://www.cellsignal.com/products/primary-antibodies/numb-c29g11-rabbit-mab/2756

InVivoMAb rat anti-CD4 (GK1.5) Bio Cell Cat# BE0003-1 https://bxcell.com/product/m-cd4/

InVivoMAb rat anti-CD8a (2.43) Bio Cell Cat# BE0061 https://bxcell.com/product/invivoplus-anti-m-lyt-2-2-cd8a/

| | |
|---|---|
| Validation | All antibodies were commercial. Specificity and validation were provided by manufacturer's technical datasheets and confirmed in literature. Link to technical datasheet has been provided above. No further validation was performed. |

# Eukaryotic cell lines

Policy information about cell lines

| | |
|---|---|
| Cell line source(s) | MCA205 (#SCC173) and AT3 (#SCC178) cells were purchased from Merck Sigma-Aldrich, CT26, B16.F10, U2OS, MC7and MCF10A were from ATCC, MCA clones were kindly provided by Pr. Laurence Zitvogel (Gustave Roussy Cancer Campus, France), OVA-expressing MCA205 cells were kindly provided by Dr. Oliver Kepp (Gustave Roussy Cancer Campus, France), HMLER cells were kindly provided by Pr. Robert Weinberg, Kdm1b OVER and Kdm1b KD MCA205, CT26 and B16.F10 cells were specifically produced for this work. |
| Authentication | MCA205 and AT3 cells were used shortly after receipt from commercial vendors and hence were not authenticated. CT26, B16.F10, U2OS, MCF7, MCF10A were routinely validated at Candiolo Cancer Institute, just after thawing via STR Profile System using PowerPlex® 16 HS (Promega), HMLER and MCA.205-OVA cells were not authenticated but in all experiments low passage number cells were used. Properties relevant to the experiments (e.g., OVA and MHC-I expression) were routinely confirmed by flow cytometry or (e.g., Kdm1b overexpression or depletion) western blot and qRT-PCR. |
| Mycoplasma contamination | All cell lines were routinely confirmed to be free from Mycoplasma contamination by PCR. |

| Commonly misidentified lines<br>(See ICLAC register) | none |
| --- | --- |

# Animals and other organisms

Policy information about studies involving animals; ARRIVE guidelines recommended for reporting animal research

| | |
| --- | --- |
| Laboratory animals | Six-to-7 week-old female C57Bl/6J, NOD SCID gamma (NSG) and C57BL/6-Tg(TcraTcrb)1100Mjb/J OT1 mice were purchased from Charles River (Calco, Italy), housed in the animal facility at the Istituto Superiore di Sanità (Rome, Italy) and employed after an acclimatization period of 7 days. Mice were maintained in specific pathogen–free conditions in a temperature-controlled environment (20° +/- 2°C) with 12h light - 12h dark cycles and received food and water ad libitum. |
| Wild animals | None |
| Field-collected samples | None |
| Ethics oversight | All the in vivo experimentations were in compliance with the EU Directive 63/2010 and included in an experimental protocol approved by the Institutional Animal Experimentation Committee at the Istituto Superiore di Sanità (Rome) and the Italian Ministry of Health (approval number 858/2015-PR). |

Note that full information on the approval of the study protocol must also be provided in the manuscript.

# Human research participants

Policy information about studies involving human research participants

| | |
| --- | --- |
| Population characteristics | Twenty breast cancer patients (all with bioptic material before, at diagnosis, and after, at surgery, neoadjuvant anthracycline-based chemotherapy) attending the Division of Medical Oncology 2 at the IRCCS Regina Elena National Cancer Institute (Rome, Italy) were included in this study as part of their standard-of-care clinical management, upon acquisition of written informed consent, between January 2015 and March 2018. Clinical characteristics: median age = 53.5 (30-77), histological type = invasive ductal carcinoma 95%, ductal carcinoma in situ 0%, invasive lobular carcinoma 5%, histological grade at diagnosis = II 30%, II/III 30%, III 35%, unknown 5%, ER status at diagnosis = positive 75%, negative 25%, PR status at diagnosis = positive 65%, negative 35%, HER2 status at diagnosis = positive 80%, negative 20%, Ki-67 status at diagnosis = positive 95%, unknown 5%, number of chemotherapy cycles = 4 5%, 5 5%, 7 5%, 8 70%, 10 5%, unknown 10%, histological grade at surgery = II 20%, III 65%, unknown 15%, ER status at surgery = positive 70%, negative 30%, PR status at surgery = positive 50%, negative 50%, HER2 status at surgery = positive 65%, negative 35%, Ki-67 status at surgery = positive 95%, unknown 5%. Diagnostic and surgical biopsies were studied. |
| Recruitment | Participants were retrospectively included in this study as a part of their standard-of-care management at the IRCCS Regina Elena National Cancer Institute (Rome, Italy). The only criteria fo inclusion were treatment with anthracyclines before surgical resection (neoadjuvant regimen) and sample availabillity. |
| Ethics oversight | IRCCS Regina Elena National Cancer Institute (Rome, Italy). This study was retrospective as a part of standard-of-care patient management, and hence did not require a dedicated study protocol. This study was conducted in accordance with the Declaration of Helsinki. All the patients signed a written informed consent to treatment and data collection. |

Note that full information on the approval of the study protocol must also be provided in the manuscript.

# ChIP-seq

## Data deposition

☒ Confirm that both raw and final processed data have been deposited in a public database such as GEO.

☒ Confirm that you have deposited or provided access to graph files (e.g. BED files) for the called peaks.

| | |
| --- | --- |
| Data access links<br>*May remain private before publication.* | *For "Initial submission" or "Revised version" documents, provide reviewer access links. For your "Final submission" document, provide a link to the deposited data.* |
| Files in database submission | *Provide a list of all files available in the database submission.* |
| Genome browser session<br>(e.g. UCSC) | *Provide a link to an anonymized genome browser session for "Initial submission" and "Revised version" documents only, to enable peer review. Write "no longer applicable" for "Final submission" documents.* |

## Methodology

| | |
| --- | --- |
| Replicates | *Describe the experimental replicates, specifying number, type and replicate agreement.* |
| Sequencing depth | *Describe the sequencing depth for each experiment, providing the total number of reads, uniquely mapped reads, length of reads and whether they were paired- or single-end.* |
| Antibodies | Rabbit recombinant anti-LSD2/AOF1 (EPR18508) Abcam Cat# AB193080 |

| Peak calling parameters | *Specify the command line program and parameters used for read mapping and peak calling, including the ChIP, control and index files used.* |
|---|---|
| Data quality | *Describe the methods used to ensure data quality in full detail, including how many peaks are at FDR 5% and above 5-fold enrichment.* |
| Software | *Describe the software used to collect and analyze the ChIP-seq data. For custom code that has been deposited into a community repository, provide accession details.* |

# Flow Cytometry

## Plots

Confirm that:

☒ The axis labels state the marker and fluorochrome used (e.g. CD4-FITC).

☒ The axis scales are clearly visible. Include numbers along axes only for bottom left plot of group (a 'group' is an analysis of identical markers).

☒ All plots are contour plots with outliers or pseudocolor plots.

☒ A numerical value for number of cells or percentage (with statistics) is provided.

## Methodology

| Sample preparation | In vitro experiments: to assess the expression of specific surface markers on putative-induced CSCs, 1 x 105 murine and human tumor cells were cultured in 6-well plates in 2 mL of growth medium and treated 72h with purified mouse IFN-α/β or recombinant human Roferon-A® (6000 U/mL) or with DOX (25 μM) or OXP (300 μM) alone or in combination with TCP (10 μM) for 48h. Cells were then collected, washed in Dulbecco's Phosphate-Buffered Saline (D-PBS) and stained with fluorescently labeled mAbs directed against human/murine CD44, CD133 and/or CD24, or with purified-CD44v6 mAb, at optimal mAb concentrations (dilution 1:20, as previously determined by titration), in a cold D-PBS solution containing 1% FBS (D-PBS-FBS 1%). Samples were incubated in the dark on ice for 30min and then washed twice with cold D-PBS-FBS 1% solution. Thereafter, cells stained with CD44v6 mAb, were co-stained with the appropriate Alexa Fluor® 488 secondary Ab (diluted at 1:500 in D-PBS-FBS 1%) on ice for 30min. Cells were washed twice before the addition of 150 μL growth medium supplemented with 1 μg/mL DAPI.  For the assessment by flow cytometry of the expression of immune checkpoint molecules, FACS-isolated ICD-CSCs from AT3 and MCA205 cells were stained at 4°C for 30min in the dark with the following murine fluorochrome-conjugated mAbs directed against: PD-L1 (diluted at 1:100); PD-L2CD1LG (diluted at 1:100), LGALS9 (diluted at 1:20) and CEACAM1 (diluted at 1:100). DAPI was used to distinguish live and dead cells, and analysis of the expression of immune checkpoint molecules was made only in live cells. To evaluate how free nucleic acids contribute to the acquisition of CSC traits, 3 x 105 murine tumor cells were cultured in 6-well plates (2 mL of medium/well) and treated with 300 μM OXP for 24h ("donor" cells). Thereafter, "donor" cells were collected, washed from OXP and incubated at 37°C for up to 4h in 1,5 mL-eppendorf microtubes containing growth medium, supplemented or not, with 200 IU/mL BNZase, 10 IU/mL RNase A, 10 IU/mL RNase H or 100 IU/mL DNase. Next, such "donor cells" were cocultured with untreated live cells ("receiving" cells) for 24h in the presence or not of the indicated nucleases before cytofluorometric-mediated assessment of CSC surface markers on "receiving" cells. For the side-population (SP) assay, 1 x 105 murine tumor cells were cultured in 6-well plates (2 mL of medium/well) and treated with 6000 U/mL IFN-α/β for 72h, or 2.5 μM DOX for 48h. Cells were then collected, washed and incubated in pre-warmed growth medium in the presence or not of 100 μM VRP for 30min at 37°C. Five μg/mL Hoescht 33342 was added to cell suspension for 90min at 37°C in the dark. For T cell proliferation and cancer cell killing assays, MCA205-OVA were UV irradiated and co-cultured with BM-derived DCs at a 2:1 ratio for 24h. DCs were then cultured at a 5:1 ratio with splenic purified CD8+ OT-1cells for 72h. Cross-primed CD8+ OT-1 cells were then labelled with 1 μM CFSE dye for 10min at 37°C, and re-stimulated with live parental or CD44L MCA205-OVA cells at 1:5 ratio. Three days later, cells were recovered and analyzed for CFSE levels on live gated CD8+ cells and PI levels on CD45- cells.<br><br>Ex vivo experiments: tumors from mice either treated with CDDP, DOX, D-PBS, TCP, DOX+TCP, acute high dose IFNs-I, chronic low dose IFNs-I, chronic low dose IFNs-I+DOX were carefully removed 15 days after treatment. Tumor burdens were cut into small pieces with scissors within digesting buffer (400 U/ml Collagenase A and 200 U/ml DNase I in RPMI 1640) and incubated for 30min at 37°C. Single cell suspensions obtained by grinding the digested tissue and filtering them through a 70-μm cell strainer were then purified based on CD45 expression, by using mouse CD45 MicroBeads, MACS columns and separators (used following manufacter's recommandations). After washing with D-PBS, CD45+ cells, uncluding tumor infiltrating lymphocytes (TILs), were resuspended at 1 x 107 cells/mL and stained at 4°C for 30min in the dark with the following murine-specific fluorochrome-conjugated mAbs directed against: CD45 (diluted at 1:25); CD8a (diluted at 1:150); and TIM-3 (diluted at 1:100). Similarly, the CD45- cellular fraction (including tumor cells) was stained as follows: CD45, CD133, CD44, CD24 and Nanog (diluted at 1:5). DAPI and Sytox blue were used to distinguish live and dead cells and only live cells were included in the analysis. |
|---|---|
| Instrument | BD FACSCantoTM II (BD Biosciences), MACSQuant® VYB Analyzer 10 (Miltenyi Biotec), CytoFLEX (Beckman Counter) |
| Software | FlowJo v.10.0.7 (FlowJo LLC, TreeStar, Inc.) |
| Cell population abundance | Sorted cells were >90% pure, as determined by FACS reanalysis |
| Gating strategy | A relevant gating strategy is described in Extended Data 1. Briefly, surface or intracellular markers were quantified within DAPI- cells upon gating on cells (SSC-A vs FSC-A) and singlets (SSC-A vs SSC-H). |

☒ Tick this box to confirm that a figure exemplifying the gating strategy is provided in the Supplementary Information.

