## [Peer Review File · Nature Immunology]

Peer Review Information

Journal: Nature Immunology

Manuscript Title: Type I IFNs promote cancer cell stemness by triggering the epigenetic regulator KDM1B

Corresponding author name(s): Ruggero De Maria, Ilio Vitale, and Antonella Sistigu

Reviewer Comments & Decisions:

Decision Letter, initial version:
--

Subject: Decision on Nature Immunology submission NI-A31940A

Message: 10th May 2021

Dear Dr Sistigu,

Your Article, "Type I IFNs promote cancer cell stemness by triggering the epigenetic regulator KDM1B" has now been seen by 2 referees. You will see from their comments copied below that while they find your work of considerable potential interest, they have raised quite substantial concerns that must be addressed. In light of these comments, we cannot accept the manuscript for publication, but would be very interested in considering a revised version that addresses these serious concerns. The Referees find that several aspects are either confusing, preliminary or not well substantiated. For example the evidence for the chronicity of IFNAR signaling affecting the outcome of the response is not felt to be very convincing (if you recall this was a point I raised at the presub stage). There are multiple other experimental and technical requests which I suspect will need to be carefully addressed for the manuscript to be endorsed by these Referees.

We hope you will find the referees' comments useful as you decide how to proceed. If you wish to submit a substantially revised manuscript, please bear in mind that we will be reluctant to approach the referees again in the absence of major revisions. Please do not hesitate to get in touch if you would like to discuss these issues further.

If you choose to revise your manuscript taking into account all reviewer and editor comments, please highlight all changes in the manuscript text file.

* If you have not done so already please begin to revise your manuscript so that it conforms to our Article format instructions at <http://www.nature.com/ni/authors/index.html>. Refer also to any guidelines provided in this letter.

The Reporting Summary can be found here:
<https://www.nature.com/documents/nr-reporting-summary.pdf>

You may use the link below to submit your revised manuscript and related files:
[REDACTED]

If you wish to submit a suitably revised manuscript we would hope to receive it within 6 months. If you cannot send it within this time, please let us know. We will be happy to consider your revision so long as nothing similar has been accepted for publication at Nature Immunology or published elsewhere.

Nature Immunology is committed to improving transparency in authorship. As part of our efforts in this direction, we are now requesting that all authors identified as 'corresponding author' on published papers create and link their Open Researcher and Contributor

Identifier (ORCID) with their account on the Manuscript Tracking System (MTS), prior to acceptance. ORCID helps the scientific community achieve unambiguous attribution of all scholarly contributions. You can create and link your ORCID from the home page of the MTS by clicking on 'Modify my Springer Nature account'. For more information please visit www.springernature.com/orcid.

Thank you for the opportunity to review your work.

Sincerely,

Zoltan Fehervari, Ph.D.
Senior Editor
Nature Immunology

The Macmillan Building
4 Crinan Street
Tel: 212-726-9207
Fax: 212-696-9752
z.fehervari@nature.com

Referee expertise:

Referee #1: cancer immunology, cell death

Referee #2: cancer immunology, clinical

Reviewers' Comments:

Reviewer #1:

Remarks to the Author:

In the manuscript "Type I IFNs promote cancer cell stemness by triggering the epigenetic regulator KDM1B" (#NI-A31940A), Musella and colleagues describe a novel signaling axis whereby type I interferon (IFN) supports the accumulation of cancer stem cells (CSCs) with increased ability to evade immune recognition via epigenetic modifications imposed by lysine demethylase 1B (KDM1B).

The message of the article is important, novel and within scope for Nature Immunology. Findings are abundant, technically diverse and well presented, and this reviewer had NO concerns about image manipulation or data fabrication. Globally, the authors must be congratulated on the abundant amount of work and quality of presentation. However, some of the conclusions, especially in the discussion, are not supported by the dataset, and hence must be appropriately revisited (or additional experiments performed in this respect). Alongside, a few experiments are missing that would considerably strengthen the paper, and some minor issues emerged that need to be addressed, as detailed here below. In summary, this reviewer believes the article by Musella and colleagues represent

a strong candidate for publication in Nature Immunology upon revision.

MAJOR ISSUES (not in order of importance, very important point indicated)

1. **IMPORTANT:** the authors interpret their dataset in several instances as to suggest that acute (vs. chronic) type I IFN signaling is responsible for CSC accumulation, and openly suggest in the discussion that "Based on our results, we surmise that, when produced at chronic low levels, IFNs-I limit CSC proliferation and survival, restraining tumor growth. At odds, acute IFN-I production, as during immunogenic chemotherapy, favors the survival of pre-existing CSCs and cancer cell de-differentiation, potentially leading to therapy resistance/failure". However, the authors never compared settings of chronic type I IFN signaling to their current experimental setting (which actually this reviewer is not sure is modelling acute exposure), especially in vivo. Since the aforementioned conclusions is openly at odds with an abundant hi-profile literature (including previous work from part of the same team, see Sistigu et al, Nat Med), the authors need to either revisit their interpretation or experimentally demonstrate that chronic type I IFN signaling indeed promote tumor control in the absence of CSC accumulation, whereas acute type I IFN is detrimental linked to CSC accumulation.
2. Results page 7. The authors are encouraged to provide some perspective on magnitude of increase (perhaps even in the discussion) in the CSC fraction, especially as post-stimulation % remain always below 10% and in some cases <2% (see Suppl. 1 CT26 cells)
3. Fig. 1C. A very significant fraction of cell death occurs, including in control conditions (for AT3 cells). The authors should repeat the experiments, especially for AT3, to ensure control cells are in good shape. Also, cell death driven by type I IFN should at least be discussed as it may be linked to CSC accumulation based on the author's own model
4. **IMPORTANT:** the authors suggest that ICD induction favors the accumulation of immunoprivileged CSCs, which is somehow at odds with previous literature. In Fig. 4C-D, they test this concept by vaccinating mice with parental cells or CSCs with different CD44 expression and challenging mice with parental cells, concluding that CD44Hi CSCs are less immunogenic as only 50% of the mice rejected the challenge injection. A reverse experiment involving vaccination with ICD-driven parental cells and challenge with the 3 cell population would add additional insights into the process, irrespective of what results ultimately will be.
5. Fig. 2b. Authors should control and discuss for differential cell death driven by OXP in CD44H vs CD44L cells, including probably change imposed by genotypes
6. **IMPORTANT.** Fig. 2b. Authors interpret saying that "ICD-CSC enrichment was impaired only in *Ifnar*^{-/-} clones", which is not what that figure shows. Figure shows abrogation in *Ifnar*^{-/-} clones and various degrees of impairment in other clones, which by the way is in line with nucleic acid degradation data. Authors should perform stats between OXP-treated cells of different genotype and rephrase the text
7. Figure 2e. Again, a large amount of cell death is disregarded, which should instead be quantified and discussed
8. Figure 2g. Results should be plotted as % of cells over total live cells or over CD45neg cells, not as FC.
9. **IMPORTANT.** According to the author model, nucleic acids transferred from dying cells to cells surviving (suboptimal) ICD induction are responsible for type I IFN signaling, which they demonstrate by adding multiple NA degraders to co-cultures. Alongside, they propose a key role for EVs in NA transfer to surviving cells, as demonstrated by *CtyD* experiments in F 3C. This reviewer is confused. Are NA degraders accessing EVs? Are NA transferred on the EV surface instead? The authors should test the ability of exogenous free NAs vs liposome-encapsulated NAs (various NAs in various mixes, in the presence vs absence of NA degraders) to drive accumulation of CD44H and CD44L CSCs. Also, it would

be great to see the effect of IRF3 KO, but this reviewer acknowledges it may be beyond the scope of the current investigation

10. Fig. 3E. This should be quantitatively or semi-quantitatively compared to EVs from untreated cells

11. Fig. 4C. Number of mice is low calling for statistical validation by Chi square (and/or increase in number of mice. Tumor-free survival curves are preferable as they document kinetic of tumor development. Also, this reviewer was surprised to see 100 cells generating tumors in NSG mice, and hence suggest double checking calculations

12. Fig. 4E. Metastatic potential should be compared in NSG mice, too, to check for the relative involvement of stemness. Even better, CD4/CD8 depletion should be implemented in immunocompetent mice, as NSG are highly immunodeficient.

13. Results. Page 12. The authors use the term upregulation in many instances, but show (Fig. 5C) increased percentage of positive cells. These two concepts are distinct. Authors should show MFI if they want to refer to upregulation or alter phrasing in the text

14. Fig. 6G. How does TCP impact on tumor growth and animal survival in this context? According to the author model, beneficially, and it would be a great addition to test it

15. Fig. 7B. The arbitrary choice of gene couples is highly questionable as it comes across as a pick the winner situation. Authors should strongly consider to test individual genes or large genesets covering all genes of interest instead, unless they can rationally justify their picks. As many papers previously linked large type I IFN signatures (as opposed to single ISGs, which have contradictory outputs) to detrimental disease outcome in patients with BC (see 19001271 or 31646105), it would also more interesting to see whether these signatures correlate with KDM1B levels or with stemness signatures in the same datasets. In the opinion of this reviewer surely more interesting that prognostic assessments based on gene couples.

16. Fig. 7F. CSC increase is observed in a minority of patients post-therapy (15%), perhaps suggesting that optimally dosed anthracycline-based chemotherapy in patients most often prevents (instead of promoting) CSC accumulation. In view of previous data from Sistigu et al (suggesting that BC patients with high MX1 levels benefit from anthracyclines), these findings may actually support a model where strong type I IFN signaling upon chemotherapy (as documented by high MX1 levels, and as opposed to indolent in pre-treatment biopsies) may be beneficial and most often prevent CSC accumulation. I would tone down conclusions in this respect (see my first point above) and include previous findings (these and others mentioned above) in the discussion

MINOR ISSUES

1. Abstract. Authors should refrain from using the word "immortal"

2. Introduction. The first paragraph is very generic and should be shortened/refocused

3. Introduction. The authors point to CSCs as "highly proliferating" citing a review from Battle and Clevers. This is not the most commonly accepted view for CSCs and even Battle and Edwards suggest that CSCs largely evade treatment by adopting a quiescent phenotype. I would rephrase to stay out of controversy

4. Introduction. Authors use reference 24 to support the concept that type I IFN can mediate pro-tumorigenic effects. Reference 24 demonstrates that the rejection of mouse cancer cells by immunocompetent hosts relies on type I IFN, and hence does not support the author sentence.

5. Introduction. The authors should avoid introducing "neologisms" in original articles. Those are better suited for potential commentaries post-publication.

6. Introduction. Type I IFN should not be called a chemoattractant, that's a secondary effect mediated by CXCL9/10 (as indicated by the authors)

7. Results. Gating on positive and negative cells is often across main populations.

Although this is obviously appropriate, this reviewer suggest to include in supplementary

material a gating strategy that include sample with no secondary antibody, so that gating approach can be nicely appreciated by readers

8. Suppl. Fig. 2B. VRP only is missing in MCA205 cells

9. Suppl. Fig. 2D. Quantitative data should be included

10. Fig. 3A. OXP treatment does not increase the percentage of CD44H cells, which is internally incoherent with 2B

11. Fig. 3B. A representative picture with less red background would be preferable if available

12. Fig. 4A. Tumor growth curves should be compared as whole by two-way anova computing time and treatment as variables, not at specific time points

13. Fig. 4B. It's unclear and add little, I would remove it or present it differently.

14. Page 13. Please avoid using "fleeting"

15. Most of the genes mentioned in page 15 are stemness-related genes, not surprising that they are upregulated in the CSC compartment. I suggest refocusing this discussion on immune genes of interest instead

16. An entire field would comment on the wrong use of "autophagic cell death" in Fig. 6e

17. Fig. 6H should be removed as alternative models may equally explain results. This is more appropriate for commentary -like pieces.

18. Fig. 7A. What about merged analysis of all datasets alongside? That would surely be beneficial from a statistical standpoint if results are consistent across cohorts.

Reviewer #2:

Remarks to the Author:

In this study, the team investigated the role type 1 IFN in CSCs. They propose that nucleic acids are released from tumor cells exposed to immunogenic chemotherapy. These nucleic acids, which are encapsulated in extracellular vesicles or not, are transported to by-standing tumor cells and induce type 1 IFN production via activating nucleic acid sensors. Moreover, type 1 IFN induce CSC enrichment by upregulating KDM1B, which make the elements in the chromatin more accessible to a number of TFs, such as HTH and bHLH.

Overall, I find this an interesting yet preliminary study. The detrimental effects of IFN on tumor treatments have been reported. Moreover, the general idea that nucleic acids from immunogenic dead cells induce type 1 IFN secretion has also been shown in their previous study(Sisigu et al Nature medicine 2014). However, their core conclusion here, that IFN promotes CSCs via upregulating chromatin remodeling factor KDM1B is potentially a novel and exciting finding. The major issues of this study are that the evidences for CSC enrichment are not robust, and the role of KDM1B has not been rigorously demonstrated.

Major points

1. They propose that IFN increases two CD44L and CD44H CSC subsets in MCA205 cells. However, the tumorigenicity of the CD44L cells was consistently lower than the one of their parental cells in Fig 4c (0% vs 66% 105 in immunocompetent mice; 0% vs 50% 102 in NSG...). Even the tumorigenicity of CD44H cells was worse than the one of their parental cells at the 102 level in NSG mice. The strong tumorigenicity is the central feature of CSCs. How to explain this discrepancy?

2. The numbers of representative plots are inconsistent to the quantitations. For example, AT3 IFNs-CD24H and IFNs-CD24L are 26% and 13% respectively in representative plots of Fig 1a. However, their means in quantitations below are about 10% and 2.5% respectively, whereas the SEM is very small. I recommend to repeat these experiments.

3. Additional experiments, such as assays for ALDH1 and asymmetric division (PKH26 and Numb staining), are required to verify the features of CSCs at various junctions.
4. I am surprised that the authors used MDA-MB-231 to investigate the CSCs. Almost all of MDA-MB-231 cells exhibit a CD44+CD24- CSC-like phenotype. Therefore, few CSC study use MDA-MB-231. I recommend the cell lines commonly used in this scenario, including MCF-7, MCF-10A or HMLER.
5. Whether the immunoactivating ISGs, such as IFN, IL-12 and MHC, increase in IFN-treated tumor cells?
6. In Fig.2, what is the rationale to investigate these selective nucleic acid sensors? Have they excluded the effects of other DNA (such AIM2) or RNA (RIG-I) sensors?
7. They propose that DNA released by dead cells activates STING in by-standing CSCs. How can DNA in the endosomes activate cytosolic cGAS?
8. Although the p values are not shown, RNase A, RNase H and DNase markedly increase CD44H and CD44L % without OXP (3rd, 4th and 5th groups in Fig3a). Is this effect really ICD-specific?
9. They show that EVs contain mRNAs for reprogramming factors and invasion molecules (Fig. 3e). What do they want to propose? mRNAs in EVs are translated into pro-tumor proteins in recipient cells? Any evidences? How to reconcile with the nucleic acid-sensing mechanism?
10. The animal number in each group is too small (n=2-4) in Fig4b,c,and e. This is probably why the CD44H is more tumorigenic in the NSG 103 group, while less tumorigenic in the 102 group compared with parental cells. It is recommended that these experiments are repeated with n=6 from at least two separate experiments, for purposes of reproducibility.
11. The evidences for the chromatin remodeling by KDM1B are circumstantial. They show the different motif enrichment for various TFs between CSCs and parental cells. However, whether it is caused by KDM1B or IFN is unknown. Moreover, they only used one inhibitor to show that KDM1B affects CD44H percentages and TIM3 expression in TILs. I recommend to perform ATAC-seq in the gain (overexpression KDM1B in parental cells) and loss (silencing KDM1B by shRNAs or knockout KDM1B by CRIPSR/Cas9 in IFN-treated cells) experiments to show whether KDM1B really regulates the accessibility of these TFs to chromatin.
12. Similarly, the authors should show that whether gain and loss of KDM1B may affect the CSC, drug resistance, tumorigenic and metastatic properties in multiple cell lines.
13. It is very important to show the clinical significance of KDM1B in their paired samples before and after NAC. Whether levels of KDM1B in human tumors are correlated with CSC features? Is it associated with therapeutic response? Is it an independent prognostic factor? Any difference in different breast cancer subtypes? These are the basic questions to address if they want to show the clinical significance of their major finding.
14. It is very difficult to accurately assess the CD44+CD24-/low phenotype by IHC. Few study use this approach to evaluate CSCs in tissue. I recommend ALDH1 staining.

Minor points.

1. "Cancer stem cells (CSCs) are immature, immortal cells within tumors, adept at resisting therapeutic pressure and responsible for local and distant disease recurrence." It is better to say "Cancer stem cells (CSCs) are a key population of tumor cells that are highly tumorigenic, chemoresistant and metastatic in many cancer types"
2. "however molecular insights into this evolutionary process still lack." It is better to say "However, molecular insights into the evolutionary process of CSCs are still limited"
3. Why the Pdl1 band is so smearing in Fig3e? The authors should show the marker bands

or add dashes to locate their exact position.
4. Please provide scale bars in Fig 5h.

Author Rebuttal to Initial comments

Point-by-point reply to the Reviewer #1

Remarks to the Authors: *In the manuscript "Type I IFNs promote cancer cell stemness by triggering the epigenetic regulator KDM1B" (#NI-A31940A), Musella and colleagues describe a novel signaling axis whereby type I interferon (IFN) supports the accumulation of cancer stem cells (CSCs) with increased ability to evade immune recognition via epigenetic modifications imposed by lysine demethylase 1B (KDM1B). The message of the article is important, novel and within scope for Nature Immunology. Findings are abundant, technically diverse and well presented, and this reviewer had NO concerns about image manipulation or data fabrication. Globally, the authors must be congratulated on the abundant amount of work and quality of presentation. However, some of the conclusions, especially in the discussion, are not supported by the dataset, and hence must be appropriately revisited (or additional experiments performed in this respect). Alongside, a few experiments are missing that would considerably strengthen the paper, and some minor issues emerged that need to be addressed, as detailed here below. In summary, this reviewer believes the article by Musella and colleagues represent a strong candidate for publication in Nature Immunology upon revision.*

Our response: We thank the reviewer for the encouraging words, and for his/her constructive remarks and suggestions, which helped us significantly improve our paper.

Specific comments have been addressed in the responses below.

Major point 1 raised by Reviewer #1: *IMPORTANT: the authors interpret their dataset in several instances as to suggest that acute (vs. chronic) type I IFN signaling is responsible for CSC accumulation, and openly suggest in the discussion that "Based on our results, we surmise that, when produced at chronic low levels, IFNs-I limit CSC proliferation and survival, restraining tumor growth. At odds, acute IFN-I production, as during immunogenic chemotherapy, favors the survival of pre-existing CSCs and cancer cell de-differentiation, potentially leading to therapy resistance/failure". However, the authors never compared settings of chronic type I IFN signaling to their current experimental setting (which actually this reviewer is not sure is modelling acute exposure), especially in vivo. Since the aforementioned conclusions is openly at odds with an abundant hi-profile literature (including previous work from part of the same team, see Sistigu et al, Nat Med), the authors need to either revisit their interpretation or experimentally demonstrate that chronic type I IFN signaling indeed promote tumor control in the absence of CSC accumulation, whereas acute type I IFN is detrimental linked to CSC accumulation.*

Our response: The point is very well taken and the referee is right in his/her understanding of the data presented. To address the question, we performed several complementary analyses in which we compared the impact of acute vs. chronic type I IFN (IFN-I) signaling on cancer stemness, demonstrating, that (i) acute, but not chronic IFNs-I induce CSC accumulation, both *in vivo* and *in vitro*, and (ii) chronic IFN-I signaling promotes tumor control in the absence of CSC accumulation.

In more detail, in a first series of *in vivo/ex vivo* experiments, MCA205 tumors grown in the subcutis of immunocompetent C57Bl/6J mice were treated locally (*i.e.*, via peritumoral injection) either once with high levels IFNs-I (1×10^5 U/mouse; high acute IFNs-I condition) or multiple times with low levels of IFNs-I (2×10^4 U/mouse, low chronic IFNs-I condition). In the same experiments, mice were also subjected to a single injection of doxorubicin (DOX, an immunogenic drug inducing high levels of IFNs-I), to a single injection of doxorubicin in combination with chronic IFNs-I (DOX+low chronic IFNs-I) or to a single injection of cisplatin (CDDP, a non-immunogenic drug inducing low levels of IFNs-I). See the schematic protocol reported in **panel a** below.

As endpoints we analysed (i) the *in vivo* tumor growth control, by routinely monitoring tumor surface, (ii) the percentage of putative CSCs, by performing *ex vivo* flow cytometry assessment of pluripotency genes and other stem cell-associated genes at different experimental time points, and (iii) the induction of viral mimicry, by conducting qRT-PCR studies of the expression of a panel of interferon-stimulated genes selected in our previous work (PMID: 25344738), which constitute a specific IFN-I signature used throughout the article. We dubbed this signature “viral mimicry”.

We were able to demonstrate that high acute IFNs-I promote a significant accumulation of CD44H CSCs, while low chronic IFNs-I do not enrich for CSCs (panel **b**). Moreover, we provided evidence that, when used as adjunctive to DOX treatment, low chronic IFNs-I prevented CSC accumulation (panel **c**). In line with this evidence, we showed that low chronic IFNs-I favor tumor control and animal survival after DOX administration, even upon treatment discontinuation (panel **d**).

Through *ex vivo* experiments, we confirmed our previous findings (PMID: 25344738), showing that cancer cells engage an IFN-I response early after (day 2) exposure to DOX while viral mimicry is significantly reduced at later time points (day 15). Of note, the levels of *Kdm1b* were increased 15 days after intratumoral injection of DOX or high acute IFNs-I (when CSC enrichment occurs) but not in chronic IFNs-

I conditions, lending further support to our findings on the link between KDM1B and acute IFN-I-mediated enrichment/induction of CSCs (panel e).

These results are shown above and have been included in the new **Figure 1f**, **Figure 2e,f** and **Supplementary Figure 1g** and discussed in the text.

In a parallel series of *in vitro* experiments, we treated MCA205 and AT3 murine cancer cell lines with two different doses of low chronic IFNs-I (1×10^3 or 3×10^3 U/mL, for 10 days) or high acute IFNs-I (6×10^3 U/mL, for 3 days), and then analysed CSC enrichment/induction by flow cytometry. As summarized below and as reported in new **Figure 1a** (high acute IFNs-I) and **Supplementary Figure 1g** (low chronic IFNs-I), differently from acute IFNs-I, low chronic IFNs-I do not induce CSC accumulation.

Major point 2 raised by Reviewer #1: Results page 7. The authors are encouraged to provide some perspective on magnitude of increase (perhaps even in the discussion) in the CSC fraction, especially as post-stimulation % remain always below 10% and in some cases <2% (see Suppl. 1 CT26 cells).

Our response: Driven by the reviewer's remark, we described our results also reporting the magnitude of increase of CSCs. The reviewer is right in his/her request, as it is well demonstrated that the frequencies of CSCs in many solid tumors tend to be relatively low, even when measured under the most permissive conditions (PMID: 22704512).

Major point 3 raised by Reviewer #1: Fig. 1C. A very significant fraction of cell death occurs, including in control conditions (for AT3 cells). The authors should repeat the experiments, especially for AT3, to ensure control cells are in good shape. Also, cell death driven by type I IFN should at least be discussed as it may be linked to CSC accumulation based on the author's own model.

Our response: We thank the reviewer for this remark. We repeated the experiments implementing few modifications in sample preparation in order to limit cell death in control conditions due to technical

issues. Moreover, in the revised manuscript we analysed the impact of type I IFNs on cell death and discussed the obtained results.

Major point 4 raised by Reviewer #1: *IMPORTANT: the authors suggest that ICD induction favors the accumulation of immunoprivileged CSCs, which is somehow at odds with previous literature. In Fig. 4C-D, they test this concept by vaccinating mice with parental cells or CSCs with different CD44 expression and challenging mice with parental cells, concluding that CD44^{Hi} CSCs are less immunogenic as only 50% of the mice rejected the challenge injection. A reverse experiment involving vaccination with ICD-driven parental cells and challenge with the 3 cell population would add additional insights into the process, irrespective of what results ultimately will be.*

Our response: Driven by the reviewer's constructive critique, we evaluated the long-term protection of C57Bl6/J mice that received parental (PAR) MCA205 cells previously treated with a lethal dose of DOX, against re-challenge with viable PAR cells, CD44H-ICD CSCs or CD44L-ICD CSCs. As the reviewer will appreciate, we confirmed that CD44H-ICD CSCs are significantly more tumorigenic and less immunogenic than PAR and CD44L-ICD CSC counterparts. Indeed, we observed that the injection of 1×10^5 CD44H-ICD CSCs resulted in tumor development in more than 80% of control unvaccinated mice, while that of 1×10^5 PAR or CD44L-ICD CSCs resulted in tumor development in less than 60% and 30%, respectively. Moreover, we demonstrated that DOX-treated PAR cells were able to vaccinate 85% of mice against PAR and CD44L-

ICD CSCs, but only 30% of mice challenged with CD44H-ICD CSCs. These findings are shown in the following figure and in new **Figure 4c**.

Major point 5 raised by Reviewer #1: Fig. 2b. Authors should control and discuss for differential cell death driven by OXP in CD44H vs CD44L cells, including probably change imposed by genotypes.

Our response: In the revised version of the manuscript, we have included and discussed data showing that the absence of major players of the IFN-I pathway does not affect cell sensitivity to OXP (new **Supplementary Figure 2b**).

Major point 6 raised by Reviewer #1: IMPORTANT. Fig. 2b. Authors interpret saying that “ICD-CSC enrichment was impaired only in *Ifnar*^{-/-} clones”, which is not what that figure shows. Figure shows abrogation in *Ifnar*^{-/-} clones and various degrees of impairment in other clones, which by the way is in line with nucleic acid degradation data. Authors should perform stats between OXP-treated cells of different genotype and rephrase the text.

Our response: We thank the reviewer for this constructive remark. We performed an extensive statistical analysis between OXP-induced CSCs of different genotype using the Bonferroni’s multiple comparisons test. All the results are reported below. In this analysis, we revealed that *Ifnar*^{-/-}, *Sting*^{-/-}, *Tlr3*^{-/-}, *Ticam*^{-/-}, *Ifih1*^{-/-} and *Mavs*^{-/-} clones show a statistically significant reduction of CD44H CSCs as

compared to both *Wt* clones. Moreover, we observed a significant reduction of the percentage of CD44L CSCs in all knock-out clones as compared to *Wt₂* clone and in *Ifnar^{-/-}* and *Mavs^{-/-}* clones as compared to *Wt₁* clone. As the reviewer correctly pointed, these results indicate various degrees of impairment also in *Sting^{-/-}*, *Tlr3^{-/-}*, *Ticam^{-/-}*, *Ifih1^{-/-}* and *Mavs^{-/-}* clones, a relevant finding that we discussed in the text. However, in **Figure 2b** we decided to report only the statistical comparison untreated vs. treated within each clone to not overload the figure and to focus more on the impact of each IFN-I pathway deficiency on CSC enrichment following treatment with immunogenic drugs (OXP).

Bonferroni's multiple comparisons test		CD44H			CD44L		
		P value	summary	change	P value	summary	change
Wt₆ vs	Ifnar-4	<0,0001	****	↓	<0,0001	****	↓
	Ifnar-6	<0,0001	****	↓	<0,0001	****	↓
	Sting	<0,0001	****	↓	>0,9999	ns	
	Tlr3_3	<0,0001	****	↓	>0,9999	ns	
	Tlr3_4	0,0013	**	↓	0,1437	ns	
	Trif_1	<0,0001	****	↓	0,7865	ns	
	Trif_2	0,0005	***	↓	>0,9999	ns	
	Mda5_1985	0,0049	**	↓	>0,9999	ns	
	Mda5_1989	0,0031	**	↓	>0,9999	ns	
	Mavs_40	<0,0001	****	↓	0,0028	**	↓
	Mavs_45	<0,0001	****	↓	0,0126	*	↓
	Wt₁₀ vs	Ifnar-4	<0,0001	****	↓	<0,0001	****
Ifnar-6		<0,0001	****	↓	<0,0001	****	↓

	Sting	0,0064	**	↓	<0,0001	****	↓
	Tlr3_3	0,024	*	↓	<0,0001	****	↓
	Tlr3_4	0,4485	ns	↓	<0,0001	****	↓
	Trif_1	0,0352	*	↓	<0,0001	****	↓
	Trif_2	0,1993	ns	↓	<0,0001	****	↓
	Mda5_1985	>0,9999	ns	↓	<0,0001	****	↓
	Mda5_1989	0,9536	ns	↓	<0,0001	****	↓
	Mavs_40	<0,0001	****	↓	<0,0001	****	↓
	Mavs_45	0,0002	***	↓	<0,0001	****	↓
Ifnar-4 vs	Sting	>0,9999	ns	↑	<0,0001	****	↑
	Tlr3_3	>0,9999	ns	↑	<0,0001	****	↑
	Tlr3_4	>0,9999	ns	↑	<0,0001	****	↑
	Trif_1	0,1152	ns	↑	<0,0001	****	↑
	Trif_2	>0,9999	ns	↑	<0,0001	****	↑
	Mda5_1985	0,2503	ns	↑	0,0007	***	↑
	Mda5_1989	0,0349	*	↑	0,0474	*	↑
	Mavs_40	0,0535	ns	↑	>0,9999	ns	
	Mavs_45	>0,9999	ns	↑	>0,9999	ns	
Ifnar-6 vs	Sting	>0,9999	ns	↑	<0,0001	****	↑
	Tlr3_3	0,6851	ns	↑	<0,0001	****	↑
	Tlr3_4	0,0464	*	↑	<0,0001	****	↑
	Trif_1	0,4951	ns	↑	<0,0001	****	↑
	Trif_2	0,1028	ns	↑	<0,0001	****	↑
	Mda5_1985	0,0138	*	↑	0,001	***	↑
	Mda5_1989	0,0212	*	↑	0,065	ns	
	Mavs_40	>0,9999	ns	↑	>0,9999	ns	
	Mavs_45	>0,9999	ns	↑	>0,9999	ns	
Sting vs	Tlr3_3	>0,9999	ns	↑	>0,9999	ns	
	Tlr3_4	>0,9999	ns	↑	>0,9999	ns	
	Trif_1	>0,9999	ns	↑	>0,9999	ns	
	Trif_2	>0,9999	ns	↑	>0,9999	ns	
	Mda5_1985	>0,9999	ns	↑	>0,9999	ns	
	Mda5_1989	>0,9999	ns	↑	0,6093	ns	
	Mavs_40	>0,9999	ns	↓	0,0008	***	↓
	Mavs_45	>0,9999	ns	↓	0,0031	**	↓
Tlr3_3 vs	Trif_1	>0,9999	ns		>0,9999	ns	
	Trif_2	>0,9999	ns		>0,9999	ns	
	Mda5_1985	>0,9999	ns	↑	>0,9999	ns	↓
	Mda5_1989	>0,9999	ns	↑	0,1305	ns	↓
	Mavs_40	>0,9999	ns	↓	0,0002	***	↓
	Mavs_45	>0,9999	ns	↓	0,0006	***	↓
Tlr3_4 vs	Trif_1	>0,9999	ns		>0,9999	ns	
	Trif_2	>0,9999	ns		>0,9999	ns	
	Mda5_1985	>0,9999	ns	↑	0,155	ns	↓
	Mda5_1989	>0,9999	ns	↑	0,0024	**	↓
	Mavs_40	0,115	ns	↓	<0,0001	****	↓
	Mavs_45	0,9405	ns	↓	<0,0001	****	↓

Trif_1 vs	Mda5_1985	>0,9999	ns	↑	0,7102	ns	↓
	Mda5_1989	>0,9999	ns	↑	0,0125	*	↓
	Mavs_40	>0,9999	ns	↓	<0,0001	****	↓
	Mavs_45	>0,9999	ns	↓	<0,0001	****	↓
Trif_2 vs	Mda5_1985	>0,9999	ns	↑	>0,9999	ns	↓
	Mda5_1989	>0,9999	ns	↑	0,1847	ns	↓
	Mavs_40	0,2498	ns	↓	0,0002	***	↓
	Mavs_45	>0,9999	ns	↓	0,0008	***	↓
Mda5_1985 vs	Mavs_40	0,0348	*	↓	0,0282	*	↓
	Mavs_45	0,3119	ns	↓	0,1052	ns	↓
Mda5_1989 vs	Mavs_40	0,0534	ns	↓	>0,9999	ns	↓
	Mavs_45	0,4646	ns	↓	>0,9999	ns	↓

Major point 7 raised by Reviewer #1: *Figure 2e. Again, a large amount of cell death is disregarded, which should instead be quantified and discussed.*

Our response: In the revised manuscript we decided to remove side population's data from **Figure 2** because we added more relevant results showing ICD-induced CSC enrichment and to avoid figure overload.

Major point 8 raised by Reviewer #1: *Figure 2g. Results should be plotted as % of cells over total live cells or over CD45neg cells, not as FC.*

Our response: The data in the new **Figure 2e** and new **Supplementary Figure 2f** have been plotted as % of CD44H⁺ and NANOG⁺ cells over live CD45⁻ cells, respectively, as suggested by the reviewer.

Major point 9 raised by Reviewer #1: *IMPORTANT. According to the author model, nucleic acids transferred from dying cells to cells surviving (suboptimal) ICD induction are responsible for type I IFN signaling, which they demonstrate by adding multiple NA degraders to co-cultures. Alongside, they propose a key role for Evs in NA transfer to surviving cells, as demonstrated by CtyD experiments in F 3C. This reviewer is confused. Are NA degraders accessing Evs? Are NA transferred on the EV surface instead? The authors should test the ability of exogenous free Nas vs liposome-encapsulated Nas (various Nas in various mixes, in the presence vs absence of NA degraders) to drive accumulation of CD44H and CD44L CSCs. Also, it would be great to see the effect of IRF3 KO, but this reviewer acknowledges it may be beyond the scope of the current investigation.*

Our response: We do understand the reviewer's hesitation with regard to the mechanisms of nucleic acid horizontal transfer as a main *stimulus* for IFN-related CSC induction. As recommended by the reviewer, we compared the effect of various nucleases (*i.e.*, RNase A for ssRNA, RNase H for dsRNA, DNase for DNA and benzonase for the complete digestion of all nucleic acids, either alone or in various mixes) on synthetic

nucleic acids (*i.e.*, Poly I:C as dsRNA analog, R848 as a ssRNA analog and CpG as a DNA analog, either alone or in various mixes) both free and encapsulated into cationic liposomes.

As the reviewer will appreciate, we found that free synthetic nucleic acids, either alone or in various combinations, induce CD44H and CD44L cells and that this induction is abrogated by specific nucleases (*i.e.*, CpG induction is abrogated by DNase and benzonase, Poly I:C induction is abrogated by RNases and benzonase and R848 induction by RNase H and benzonase). Of note, CD44H and CD44L enrichment under exposure to R848-Poly I:C-CpG combination is affected by all nucleases, but abrogated only by benzonase, thus validating our hypothesis that the activation of each of the pathways upstream of IFNs-I could induce cancer cell stemness and only the blocking of all of them could prevent this reprogramming. When

exposing MCA205 cells to liposomes previously charged with the synthetic nucleic acids as above, either alone or in various combinations, we could appreciate an enrichment of CD44H and CD44L cells, which validates our data on EVs. Of note, in this setting, the addition of nucleases does not significantly affect CSC percentages, suggesting that nucleic acids are in part transferred on the surface of EVs and in part internalized and, once engulfed, are released onto the cytoplasm of receiving cells. These findings are summarized in the heatmaps above but were not included in the revised manuscript due to lack of space. Please, note that heatmap color code refers to CD44H and CD44L percentages and statistic is relative to the effect of enzymes on CSC induction.

Concerning IRF3 role in CSC enrichment, as shown in the *ex-vivo* analysis of tumors following DOX-based chemotherapy and acute vs. chronic IFN-I treatment (see major point 1), we did not find a significant modulation of expression in any condition analysed. We did not deepen this analysis, but it is likely that IRF3 is not crucial for CSC induction.

Major point 10 raised by Reviewer #1: *Fig. 3E. This should be quantitatively or semi-quantitatively compared to Evs from untreated cells.*

Our response: qRT-PCR data have been included in new **Figure 3e**.

Major point 11 raised by Reviewer #1: *Fig. 4C. Number of mice is low calling for statistical validation by Chi square (and/or increase in number of mice). Tumor-free survival curves are preferable as they document kinetic of tumor development. Also, this reviewer was surprised to see 100 cells generating tumors in NSG mice, and hence suggest double checking calculations.*

Our response: Driven by the reviewer's critique, we extended our *in vivo* analyses to 12-to-15 mice per group, also reporting tumor-free survival curves. We were able to confirm that CD44H ICD-CSCs are significantly more tumorigenic than CD44L ICD-CSCs and PAR cells, and this both in immunocompetent and immunodeficient hosts. In these experiments, we also confirmed that in some cases CD44H ICD-CSCs generate tumors in NSG mice even at the lowest number of injected cells (100), which is in line with their stemness nature. These data are shown here (panel **a** tumorigenicity, panel **b** challenge) and have been included in the new **Figure 4b** and new **Supplementary Figure 4b**.

Major point 12 raised by Reviewer #1: Fig. 4E. Metastatic potential should be compared in NSG mice, too, to check for the relative involvement of stemness. Even better, CD4/CD8 depletion should be implemented in immunocompetent mice, as NSG are highly immunodeficient.

Our response: In the revised version of our manuscript, we have included data showing that, under CD4 and CD8 T cell depletion, CD44L ICD-CSCs develop lung metastases, yet significantly less than CD44H ICD-

CSCs and PAR cells. Instead, in highly immunodeficient NSG mice, also CD44L ICD-CSCs were able to extensively graft and grow in the lungs. These findings confirm the considerable difference in terms of immunogenicity between CD44H and CD44L ICD-CSCs, the former showing complete immune evasion while the latter still controlled partially by T cells and mostly by other immune cell subsets (likely natural killer cells). Data are shown below and in the new **Figure 4d** and new **Supplementary Figure 4c**.

Major point 13 raised by Reviewer #1: Results. Page 12. The authors use the term upregulation in many instances, but show (Fig. 5C) increased percentage of positive cells. These two concepts are distinct. Authors should show MFI if they want to refer to upregulation or alter phrasing in the text.

Our response: Driven by the reviewer's suggestion, we rephrased the text.

Major point 14 raised by Reviewer #1: Fig. 6G. How does TCP impact on tumor growth and animal survival in this context? According to the author model, beneficially, and it would be a great addition to test it.

Our response: We thank the reviewer for raising this point. We combined local treatment of DOX (2,9 mg/kg at T0) with systemic (intraperitoneal) injection of tranlycypromine (TCP, 5 mg/kg at T0 and then twice per week till the end of the experiment), as reported in the schematic protocol of panel a below. We found that combined therapy significantly improves tumor growth control and mice survival (panel b), while reducing CSC appearance and TIM3 surface levels on tumor-infiltrating CD8 T cells (panel a, *ex vivo* analyses). These data are shown here and have been included in new **Figure 6f,g**.

We further corroborated these results by engineering various cancer cell lines to either overexpress (OVER) or silence (KD) *Kdm1b*, as suggested by reviewer #2 (see our reply to major point 12 of reviewer #2 and new **Figure 6h-k**, new **Supplementary Figure 5b** and new **Supplementary Figure 6**).

Major point 15 raised by Reviewer #1: Fig. 7B. The arbitrary choice of gene couples is highly questionable as it comes across as a pick the winner situation. Authors should strongly consider to test individual genes or large genesets covering all genes of interest instead, unless they can rationally justify their picks. As many papers previously linked large type I IFN signatures (as opposed to single ISGs, which have contradictory outputs) to detrimental disease outcome in patients with BC (see 19001271 or 31646105), it would also more interesting to see whether these signatures correlate with *KDM1B* levels or with stemness signatures in the same datasets. In the opinion of this reviewer surely more interesting that prognostic assessments based on gene couples.

Our response: Following the suggestion of the referee, instead of gene pairs, we analysed multiple IFN-I signatures, including those described in the article suggested by the reviewer (PMID: 31646105 and PMID: 19001271) and the “viral mimicry” signature characterized in our previous work (PMID: 25344738) and consisting of *Ifnb1*, *Mx1*, *Oas2*, *Cxcl10*, *Dhx58* and *Il12*, as well as multiple stem cell signatures. We included these results in the revised manuscript.

In particular, we revealed a positive correlation between the expression of *KDM1B* and the stem cells signatures characterized in PMID: 21169407 and PMID: 22909066 and a signature comprising the Yamanaka factors (*KLF4*, *MYC*, *OCT3/4*, *SOX2*) in distinct breast cancer (BC) patient datasets (at least in two out of three in each dataset reported in **Figure 7a**). On the contrary, we observed a weak positive correlation between *KDM1B* the “viral mimicry” signatures only in some datasets. These data are shown below (panel **a**) and in the new **Figure 7a**. Similar results have been observed in the METABRIC cohort.

We then used these signatures to address the question whether *KDM1B* expression constitute a prognostic or a predictive factor, as requested by reviewer 2. Using the METABRIC cohort, we found that *KDM1B* is not an independent prognostic factor but is associated with dismal prognosis (lower DSS and

DRFI) when combined with IFN-I-related and stem cell-related signatures. These results are shown below (panel **b** and **c**) and have been incorporated into new **Figure 7b** and new **Supplementary Figure 7b**.

Major point 16 raised by Reviewer #1: Fig. 7F. CSC increase is observed in a minority of patients post-therapy (15%), perhaps suggesting that optimally dosed anthracycline-based chemotherapy in patients most often prevents (instead of promoting) CSC accumulation. In view of previous data from Sistigu et al (suggesting that BC patients with high MX1 levels benefit from anthracyclines), these findings may actually support a model where strong type I IFN signaling upon chemotherapy (as documented by high MX1 levels, and as opposed to indolent in pre-treatment biopsies) may be beneficial and most often prevent CSC accumulation. I would tone down conclusions in this respect (see my first point above) and include previous findings (these and others mentioned above) in the discussion.

Our response: Driven by the reviewer's constructive critique, we reduced our emphasis and discussed our findings accordingly. See also our response to major point 13 of Reviewer #2.

The Reviewer also raised a series of minor points.

Minor point 1 raised by Reviewer #1: Abstract. Authors should refrain from using the word "immortal".

Our response: In the revised version of our manuscript, the sentence "Cancer stem cells (CSCs) are immature, immortal cells within tumors, adept at resisting therapeutic pressure and responsible for local and distant disease recurrence" has been corrected with "Cancer stem cells (CSCs) are a subpopulation of cancer cells endowed with high tumorigenic, chemoresistant and metastatic potential", as also suggested by reviewer #2.

Minor point 2 raised by Reviewer #1: *Introduction. The first paragraph is very generic and should be shortened/refocused.*

Our response: Driven by the reviewer's critique, we have shortened this part of the Introduction to focus immediately on CSCs.

Minor point 3 raised by Reviewer #1: *Introduction. The authors point to CSCs as "highly proliferating" citing a review from Battle and Clevers. This is not the most commonly accepted view for CSCs and even Battle and Edwards suggest that CSCs largely evade treatment by adopting a quiescent phenotype. I would rephrase to stay out of controversy.*

Our response: In the revised version, this part of the Introduction has been rephrased.

Minor point 4 raised by Reviewer #1: *Introduction. Authors use reference 24 to support the concept that type I IFN can mediate pro-tumorigenic effects. Reference 24 demonstrates that the rejection of mouse cancer cells by immunocompetent hosts relies on type I IFN, and hence does not support the author sentence.*

Our response: As correctly suggested by the reviewer, in the manuscript from Dunn *et al* (PMID: 15951814) host IFN- α/β are reported to prevent the outgrowth of primary carcinogen methylcholanthrene-induced tumors, but also to edit less immunogenic cancer cell variants through indirect, host-related hematopoietic cell effects. Hence, we cited this paper to underlie the tumor promoting role of IFNs-I. However, as this role is not cancer-cell autonomous, in the revised version of the manuscript we removed the reference.

Minor point 5 raised by Reviewer #1: *Introduction. The authors should avoid introducing "neologisms" in original articles. Those are better suited for potential commentaries post-publication.*

Our response: Driven by the reviewer's critique, we deleted the sentence "regress to progress" from the Introduction.

Minor point 6 raised by Reviewer #1: *Introduction. Type I IFN should not be called a chemoattractant, that's a secondary effect mediated by CXCL9/10 (as indicated by the authors).*

Our response: Following the reviewer's suggestion, we rephrased the sentence accordingly from "These include Type I interferons (IFNs-I), a family of cytokines that act as chemoattractants for T cells by triggering the production of the IFN-stimulated gene (ISG) C-X-C motif chemokine ligand 10 (CXCL10) upon binding to interferon alpha and beta receptor (IFNAR)" into "Type I interferons (IFNs-I) are a family of pro-inflammatory cytokines which, upon binding to the interferon alpha and beta receptor (IFNAR), trigger

the production of the IFN-stimulated gene (ISG) C-X-C motif chemokine ligand 10 (CXCL10), in turn acting as chemoattractant for inflammatory monocytes and T cells”.

Minor point 7 raised by Reviewer #1: *Results. Gating on positive and negative cells is often across main populations. Although this is obviously appropriate, this reviewer suggest to include in supplementary material a gating strategy that include sample with no secondary antibody, so that gating approach can be nicely appreciated by readers.*

Our response: The gating strategy has been better explained in the new **Figure 1a** and in **Extended Data 1**.

Minor point 8 raised by Reviewer #1: *Suppl. Fig. 2B. VRP only is missing in MCA205 cells.*

Our response: In the revised version of the manuscript, we moved old **Figure 2f** in new **Supplementary Figure 2d** and removed old **Supplementary Figure 2b,c** because the results presented were redundant.

Minor point 9 raised by Reviewer #1: *Suppl. Fig. 2D. Quantitative data should be included.*

Our response: Driven by the reviewer’s critique, quantification has been included in the new **Supplementary Figure 2e**.

Minor point 10 raised by Reviewer #1: *Fig. 3A. OXP treatment does not increase the percentage of CD44H cells, which is internally incoherent with 2B.*

Our response: OXP treatment instead induces both CD44H and CD44L increase, please compare column 1 with column 6 of original graph. In the revised version we reshaped the graph (also by checking data and performing additional experiments as suggested by reviewer #2 point 8) and plotted separately CD44L and CD44H cells, so results stand out easier.

Minor point 11 raised by Reviewer #1: *Fig. 3B. A representative picture with less red background would be preferable if available.*

Our response: In the revised version, background from **Figure 3b** has been reduced.

Minor point 12 raised by Reviewer #1: *Fig. 4A. Tumor growth curves should be compared as whole by two-way anova computing time and treatment as variables, not at specific time points.*

Our response: Driven by the reviewer’s critique, we have calculated stats as suggested (new **Figure 4a**).

Minor point 13 raised by Reviewer #1: *Fig. 4B. It's unclear and add little, I would remove it or present it differently.*

Our response: Following the reviewer's suggestion, we removed these data from **Figure 4** and **Supplementary Figure 4**.

Minor point 14 raised by Reviewer #1: *Page 13. Please avoid using "fleeting".*

Our response: In the revised version of our manuscript, we corrected "fleeting" with "transient".

Minor point 15 raised by Reviewer #1: *Most of the genes mentioned in page 15 are stemness-related genes, not surprising that they are upregulated in the CSC compartment. I suggest refocusing this discussion on immune genes of interest instead.*

Our response: Driven by the reviewer's critique, we focused more our discussion on immune related genes.

Minor point 16 raised by Reviewer #1: *An entire field would comment on the wrong use of "autophagic cell death" in Fig. 6e.*

Our response: We apologize for this mistake. In the revised version, autophagic cell death has been corrected with macroautophagy.

Minor point 17 raised by Reviewer #1: *Fig. 6H should be removed as alternative models may equally explain results. This is more appropriate for commentary-like pieces.*

Our response: In the revised version, schematic model has been removed.

Minor point 18 raised by Reviewer #1: *Fig. 7A. What about merged analysis of all datasets alongside? That would surely be beneficial from a statistical standpoint if results are consistent across cohorts.*

Our response: We agree with the reviewer that merging the cohorts could, in principle, improve the statistics. However, the analysis of these cohorts was performed with different sample preparation and sequencing instrumentations. This makes it problematic to merge data reliably, so we decided to leave the cohorts separated.

Point-by-point reply to the Reviewer #2

Remarks to the Authors: *In this study, the team investigated the role type 1 IFN in CSCs. They propose that nucleic acids are released from tumor cells exposed to immunogenic chemotherapy. These nucleic acids, which are encapsulated in extracellular vesicles or not, are transported to by-standing tumor cells and induce type 1 IFN production via activating nucleic acid sensors. Moreover, type 1 IFN induce CSC enrichment by upregulating KDM1B, which make the elements in the chromatin more accessible to a number of TFs, such as HTH and bHLH. Overall, I find this an interesting yet preliminary study. The detrimental effects of IFN on tumor treatments have been reported. Moreover, the general idea that nucleic acids from immunogenic dead cells induce type 1 IFN secretion has also been shown in their previous study (Sistigu et al Nature Medicine 2014). However, their core conclusion here, that IFN promotes CSCs via upregulating chromatin remodeling factor KDM1B is potentially a novel and exciting finding. The major issues of this study are that the evidences for CSC enrichment are not robust, and the role of KDM1B has not been rigorously demonstrated.*

Our response: We thank the reviewer for his/her appraisal of our work and appreciate his/her very detailed and constructive criticism to which we responded point-by-point by providing more mechanistic insights into our observations, and by validating and translating the relevance of preclinical data in breast cancer (BC) patients. In particular, as the reviewer will appreciate, we provide further evidence of the stem-like nature of the subpopulation of cells enriched by immunogenic chemotherapy and performed a large number of experiments (including cell engineering, *in vivo* analysis of tumorigenicity, immunogenicity and metastatic potential, and ATAC-/Chip-seq studies) clearly demonstrating the role of Type I IFN and KDM1B in CSC enrichment.

Major point 1 raised by Reviewer #2: *They propose that IFN increases two CD44L and CD44H CSC subsets in MCA205 cells. However, the tumorigenicity of the CD44L cells was consistently lower than the one of their parental cells in Fig 4c (0% vs 66% 105 in immunocompetent mice; 0% vs 50% 102 in NSG...). Even the tumorigenicity of CD44H cells was worse than the one of their parental cells at the 102 level in NSG mice. The strong tumorigenicity is the central feature of CSCs. How to explain this discrepancy?*

Our response: The point raised by the reviewer is very well taken. As outlined in our response to major point 11 raised by Reviewer #1, we extended our *in vivo* analyses to 12-to-15 mice per group, providing strong evidence of the low tumorigenic potential of CD44L cells in both immunocompetent and immunodeficient hosts (*i.e.*, these cells behave as non-CSCs) and the high tumorigenic potential of CD44H in both immunocompetent and immunodeficient hosts (*i.e.*, these cells behave as CSCs).

In more detail, we observed that 1×10^5 CD44L cells developed tumors in 3/15 (20%) immunocompetent mice while parental (PAR) cells in 9/15 (60%) mice and that 1×10^4 CD44L cells did not develop tumors in immunocompetent mice while PAR cells engrafted in 1/12 (8%) mice. In these experiment, CD44H cells developed tumors in immunocompetent mice always more frequently than PAR or CD44L cells, thus confirming their high tumorigenic and immunoevasive nature.

Moreover, when injected in immunodeficient NSG mice, 1×10^4 , 1×10^3 and 1×10^2 CD44L cells developed tumors, respectively, in 14/15 (93%), 7/15 (47%) and 0/15 (0%) of cases, while PAR cells in 12/12 (100%), 7/12 (58%), and 2/12 (17%). Again, CD44H cells proved more tumorigenic, as 1×10^3 and 1×10^2 CD44H developed tumors in 12/15 (80%) and 3/15 (20%) of cases. We ascribe the decrease in the tumorigenicity of CD44H at 1×10^2 (which, however, remains slightly higher than that of 1×10^2 PAR cells) to the fact that these cells were injected into mice upon FACS sorting, and, possibly, when sorted at low number had more difficulty to fully recover upon this procedure.

Lending further support to their CSC nature, CD44H (but not CD44L) cells displayed stem-like properties in all the series of experiments of phenotypic characterization (**Figure 1**), immunogenicity (**Figure 4**, **Figure 5** and **Supplementary Figure 4**), therapy resistance (**Figure 4**), and metastatic potential (**Figure 4**) performed in this study.

Altogether these results indicate that CD44H is the true CSC subpopulation while CD44L are not CSCs. They also suggest the intriguing hypothesis that we are now exploring, suggesting that IFNs-I can contribute to intratumor heterogeneity in response to microenvironmental changes by promoting the emergence of cancer cell variants showing different behaviours and stemness properties.

Major point 2 raised by Reviewer #2: *The numbers of representative plots are inconsistent to the quantitations. For example, AT3 IFNs-CD24H and IFNs-CD24L are 26% and 13% respectively in representative plots of Fig 1a. However, their means in quantitations below are about 10% and 2.5% respectively, whereas the SEM is very small. I recommend to repeat these experiments.*

Our response: Driven by the reviewer's remark, we modified **Figure 1a**, clearly explaining the gating strategies we used for data analysis and quantification. To help the reader follow our approach, we also used a specific color code and reported the percentage of the putative CSC population within the bottom plots.

Specifically, in this figure we quantified the percentage of CD133⁺ cells within the two fractions of CD24⁺CD44⁺ cells: CD24⁺CD44^{high} (dark violet) and CD24⁺CD44^{low} (light violet) for MCA205 cells and CD44⁺CD24^{high} (light orange) or CD44⁺CD24^{low} (dark orange) for AT3 cells.

To be more explicit, for example for MCA205 the percentage of CD133⁺CD24⁺CD44^{low} cells (named CD44L throughout the text) is calculated as follows: $(0.4\% \times 39\%) / 100 = 0.16\%$ in CTR group and $(1.7\% \times 63\%) / 100 = 1.07\%$ in IFN-I group, while the percentage of CD133⁺CD24⁺CD44^{high} cells (CD44H) is calculated as follows: $(2.0\% \times 36\%) / 100 = 0.72\%$ in CTR group and $(9.1\% \times 69\%) / 100 = 6.28\%$ in IFN-I group. In addition, we showed gating strategy in new **Extended Data 1**.

Major point 3 raised by Reviewer #2: *Additional experiments, such as assays for ALDH1 and asymmetric division (PKH26 and Numb staining), are required to verify the features of CSCs at various junctions.*

Our response: Driven by this remark, we conducted a series of novel experiments.

We first assessed asymmetric division by analysing, via immunofluorescence microscopy, the level of Numb in FACS-sorted MCA205 CD44H and CD44L ICD-CSCs. Through image analysis quantification of the integrated fluorescent signal in the two daughter cells deriving from cell divisions, we found that Numb segregates asymmetrically in 40% of mitoses in CD44H cells. On the contrary, in 96% of the cases CD44L divide in a symmetric fashion (*i.e.*, the intensity of Numb is similar in the two daughter cells).

To further confirm this result, we performed automated time-lapse fluorescent microscope analyses on CD44H cells upon PKH26 staining to track cell division, confirming asymmetric PKH26 distribution in daughter cells.

These results further support our *in vitro* and *in vivo* observation that stemness is a peculiar feature of CD44H cells (*see our response to major point 1*).

These data are shown below and in the new **Figure 4e,f**.

We also performed flow cytometry analysis of Aldefluor staining in multiple cell lines, namely AT3, MCA205, B16.F10 and CT26. However, as reported in the figure below, we could detect ALDH⁺ cell populations only in AT3 cells. Moreover, we did not observe any differences in the percentage of ALDH⁺ cells between PAR cells and CD24L ICD-CSCs from “donor”-“receiving” co-cultures (35.2% vs 28.1%), and even FACS-isolated CD24L ICD-CSCs presented similar percentage of ALDH⁺ cells (31.8%). Concerning

other cell lines, MCA205, B16.F10 and CT26 cells were always negative to Aldefluor staining with no induction of ALDH⁺ cells in any FACS-sorted ICD-CSCs analysed. These results indicate that, at least in our setting, ALDH activity is not a reliable CSC marker as it does not correlate to all the other CSC phenotypic and functional features we analysed, including *in vivo* tumorigenic potential, metastatic potential, drug resistance, and immune privilege. This finding is in line with some previous studies arguing against ALDH as a universal marker for CSCs (PMID: 20505780; PMID: 31015586). Of note, in a recent study with various cell lines including CT26, it has been shown that the absence of AldeRed signals may depend on reagent efflux through the ABC transporters MDR1, BCRP and MRP (PMID: 31015586). Our hypothesis, which we did not test experimentally, is that AldeRed reagents are not retained by cells, thus explaining the negative staining. Irrespective of this result, in this study we provided compelling evidence on the enrichment of CSCs upon immunogenic chemotherapy and, by analysing phenotypic and functional markers, on the CSC nature of these cells (see also our response to major point 1).

Major point 4 raised by Reviewer #2: *I am surprised that the authors used MDA-MB-231 to investigate the CSCs. Almost all of MDA-MB-231 cells exhibit a CD44⁺CD24⁻ CSC-like phenotype. Therefore, few CSC study use MDA-MB-231. I recommend the cell lines commonly used in this scenario, including MCF-7, MCF-10A or HMLER.*

Our response: As suggested by the reviewer, we exposed MCF10A (normal breast cell line), MCF7 (breast carcinoma cell line) and HMLER (breast carcinoma cell line enriched in CSCs) to recombinant human IFN- α 2a before analysing the expression of standard human CSC markers by flow cytometry. We found that IFN-I treatment does not change the phenotype of MCF10A cells (which have a high basal CD44⁺CD24^{-low}

population, as already reported in PMID: 26147507 and PMID: 23318426, among the others) and of HMLER cells, while it increases the percentage of CD44⁺CD24^{-low} and CD44v6⁺CD24^{-low} in MCF7 cells. These results, which have been summarized here and in the new **Supplementary Figure 1b**, indicate that IFNs-I are able to enrich for CSCs in neoplastic cells. We speculate that when the percentage of CSCs is already very high (as in the case of HMLER cells), this increase is not appreciable.

Major point 5 raised by Reviewer #2: *Whether the immunoactivating ISGs, such as IFN, IL-12 and MHC, increase in IFN-treated tumor cells?*

Our response: To address this point, we performed *ex vivo* qRT-PCR analysis to determine *Ifnb1* and *Il12* gene expression at day 2 and at day 15 post treatment with a single injection of acute high levels (1×10^5 U/mouse) IFNs-I or at the same time points during exposure to chronic low levels (2×10^4 U/mouse, every other day) IFNs-I. For more experimental details please see our response to major point 1 by Reviewer#1. We found a high increase in levels of *Il-12* and to a lesser extent of *Ifnb1* at day 2 post treatment with IFN-I 100 and IFN-I 20. Of note, the levels of *Il-12* (but not *Ifnb1*) decreased but remained relatively high after 15 days from the treatment, as shown below. By flow cytometry, we also validated the widely known role of IFNs-I to increase MHC-I surface expression, as shown below.

Major point 6 raised by Reviewer #2: In Fig.2, what is the rationale to investigate these selective nucleic acid sensors? Have they excluded the effects of other DNA (such AIM2) or RNA (RIG-I) sensors?

Our response: The referee correctly suggests that the nucleic acid sensors AIM2 and RIG-I could also play a role in CSC enrichment. To abrogate AIM2 signaling, we added the anti-inflammatory drug thalidomide (AIM2 inh) to our “donor”-“receiving” co-culture setting. As specific pharmacological inhibitors of RIG-I were not available commercially, we took advantage of the TBK1/IKKε inhibitors amlexanox (RIG-I inh#1), BX795 (RIG-I inh#2) and MRT67307 (RIG-I inh#3), which has been previously used in other studies (see PMID: 28588282 and PMID: 32541772), in both MCA205 cells and *Ifih*^{-/-} cells (note that in this latter case RIG-I is the sole dsRNA sensor). As shown below, we found that abrogation of AIM2 signaling and RIG-I→MAVS→TBK1/IKKε axis significantly reduces, but do not completely abrogate CD44L and CD44H ICD-CSCs enrichment as compared to OXP treatment alone. These findings further corroborate our results indicating that all pathways upstream of IFN-I production are involved in CSC induction, meaning that genetic silencing or pharmacological inhibition of one of these pathways impairs, but not abrogates, CSC enrichment, unless combined with the absence of downstream IFNAR or KDM1B signaling.

These data have been added in the new **Supplementary Figure 2c** and **Extended Data 2**.

Major point 7 raised by Reviewer #2: They propose that DNA released by dead cells activates STING in bystanding CSCs. How can DNA in the endosomes activate cytosolic cGAS?

Our response: We agree with the reviewer that the activation of STING by exogenous internalized DNA is surprising (but original). To address this issue, we measured 2'3' cyclic GMP-AMP (cGAMP) levels in MCA205 cell lysates at different time points upon co-culture with DNA isolated from OXP-treated MCA205 cells. Using this experimental design, we detected significantly increased levels of cGAMP 8h post stimulation with exogenous, yet self, DNA (see Figure below), which is probably internalized through endosomes and then released in the cytosol where it activates cGAS. Despite the clarity of our data, we are aware that we do not provide direct evidence of endosome cargo release in acceptor cells. As a better understanding of this mechanism would be of great interest, we are intended to examine it in depth in *ad hoc* subsequent studies.

Major point 8 raised by Reviewer #2: *Although the p values are not shown, RNase A, RNase H and DNase markedly increase CD44H and CD44L % without OXP (3rd, 4th and 5th groups in Fig3a). Is this effect really ICD-specific?*

Our response: Driven by the reviewer's remark, we increased the number of replicates and performed statistical analyses confirming that the enzymes alone do not induce CSC surface markers. These data are shown below and in new **Figure 3a** and discussed in the text.

Major point 9 raised by Reviewer #2: They show that EVs contain mRNAs for reprogramming factors and invasion molecules (Fig. 3e). What do they want to propose? mRNAs in EVs are translated into pro-tumor proteins in recipient cells? Any evidences? How to reconcile with the nucleic acid-sensing mechanism?

Our response: The reviewer raises a highly interesting point. It has long been known that EVs can carry genetic information. In particular, the so-called oncosomes are shown to horizontally transfer oncogenes in the form of DNA or mRNAs, in turn contributing to malignant transformation and reprogramming of recipient cells (PMID: 25721812, PMID: 11353826). EV transferred genetic cargo has been profiled (PMID: 24398677) and reported to be transcriptionally incompetent in the EVs because of the absence of RNA polymerases and ribosomes (PMID: 17486113). However, once transferred in recipient cells, DNA and mRNAs can theoretically regain transcriptional and translational potential, respectively (PMID: 17486113). We thus speculate that in our system two different mechanisms might act in concert to reprogram cancer cells toward stemness: (i) a direct mechanism, as “donor” cells might transfer and induce specific stem-related signaling pathways that “receiving” cells take up and transcribe/translate; (ii) an indirect mechanism, as transferred nucleic acids induce IFN-I production and thus, downstream, KDM1B-related reprogramming.

This hypothesis is discussed in the revised manuscript.

Major point 10 raised by Reviewer #2: The animal number in each group is too small (n=2-4) in Fig4b,c, and e. This is probably why the CD44H is more tumorigenic in the NSG 103 group, while less tumorigenic in the 102 group compared with parental cells. It is recommended that these experiments are repeated with n=6 from at least two separate experiments, for purposes of reproducibility.

Our response: Driven by the suggestion of both referees, we extended our *in vivo* analyses to 12-to-15 mice per group. We were able to confirm that CD44H ICD-CSCs are significantly more tumorigenic than

CD44L ICD-CSCs and PAR cells both in immunocompetent and immunodeficient hosts. Indeed, 1×10^5 CD44H cells developed tumors in 14/15 (93%) immunocompetent mice, while PAR and CD44L cells in 9/15 (60%) and 3/15 (20%) mice, respectively. Moreover, when injected in immunodeficient NSG mice, 1×10^3 CD44H cells developed tumors in 12/15 (80%) of cases, while PAR and CD44L cells in 7/12 (58%) and 7/15 (47%), respectively. We further corroborated that, in the absence of an intact immune system, 1×10^2 CD44H and PAR cells developed palpable disease in 3/15 (20%) and 2/12 mice (17%), respectively, while CD44L cells did not. Importantly, we also showed that only 4/36 (11%) immunocompetent mice rejecting CD44H ICD-CSCs were vaccinated against viable PAR cells. Conversely, CD44L ICD-CSCs or PAR MCA205 cells endowed animals with more than 50% long-term protection against tumor re-challenge. These data are shown here (panel **a** tumorigenicity, panel **b** challenge) and have been included in the new **Figure 4b** and new **Supplementary Figure 4b**. Please see also our response to major point 1.

Major point 11 raised by Reviewer #2: The evidences for the chromatin remodeling by KDM1B are circumstantial. They show the different motif enrichment for various TFs between CSCs and parental cells. However, whether it is caused by KDM1B or IFN is unknown. Moreover, they only used one inhibitor to show that KDM1B affects CD44H percentages and TIM3 expression in TILs. I recommend to perform ATAC-seq in the gain (overexpression KDM1B in parental cells) and loss (silencing KDM1B by shRNAs or

knockout KDM1B by CRISPR/Cas9 in IFN-treated cells) experiments to show whether KDM1B really regulates the accessibility of these TFs to chromatin.

Our response: As recommended by the reviewer, in this round of revision we investigated in-depth the role of KDM1B in CSC induction. In particular, we generated MCA205 cells engineered with lentiviral vectors to either silence (*Kdm1b*^{KD}) or overexpress this gene (*Kdm1b*^{OVER}), and then we performed ATAC-seq analyses on *Kdm1b*-depleted and or *Kdm1b*-overexpressing MCA205. We found, in *Kdm1b*^{OVER} cells, open peaks for genes involved in cancer stemness (*Klf4*, *Myc*, *Pou5f1*, *Sox2*, *Nanog* and *Hes1*), embryonic development (*Tbx4*), epithelial-to-mesenchymal transition (EMT) (*Gata6* and *Tfcp2*), cancer cell invasiveness and metastatization (*Spire1* and *Trpm4*), tumorigenesis, tumor progression and therapy resistance (*Csf1r*, *Itga*, *Baiap*, and *Slc6a6*) and immune escape (*Gpr17*). Of note, all these peaks were closed or significantly less open in *Kdm1b*^{KD} cells, as shown below and in the new **Supplementary Figure 5c**, thus supporting the epigenetic regulation of these stem- and immune-related genes by KDM1B.

To further address the epigenetic role of KDM1B, we also performed Chip-seq analysis on CD44H ICD-CSCs isolated from MCA205 cells. As the reviewer will appreciate, we deciphered the gene regulatory mechanisms downstream of KDM1B. Specifically, we found that KDM1B interacts with genes involved in stemness maintenance, embryonic development, EMT, invasiveness, wound healing and sprouting angiogenesis, cell-to-cell and cell-to-extracellular matrix (ECM) adhesion, response to stress (DNA damage, hypoxia, starvation), epigenetics, regulation of gene expression (at transcriptional, translational

and post-translational levels), senescence and apoptosis, metabolism, cell cycle and viral signature. These data are shown below and in the new **Supplementary Figure 5d** and **Extended Data 3**.

Major point 12 raised by Reviewer #2: Similarly, the authors should show that whether gain and loss of KDM1B may affect the CSC, drug resistance, tumorigenic and metastatic properties in multiple cell lines.

Our response: Driven by the reviewer’s constructive critique, we generated MCA205, CT26 and B16.F10 cells engineered with lentiviral vectors to either silence (*Kdm1b*^{KD}) or overexpress (*Kdm1b*^{OVER}) this gene, and then run several complementary analyses comparing *Kdm1b*^{KD} and *Kdm1b*^{OVER} genotypes. We used human ΔLNGFR and GFP as reporter genes for *Kdm1b*^{OVER} and *Kdm1b*^{KD} cells, respectively. As the reviewer could appreciate in the western blot below, we failed to engineer only AT3 cells, which were particularly reluctant to lentiviral vectors, but were able to produce KDM1B-depleted and -overexpressing MCA205, CT26 and B16.F10 cells.

First, through *in vivo* analyses in immunocompetent histocompatible mice, we showed that *Kdm1b*^{OVER} cells are extremely more tumorigenic than *Kdm1b*^{KD} cells. Indeed, 1×10^5 *Kdm1b*^{OVER} cells developed tumors in 12/12 (100%) immunocompetent mice, while *Kdm1b*^{KD} cells in 1/12 (8%) mice. Instead, when injected in immunodeficient NSG mice, we did not find any difference in the growth of *Kdm1b*^{OVER} and *Kdm1b*^{KD} cells, with tumors growing after the injection of 1×10^5 , 1×10^4 and 1×10^3 cells. Instead, 100 *Kdm1b*^{OVER} cells developed tumors in 2/6 NSG mice, while *Kdm1b*^{KD} cells do not. Since *Kdm1b*^{KD} cells express GFP as reporter gene, which could be criticized as immunogenic, we also inoculated in mice 1×10^5 *Wt* GFP expressing control MCA205 cells, which as we expected, developed tumors in 6/6 (100%) mice (not shown).

In parallel we performed in vitro ELDA assays on CT26 and B16.F10 cells and found that $Kdm1b^{OVER}$ cells presented a significant higher self-renewal ability than $Kdm1b^{KD}$ cells.

Second, we evaluated the *in vivo* response to chemotherapy of established MCA205 PAR or *Kdm1b*^{OVER} tumors (40-45 mm²) in immunocompetent mice by treating animals with doxorubicin (DOX). Of note, we could not treat *Kdm1b*^{KD} tumors as they grew till 9 mm² and then spontaneously regressed in 12/12 mice. Treatment with DOX failed to reduce tumor growth and to improve mice survival when *Kdm1b* was overexpressed (*Kdm1b*^{OVER}). On the contrary, the treatment was effective in reducing the *in vivo* growth of PAR cells, also improving mice survival.

Intriguingly, *ex vivo* flow cytometric analysis showed that *Kdm1b*^{OVER} explanted tumors have higher basal CD44H CSC percentages compared to *Kdm1b*^{KD} cells and comparable to those of ICD-induced CSCs in *Wt* tumors (see new **Figure 6h** and new **Figure 2e**). Flow cytometry and qRT-PCR in the *in vitro* “donor”-“receiving” co-culture system, confirmed this finding. These data are shown below and have been included in the new **Figure 6h** and new **Supplementary Figure 6a,b**.

Consistent with our *in vivo* approach, *in vitro* analysis on multiple *Kdm1b*^{OVER} and *Kdm1b*^{KD} cell lines, confirmed that tumor cells lacking KDM1B (*Kdm1b*^{KD}) are significantly more sensitive to chemotherapy with OXP, DOX and CDDP than their *Kdm1b*^{OVER} counterparts.

Third, we assessed the metastatic potential of *Kdm1b*^{OVER} and *Kdm1b*^{KD} cells in *in vivo* and *in vitro* assays. First, MCA205 *Kdm1b*^{OVER} and *Kdm1b*^{KD} cells were inoculated in the tail vein of immunocompetent mice and tumor engrafts in the lungs were evaluated two weeks later. *Kdm1b*^{OVER} developed lung metastases, while *Kdm1b*^{KD} cells did not. These results are shown here and have been included in the new **Figure 6h-k**, new **Supplementary Figure 5b** and new **Supplementary Figure 6**.

In a second series of *in vitro* experiments, we compared the invasive potential of $Kdm1b^{OVER}$ and $Kdm1b^{KD}$ cells in MCA205, CT26 and B16.F10 cell lines through the transwell cell migration/invasion assay, using FBS as chemoattractant. We confirmed that KDM1B overproduction endows cancer cells with a significantly high migration ability, as shown below and in the new **Supplementary Figure 6c**.

Major point 13 raised by Reviewer #2: It is very important to show the clinical significance of KDM1B in their paired samples before and after NAC. Whether levels of KDM1B in human tumors are correlated with CSC features? Is it associated with therapeutic response? Is it an independent prognostic factor? Any difference in different breast cancer subtypes? These are the basic questions to address if they want to show the clinical significance of their major finding.

Our response: To address the questions and to cross-validate mouse and human data, we run several complementary analyses, taking also advantage of available databases for prognostic evaluation.

First, we extended the retrospective analyses on publicly available BC patient datasets. In particular, following the advice of the first reviewer, we introduced multiple stem signatures and IFN-I signatures (including the “viral mimicry” signature). As reported in our response to major point 15 of reviewer#1, we found a positive correlation between the expression of *KDM1B* and stem signatures reported in PMID:

21169407 and in PMID: 22909066 as well as with a signature composed of Yamanaka factors (*KLF4*, *MYC*, *OCT3/4*, *SOX2*). These data are shown below (panel **a**) and in the new **Figure 7a**.

Second, we addressed the question whether *KDM1B* expression constitutes a prognostic or a predictive factor. We found that *KDM1B* positivity is not an independent prognostic factor, but is associated with dismal prognosis (lower DSS and DRFI) when combined with IFN-I-related and stem-related signatures. These results are shown below (panel **b** and **c**) and have been incorporated into new **Figure 7b** and new **Supplementary Figure 7b**.

Third, we assessed *KDM1B* expression in our cohort of locally advanced BCs (20 serial biopsies) pre (at diagnosis, T0) and post (at surgery, T1) neoadjuvant FEC60 therapy. *KDM1B* turned out to positively correlate with increased CSC marker (either CD44^{pos}CD24^{low/neg} or CD133^{pos}) Allred scores at T1 (3 cases). In parallel, in 6 out of 7 cases in which CSC positivity was reduced at T1, *KDM1B* scores either were reduced (4 cases) or were already negative at T0 (2 cases). These data are shown below (panel **d**) and have been included in the new **Figure 7c**. Of interest, the 3 patients with increased CSC and *KDM1B* scores, were young (median age 40) and HER2 negative (2 triple negative, 1 luminal A).

Fourth, intrigued by a possible (although very preliminary given the small sample size) link between *KDM1B*-stemness and HER2 absence, we performed a retrospective analysis on the METABRIC cohort (1903 cases). We allocated patients into two main groups: HER2 positive group (HER2^{pos}) and HER2 negative group (HER2^{neg}). *KDM1B* positivity combined with IFN-I signatures and with CSC signatures (as above) was associated with dismal prognosis in HER2^{neg} group. These results are shown below (panel **e** and panel **f**) and have been included in the new **Supplementary Figure 7c,d**.

f

	31646105	19001271	viral mimicry	Yamanaka	21169407	22909066
DSS	ns	ns	ns	ns	ns	ns
HER2 ^{pos}	$P=0.694728$	$P=0.199776$	$P=0.576278$	$P=0.560716$	$P=0.437042$	$P=0.302876$
DRFI	ns	ns	ns	ns	ns	ns
DRFI	$P=0.693709$	$P=0.282513$	$P=0.908589$	$P=0.363076$	$P=0.608721$	$P=0.888847$
DSS	**	**	***	*	***	***
HER2 ^{neg}	$P=0.007595$	$P=0.001782$	$P=0.0001499$	$P=0.038689$	$P=0.000014$	$P=0.000000$
DRFI	*	ns	***	*	***	***
DRFI	$P=0.023796$	$P=0.065748$	$P=0.001081$	$P=0.029248$	$P=0.000174$	$P=0.000000$

Major point 14 raised by Reviewer #2: It is very difficult to accurately assess the CD44+CD24-/low phenotype by IHC. Few study use this approach to evaluate CSCs in tissue. I recommend ALDH1 staining.

Our response: We performed an extensive IHC staining optimization with an anti-hALDH1 antibody (Ab), but we could not detect any specific signal. Driven by these technical difficulties, by the limited number of available sections making impossible a further optimization with other ALDH Abs, by the fact that ALDH seems to be an unreliable marker in our setting (see our response to major point 3) and (above all) by the fact that CD44/CD24 Abs are reported in the “ProteinAtlas” database (proteinatlas.org, see BC IHC) and are routinely used from the pathologists and histologists at the Regina Elena Cancer Institute, with whom we worked in cooperation for the optimization of the IHC staining and the determination of CD44^{pos}CD24^{neg/low} phenotype characterizing CSCs, we decided to pursue the use of the CD44^{pos}CD24^{neg/low} based approach to identify breast CSCs. We also included CD133 because this staining was already optimized and because of the immunotherapeutic relevance of CD133 (PMID: 31636668). We are more than confident of the quality of the staining and our IHC results.

The Reviewer also raised a series of minor points.

Minor point 1 raised by Reviewer #2: *“Cancer stem cells (CSCs) are immature, immortal cells within tumors, adept at resisting therapeutic pressure and responsible for local and distant disease recurrence.” It is better to say “Cancer stem cells (CSCs) are a key population of tumor cells that are highly tumorigenic, chemoresistant and metastatic in many cancer types”*

Our response: In the revised version of our manuscript, this sentence has been corrected as suggested.

Minor point 2 raised by Reviewer #2: *“however molecular insights into this evolutionary process still lack.” It is better to say “ However, molecular insights into the evolutionary process of CSCs are still limited”*

Our response: In the revised version of our manuscript, this sentence has been corrected as suggested.

Minor point 3 raised by Reviewer #2: *Why the Pdl1 band is so smearing in Fig3e? The authors should show the marker bands or add dashes to locate their exact position.*

Our response: Following the suggestions of this reviewer and reviewer#1, we increased the sample size in data from qRT-PCR. Data are now reported in new **Figure 3e**.

Minor point 4 raised by Reviewer #2: *Please provide scale bars in Fig 5h.*

Our response: In the revised version scale bars were properly shown.

Decision Letter, first revision:

Subject: Your manuscript, NI-A31940B

Message: Our ref: NI-A31940B

28th Jun 2022

Dear Dr. Sistigu,

Thank you for your patience as we've prepared the guidelines for final submission of your Nature Immunology manuscript, "Type I IFNs promote cancer cell stemness by triggering the epigenetic regulator KDM1B" (NI-A31940B). Please carefully follow the step-by-step instructions provided in the attached file, and add a response in each row of the table to indicate the changes that you have made. Please also check and comment on any additional marked-up edits we have proposed within the text. Ensuring that each point is addressed will help to ensure that your revised manuscript can be swiftly handed over to our production team.

When you upload your final materials, please include a point-by-point response to any remaining reviewer comments and please make sure to upload your checklist.

If you have not done so already, please alert us to any related manuscripts from your group that are under consideration or in press at other journals, or are being written up for submission to other journals (see: <https://www.nature.com/nature-portfolio/editorial-policies/plagiarism#policy-on-duplicate-publication> for details).

In recognition of the time and expertise our reviewers provide to Nature Immunology's editorial process, we would like to formally acknowledge their contribution to the external peer review of your manuscript entitled "Type I IFNs promote cancer cell stemness by triggering the epigenetic regulator KDM1B". For those reviewers who give their assent, we will be publishing their names alongside the published article.

Nature Immunology offers a Transparent Peer Review option for new original research manuscripts submitted after December 1st, 2019. As part of this initiative, we encourage our authors to support increased transparency into the peer review process by agreeing to have the reviewer comments, author rebuttal letters, and editorial decision letters published as a Supplementary item. When you submit your final files please clearly state in your cover letter whether or not you would like to participate in this initiative. Please note that failure to state your preference will result in delays in accepting your manuscript for publication.

Cover suggestions

As you prepare your final files we encourage you to consider whether you have any images or illustrations that may be appropriate for use on the cover of Nature Immunology.

Covers should be both aesthetically appealing and scientifically relevant, and should be

supplied at the best quality available. Due to the prominence of these images, we do not generally select images featuring faces, children, text, graphs, schematic drawings, or collages on our covers.

Nature Immunology has now transitioned to a unified Rights Collection system which will allow our Author Services team to quickly and easily collect the rights and permissions required to publish your work. Approximately 10 days after your paper is formally accepted, you will receive an email in providing you with a link to complete the grant of rights. If your paper is eligible for Open Access, our Author Services team will also be in touch regarding any additional information that may be required to arrange payment for your article.

Please note that *Nature Immunology* is a Transformative Journal (TJ). Authors may publish their research with us through the traditional subscription access route or make their paper immediately open access through payment of an article-processing charge (APC). Authors will not be required to make a final decision about access to their article until it has been accepted. [Find out more about Transformative Journals](https://www.springernature.com/gp/open-research/transformative-journals).

If you have any questions about costs, Open Access requirements, or our legal forms, please contact ASJournals@springernature.com.

[REDACTED]

Best regards,

Elle Morris
Senior Editorial Assistant
Nature Immunology
Phone: 212 726 9207
Fax: 212 696 9752
E-mail: immunology@us.nature.com

On behalf of

Nick Bernard, PhD
Senior Editor
Nature Immunology

Reviewer #1:

Remarks to the Author:

Journal: Nature Immunology

Manuscript: # NI-A31940B

Title: Type I IFNs promote cancer cell stemness by triggering the epigenetic regulator KDM1B

Authors: Musella et al.

In the manuscript "Type I IFNs promote cancer cell stemness by triggering the epigenetic regulator KDM1B" (#NI-A31940A), Musella and colleagues describe a novel signaling axis whereby type I interferon (IFN) supports the accumulation of cancer stem cells (CSCs) with increased ability to evade immune recognition via epigenetic modifications imposed by lysine demethylase 1B (KDM1B).

The authors embarked in a extraordinary amount of novel experiments to address the concerns of this Reviewer, which are globally cleared, with one important exception (see below). Overall, the message of the article remains important, novel and within scope for Nature Immunology and this reviewer consider the article by Musella and colleagues a very strong candidate for publication in Nature Immunology upon revision.

MAJOR ISSUE

1. IMPORTANT: This Reviewer had concerns on the strong interpretation that the authors provided on their results, specifically the comparison between acute/high-dose and chronic/low-dose type I IFN signaling. While additional experiments have been performed and included in the article to support the aforementioned interpretation, the experimental approaches that were undertaken failed to provide further clarity in this issue. Specifically, this Reviewer is not convinced that in Figure 1F, 1×10^5 U type I vs 2×10^4 U appropriately model high vs low dose in the absence of a reporting system that actually measures IFNAR signaling in the TME over the course of the experiment. One may indeed interpret 2×10^4 U delivered over the course of the experiment as chronic high-dose (not low-dose) and 1×10^5 U delivered only initially as acute high-dose followed by prolonged chronic signaling as per progressive local type I IFN degradation. Similar considerations can be

done for Figure 2F, especially (but not exclusively) in the absence of a comparison with DOX + high acute IFN on efficacy.

This is not central to the message of the paper, so this Reviewer believes NO additional experiments are needed. However, authors must attentively rephrase their manuscript throughout (including results and discussion) and relabel their figures to avoid direct reference to high vs low dose and acute vs chronic exposure. Both doses and number of administrations should instead be reported and discussed objectively, e.g., one single dose of 1×10^5 U type I IFN (instead of acute high-dose).

MINOR ISSUES

1. Reference 12 is outdated and could be replaced or complemented with Kroemer et al. Nat Immunol 2022.

2. Using singular for type I IFN (IFN-I) in all instances may be preferable than using plural only once in the abstract and introduction

3. The discussion section would benefit from some restructuring in favor of (1) removing overlap with the results section (in some cases data are represented with excessive degree of granularity), and (2) additional discussion of study caveats.

4. This Reviewer noted a considerable increase in the number of authors compared to the original manuscript. While such an increase may be justified given the extraordinary amount of work performed during the revision, this Reviewer strongly suggest to alter the author contribution statement to include specific information. For instance, I would suggest that, for authors who performed experiments, the type of experiments be indicated instead of very generic sentences like "with the help of N.M., C.G., E.M., G.M., F.G. L.M., S.S.A.R., D.M., M.S.; A.G., M.S., M.P., M.P., G.C., M.F.,". This will surely help appreciating the input of every author to this excellent work.

Reviewer #2:

Remarks to the Author:

The manuscript has been greatly improved after the revision. I still have two suggestions. The revised data show that abrogation of AIM2 signaling significantly reduced CD44L and CD44H ICD-CSCs enrichment, suggesting inhibition of one of these DNA-sensing pathways impairs CSC enrichment (major point 6 raised by Reviewer #2) . I suggest the authors to add AIM2 in the schematic in Fig. 2a. As the authors provide no experimental data to answer whether or how exogenous DNA is internalized through endosomes and then released in the cytosol (major point 7 raised by Reviewer #2) , please at least provide discussion about the potential mechanisms.

Author Rebuttal, first revision:

Point-by-point reply to the Reviewer #1

Remarks to the Authors: In the manuscript "Type I IFNs promote cancer cell stemness by triggering the epigenetic regulator KDM1B" (#NI-A31940A), Musella and colleagues describe a novel signaling axis whereby type I interferon (IFN) supports the accumulation of cancer stem cells (CSCs) with increased

ability to evade immune recognition via epigenetic modifications imposed by lysine demethylase 1B (KDM1B).

The authors embarked in a extraordinary amount of novel experiments to address the concerns of this Reviewer, which are globally cleared, with one important exception (see below). Overall, the message of the article remains important, novel and within scope for Nature Immunology and this reviewer consider the article by Musella and colleagues a very strong candidate for publication in Nature Immunology upon revision.

Our response: We thank the reviewer for his/her positive comments and for noticing (and appreciating) the large amount of work and experimental effort we did during this one year of revision.

Specific comments have been addressed in the response below.

MAJOR ISSUE

1. IMPORTANT: This Reviewer had concerns on the strong interpretation that the authors provided on their results, specifically the comparison between acute/high-dose and chronic/low-dose type I IFN signaling. While additional experiments have been performed and included in the article to support the aforementioned interpretation, the experimental approaches that were undertaken failed to provide further clarity in this issue. Specifically, this Reviewer is not convinced that in Figure 1F, 1×10^5 U type I vs 2×10^4 U appropriately model high vs low dose in the absence of a reporting system that actually measures IFNAR signaling in the TME over the course of the experiment. One may indeed interpret 2×10^4 U delivered over the course of the experiment as chronic high-dose (not low-dose) and 1×10^5 U delivered only initially as acute high-dose followed by prolonged chronic signaling as per progressive local type I IFN degradation. Similar considerations can be done for Figure 2F, especially (but not exclusively) in the absence of a comparison with DOX + high acute IFN on efficacy.

This is not central to the message of the paper, so this Reviewer believes NO additional experiments are needed. However, authors must attentively rephrase their manuscript throughout (including results and discussion) and relabel their figures to avoid direct reference to high vs low dose and acute vs chronic exposure. Both doses and number of administrations should instead be reported and discussed objectively, e.g., one single dose of 1×10^5 U type I IFN (instead of acute high-dose).

Our response: We thank the reviewer for pointing out this misinterpretation of the results, and we agree with him/her about the necessity of a reporting system measuring IFNAR signaling in the TME in order to correctly distinguish the chronic vs. acute IFN-I setting. We rephrased the manuscript changing the description and interpretation of the results, and, in specifically, eliminating the word “acute” and “chronic” and only reporting the precise dose of IFN-I we used and whether we treated cells/animals with a single dose or multiple doses.

MINOR ISSUES

1. Reference 12 is outdated and could be replaced or complemented with Kroemer et al. Nat Immunol 2022.

Our response: The Reference has been replaced.

2. Using singular for type I IFN (IFN-I) in all instances may be preferable than using plural only once in the abstract and introduction

Our response: We modified the abbreviation as suggested by the reviewer.

3. The discussion section would benefit from some restructuring in favor of (1) removing overlap with the results section (in some cases data are represented with excessive degree of granularity), and (2) additional discussion of study caveats.

Our response: We thanks the reviewer for this constructive remark. We shortened the Discussion section, eliminating the redundant description of the results present in the previous version. We have also discussed some study caveats/limitations, including the necessity to confirm our results using (i) reporting systems measuring IFNAR signaling in the TME, (ii) human experimental models, and (iii) a larger cohort of patients with patient follow-up.

Point-by-point reply to the Reviewer #2

Remarks to the Authors: The manuscript has been greatly improved after the revision. I still have two suggestions.

Our response: We thank the reviewer for his/her comments and appreciation of our revised manuscript.

The revised data show that abrogation of AIM2 signaling significantly reduced CD44L and CD44H ICD-CSCs enrichment, suggesting inhibition of one of these DNA-sensing pathways impairs CSC enrichment (major point 6 raised by Reviewer #2. I suggest the authors to add AIM2 in the schematic in Fig. 2a.

Our response: We agree with the reviewer's comment and we modified Figure 2 as suggested.

As the authors provide no experimental data to answer whether or how exogenous DNA is internalized through endosomes and then released in the cytosol (major point 7 raised by Reviewer #2) , please at least provide discussion about the potential mechanisms.

Our response: Driven by the comment of the reviewer, we discussed the potential mechanism of internalization of exogenous DNA which we surmise occurs via extracellular vesicles, but that (as clearly stated in the text) remain to be determined by further experimentation.

Final Decision Letter:

Subject: Decision on Nature Immunology submission NI-A31940C

Message: In reply please quote: NI-A31940C

Dear Dr. Sistigu,

I am delighted to accept your manuscript entitled "Type I IFNs promote cancer cell stemness by triggering the epigenetic regulator KDM1B" for publication in an upcoming issue of Nature Immunology.

Over the next few weeks, your paper will be copyedited to ensure that it conforms to Nature Immunology style. Once your paper is typeset, you will receive an email with a link to choose the appropriate publishing options for your paper and our Author Services team will be in touch regarding any additional information that may be required.

Please note that *Nature Immunology* is a Transformative Journal (TJ). Authors may publish their research with us through the traditional subscription access route or make their paper immediately open access through payment of an article-processing charge (APC). Authors will not be required to make a final decision about access to their article until it has been accepted. [Find out more about Transformative Journals](https://www.springernature.com/gp/open-research/transformative-journals).

Authors may need to take specific actions to achieve [compliance](https://www.springernature.com/gp/open-research/funding/policy-compliance-faqs) with funder and institutional open access mandates. If your research is supported by a funder that requires immediate open access

(e.g. according to [Plan S principles](https://www.springernature.com/gp/open-research/plan-s-compliance)) then you should select the gold OA route, and we will direct you to the compliant route where possible. For authors selecting the subscription publication route, the journal's standard licensing terms will need to be accepted, including [self-archiving policies](https://www.springernature.com/gp/open-research/policies/journal-policies). Those licensing terms will supersede any other terms that the author or any third party may assert apply to any version of the manuscript.

Your paper will be published online soon after we receive your corrections and will appear in print in the next available issue. Content is published online weekly on Mondays and Thursdays, and the embargo is set at 16:00 London time (GMT)/11:00 am US Eastern time (EST) on the day of publication. Now is the time to inform your Public Relations or Press Office about your paper, as they might be interested in promoting its publication. This will allow them time to prepare an accurate and satisfactory press release. Include your manuscript tracking number (NI-A31940C) and the name of the journal, which they will need when they contact our office.

About one week before your paper is published online, we shall be distributing a press release to news organizations worldwide, which may very well include details of your work. We are happy for your institution or funding agency to prepare its own press release, but it must mention the embargo date and Nature Immunology. Our Press Office will contact you closer to the time of publication, but if you or your Press Office have any enquiries in the meantime, please contact press@nature.com.

Also, if you have any spectacular or outstanding figures or graphics associated with your manuscript - though not necessarily included with your submission - we'd be delighted to consider them as candidates for our cover. Simply send an electronic version (accompanied by a hard copy) to us with a possible cover caption enclosed.

If you have not already done so, we strongly recommend that you upload the step-by-step protocols used in this manuscript to the Protocol Exchange. Protocol Exchange is an open online resource that allows researchers to share their detailed experimental know-how. All uploaded protocols are made freely available, assigned DOIs for ease of citation and fully searchable through [nature.com](https://www.nature.com). Protocols can be linked to any publications in which they

are used and will be linked to from your article. You can also establish a dedicated page to collect all your lab Protocols. By uploading your Protocols to Protocol Exchange, you are enabling researchers to more readily reproduce or adapt the methodology you use, as well as increasing the visibility of your protocols and papers. Upload your Protocols at www.nature.com/protocolexchange/. Further information can be found at www.nature.com/protocolexchange/about .

Please note that we encourage the authors to self-archive their manuscript (the accepted version before copy editing) in their institutional repository, and in their funders' archives, six months after publication. Nature Portfolio recognizes the efforts of funding bodies to increase access of the research they fund, and strongly encourages authors to participate in such efforts. For information about our editorial policy, including license agreement and author copyright, please visit www.nature.com/ni/about/ed_policies/index.html

Sincerely,

Nick Bernard, PhD
Senior Editor
Nature Immunology